# Theoretical Analysis of Consistency Regularization with Limited Augmented Data

## Abstract

Data augmentation is popular in the training of large neural networks; currently, however, there is no clear theoretical comparison between different algorithmic choices on how to use augmented data. In this paper, we take a small step in this direction; we present a simple new statistical framework to analyze data augmentation - specifically, one that captures what it means for one input sample to be an augmentation of another, and also the richness of the augmented set. We use this to interpret consistency regularization as a way to reduce function class complexity, and characterize its generalization performance. Specializing this analysis for linear regression shows that consistency regularization has strictly better sample efficiency as compared to empirical risk minimization (ERM) on the augmented set. In addition, we also provide generalization bounds under consistency regularization for logistic regression and two-layer neural networks. We perform experiments that make a clean and apples-to-apples comparison (i.e. with no extra modeling or data tweaks) between ERM and consistency regularization using CIFAR-100 and WideResNet; these demonstrate the superior efficacy of consistency regularization.

## 1 Introduction

Modern machine learning models, especially deep learning models, require abundant training samples. Since data collection and human annotation are expensive, data augmentation has been a ubiquitous practice in creating artificial labeled samples and improving generalization performance. This practice is corroborated by the fact that the semantics of images remain the same through simple translations like obscuring, flipping, rotation, color jitter, rescaling (Shorten & Khoshgoftaar, 2019).

Conventional algorithms use data augmentation to expand the training data set (Krizhevsky et al., 2012; Simard et al., 1998; Cubuk et al., 2018; Simonyan & Zisserman, 2014; He et al., 2016). As an alternative, consistency regularization enforces the model to output similar predictions on the original and augmented samples and contributes to many recent state-of-the-art supervised or semi-supervised algorithms. This idea was first proposed in (Bachman et al., 2014) and popularized by Laine & Aila (2016); Sajjadi et al. (2016), and gained more attention recently with the success of FixMatch (Sohn et al., 2020) for semi-supervised few-shot learning, and AdaMatch (Berthelot et al., 2021) for domain adaptation.

Several recent papers (see e.g. (Chen et al., 2020a; Mei et al., 2021; Lyle et al., 2019)) attempt to provide a theoretical understanding of data augmentation (DA); they focus on establishing that augmenting data saves on the number of labeled samples needed for the same level of accuracy. However, none of these explicitly compare in an apples to apples way the efficacy (in terms of the number of augmented samples) of one algorithmic choice of *how to use the augmented samples* vs another algorithmic choice. Another dimension un-explored in previous work is any characterization of the quality of augmentation.

In this paper, we hope to answer the following research question:

*Is it possible to develop a theoretical framework to compare the sample efficiency of different algorithms that use augmented data?*

We present a new theoretical framework that casts consistency regularization as a way to reduce function class complexity, which immediately connects to the well-established theory on generalization and gives rise to a generalization bound for consistency regularization under a general bounded loss function. When specialized to linear regression, this new theoretical framework shows that the consistency regularization is strictly more sample efficient than empirical risk minimization (ERM) on the augmented dataset. In addition, using this framework, we also provide generalization bounds under consistency regularization for logistic regression, two-layer neural networks, and expansion-based data augmentations.

As summary, our **main contributions** are:

- **A statistical framework of consistency regularization.** We first present a simple new statistical framework to analyze data augmentation - with a formal theoretical definition of data augmentation and its strength. We then use this framework to give a generalization bound of consistency regularization and provide instantiations for linear regression, logistic regression, two-layer neural networks, and expansion-based data augmentations.
- **Theoretically proving the efficacy of consistency regularization.** When specializing our framework with consistency regularization to linear/logistic regression, it yields a strictly smaller generalization error than ERM with the same augmented data.
- **Empirical comparisons between consistency regularization and ERM.** We perform experiments that make a clean and apples-apples comparison (i.e., with no extra modeling or data tweaks) between consistency regularization and ERM using CIFAR-100 and WideResNet. Our empirical results demonstrate the superior efficacy of consistency regularization.

## 2 RELATED WORK

**Empirical findings.** Data augmentation (DA) is an essential recipe for almost every state-of-the-art supervised learning algorithm since the seminal work of (Krizhevsky et al., 2012) (see reference therein (Simard et al., 1998; Cubuk et al., 2018; Simonyan & Zisserman, 2014; He et al., 2016; Kuchnik & Smith, 2018)). It started from adding augmented data to the training samples via (random) perturbations, distortions, scales, crops, rotations, and horizontal flips. More sophisticated variants were subsequently designed; a non-exhaustive list includes Mixup (Zhang et al., 2017), Cutout (DeVries & Taylor, 2017), and Cutmix (Yun et al., 2019). The choice of data augmentation and their combinations require domain knowledge and experts' heuristics, which triggered some automated search algorithm to find the best augmentation strategies (Lim et al., 2019; Cubuk et al., 2019). The effects of different DAs have been systematically explored in (Tensmeyer & Martinez, 2016).

Recent practices not only add augmented data to the training set but also enforce the predictor output to be similar by adding consistency regularization (Bachman et al., 2014; Laine & Aila, 2016; Sohn et al., 2020). One benefit of consistency regularization is the feasibility of exploiting unlabeled data. Therefore input consistency on augmented data also formed a major component to state-of-the-art algorithms for semi-supervised learning (Laine & Aila, 2016; Sajjadi et al., 2016; Sohn et al., 2020; Xie et al., 2020), self-supervised learning (Chen et al., 2020b), and unsupervised domain adaptation (French et al., 2017; Berthelot et al., 2021).

**Theoretical studies.** Many interpret the effect of DA as some form of regularization (He et al., 2019). Some work focuses on linear transformations and linear models (Wu et al., 2020) or kernel classifiers (Dao et al., 2019). Convolutional neural networks by design enforce translation equivariance symmetry (Benton et al., 2020; Li et al., 2019); further studies have hard-coded CNN's invariance or equivariance to rotation (Cohen & Welling, 2016; Marcos et al., 2017; Worrall et al., 2017; Zhou et al., 2017), scaling (Sosnovik et al., 2019; Worrall & Welling, 2019) and other types of transformations.

A line of work view data augmentation as invariant learning by averaging over group actions (Chen et al., 2020a; Mei et al., 2021; Lyle et al., 2019). They consider an ideal setting that is equivalent to ERM with all possible augmented data, bringing a clean mathematical interpretation. We are interested in a more realistic setting with limited augmented data. In this setting, it is crucial to utilize the limited data with proper training methods, the difference of which cannot be revealed under the previous studied settings.

Some more recent work investigates the feature representation learning procedure with DA for self-supervised learning tasks (Wen & Li, 2021; HaoChen et al., 2021; von Kügelgen et al., 2021; Garg & Liang, 2020). Cai et al. (2021); Wei et al. (2021) studied the effect of data augmentation with label propagation. Data augmentation is also deployed to improve robustness (Rajput et al., 2019), to facilitate domain adaptation and domain generalization (Cai et al., 2021; Sagawa et al., 2019) .

## 3   DATA AUGMENTATION CONSISTENCY AND HOW IT LEARNS EFFICIENTLY

In this section, we first formally define data augmentation and introduce the problem setup. We then define data augmentation consistency (DAC) regularization and show how it effectively reduces the function class complexity, which connects to a generalization bound for bounded loss functions via Rademacher complexity. Subsequently, we specialize our general result to linear regression, which firmly shows that the DAC regularization provably learns more efficiently than minimizing the empirical risk on the augmented dataset. The following section will present more applications (including logistic regression, neural network, etc.).

### 3.1   PROBLEM SETUP AND DATA AUGMENTATION

Consider the standard supervised learning problem setup: $\mathbf{x} \in \mathcal{X}$ are input features, and $y \in \mathcal{Y}$ is its label (or response). Let $P^*$ be the true distribution of $(\mathbf{x}, y)$ (i.e., the label distribution follows $y \sim P^*(y|\mathbf{x})$). We can then formally define data augmentation as:

**Definition 1** (Data augmentation). *For any sample $\mathbf{x} \in \mathcal{X}$, we say $\mathbf{x}' \in \mathcal{X}$ is its augmentation, if and only if $P^*(y|\mathbf{x}) = P^*(y|\mathbf{x}')$.*

The definition above specifies what it means for one input sample to be an augmentation of another. While the definition covers any $\mathbf{x}'$ with the same label distribution as $\mathbf{x}$, our results only use the augmented samples that can be achieved via certain transformations (e.g., random cropping, rotation). However, our definition does not cover augmentations that alter the labels (e.g., MixUp (Zhang et al., 2017)).

Now we introduce the learning problem on an augmented dataset: Let $(\mathbf{X}, \mathbf{y}) \in \mathcal{X}^N \times \mathcal{Y}^N$ be a training set consisting of $N$ *i.i.d.* samples. Besides the original $(\mathbf{X}, \mathbf{y})$, each training sample in it is provided with $\alpha$ augmented samples. The input features of the augmented dataset can be written as:

$$\widetilde{\mathcal{A}}(\mathbf{X}) = [\mathbf{x}_1; \cdots ; \mathbf{x}_N; \mathbf{x}_{1,1}; \cdots ; \mathbf{x}_{N,1}; \cdots ; \mathbf{x}_{1,\alpha}; \cdots ; \mathbf{x}_{N,\alpha}] \in \mathcal{X}^{(1+\alpha)N},$$

where $\mathbf{x}_i$ is in the original training set and $\mathbf{x}_{i,j}, \forall j \in [\alpha]$ are the augmentations of $\mathbf{x}_i$. The labels of the augmented samples are kept the same, which can be denoted as $\widetilde{\mathbf{M}}\mathbf{y} \in \mathcal{Y}^{(1+\alpha)N}$, where $\widetilde{\mathbf{M}} \in \mathbb{R}^{(1+\alpha)N \times N}$ is a vertical stack of $(1 + \alpha)$ identity mappings. Specifically, when the input $\mathbf{x}_i$s are $d$-dimensional real vectors, we have the following notion of augmentation strength $d_{aug}$:

**Definition 2** (Strength of augmentations). *For any $\delta \in (0, 1)$, let*

$$d_{aug}(\delta) \triangleq \underset{d_{aug}}{\operatorname{argmax}} \ \mathbb{P}_{\widetilde{\mathcal{A}}, \mathbf{X}} \left[ \operatorname{rank}\left( \widetilde{\mathcal{A}}(\mathbf{X}) - \widetilde{\mathbf{M}}\mathbf{X} \right) < d_{aug} \right] \leq \delta, \quad d_{aug} \triangleq d_{aug}\left(1/N\right).$$

Intuitively, *strength of augmentations $d_{aug}(\delta)$* means that with probability at least $1 - \delta$, the augmentations perturb at least $d_{aug}(\delta)$ dimensions; whereas $d_{aug}$ can be intuitively understood as the minimum number of dimensions that the augmentations in $\widetilde{\mathcal{A}}(\mathbf{X})$ perturbed with high probability. A lager $d_{aug}$ corresponds to a stronger data augmentations. For instance, when $\widetilde{\mathcal{A}}(\mathbf{X}) = \widetilde{\mathbf{M}}\mathbf{X}$ almost surely (*e.g.*, when the augmentations are identical copies of the original samples, corresponding to the weakest augmentation – no augmentations at all), we have $d_{aug}(\delta) = d_{aug} = 0$ for all $\delta \in (0, 1)$. On the other hand, if the augmentations are randomly generated, then it is more likely to see larger $d_{aug}$ (i.e., more dimensions being perturbed) with larger $\alpha$ (i.e., more augmentations).

In the next subsection, we formally introduce "data augmentation consistency regularization" and present a generalization bound under bounded loss functions. We subsequently specialize the bound to linear regression and show that consistency regularization is strictly more sample efficient than empirical risk minimization (ERM) on the augmented dataset.

### 3.2 DATA AUGMENTATION CONSISTENCY REGULARIZATION

Let $\mathcal{H} = \{h : \mathcal{X} \to \mathcal{Y}\}$ be a well-specified function class (e.g. for regression problems, $\exists h^* \in \mathcal{H}$, s.t. $h^*(\mathbf{x}) = \mathbb{E}[y|\mathbf{x}]$) that we hope to learn from. Without loss of generality, we assume that each function $h \in \mathcal{H}$ can be expressed as $h = f_h \circ \phi_h$, where $\phi_h \in \Phi = \{\phi : \mathcal{X} \to \mathcal{W}\}$ is a proper representation mapping and $f_h \in \mathcal{F} = \{f : \mathcal{W} \to \mathcal{Y}\}$ is a predictor on top of the learned representation. We tend to decompose $h$ such that $\phi_h$ is a powerful feature extraction function whereas $f_h$ can be as simple as a linear combiner. For instance, in a deep neural network, all the layers before the final prediction layer can be viewed as feature extraction $\phi_h$, and the predictor $f_h$ corresponds to the final linear combination of the features.

For a loss function $l : \mathcal{Y} \times \mathcal{Y} \to \mathbb{R}$ and a metric $\varrho$ properly defined on the representation space $\mathcal{W}$, learning with data augmentation consistency (DAC) regularization can be expressed as:

$$\underset{h \in \mathcal{H}}{\arg\min} \sum_{i=1}^{N} l(h(\mathbf{x}_i), y_i) + \lambda \underbrace{\sum_{j=1}^{\alpha} \sum_{i=1}^{N} \varrho\left(\phi_h(\mathbf{x}_i), \phi_h(\mathbf{x}_{i,j})\right)}_{\text{DAC regularization}}. \tag{1}$$

Note that the DAC regularization in Equation (1) can be easily implemented empirically as a regularizer. Intuitively, DAC regularization penalizes the representation difference between the original sample $\phi_h(\mathbf{x}_i)$ and the augmented sample $\phi_h(\mathbf{x}_{i,j})$, with the belief that similar samples (i.e., original and augmented samples) should have similar representations. When the data augmentations do not alter the labels, it is reasonable to enforce a strong regularization (i.e., $\lambda \to \infty$) - given that the conditional distribution of $y$ does not change. The function $\widehat{h}^{dac}$ learned with data augmentation consistency regularization can then be written as the solution of a constrained optimization problem:

$$\widehat{h}^{dac} \triangleq \underset{h \in \mathcal{H}}{\arg\min} \sum_{i=1}^{N} l(h(\mathbf{x}_i), y_i) \quad \text{s.t.} \quad \phi_h(\mathbf{x}_i) = \phi_h(\mathbf{x}_{i,j}), \ \forall i \in [N], j \in [\alpha]. \tag{2}$$

The constraint in Equation (2) effectively reduces the size of the function class $\mathcal{H}$. To rigorously capture such reduction, we define the following data augmentation consistency (DAC) operator over $\mathcal{H}$. Given the original training samples $\mathbf{X}$ and the augmented dataset $\widetilde{\mathcal{A}}(\mathbf{X})$, we have:

**Definition 3** (Data Augmentation Consistency Operator).

$$T_{\widetilde{\mathcal{A}}, \mathbf{X}}^{dac}(\mathcal{H}) \triangleq \{h \mid h \in \mathcal{H}, \phi_h(\mathbf{x}_i) = \phi_h(\mathbf{x}_{i,j}), \ \forall i \in [N], j \in [\alpha]\}.$$

Particularly, we assume that a proper representation class $\Phi$ is chosen with respect to the augmentations such that $h^* \in T_{\widetilde{\mathcal{A}}, \mathbf{X}}^{dac}(\mathcal{H})$. The DAC operator maps the original function class $\mathcal{H}$ to a (potentially much smaller) subset $T_{\widetilde{\mathcal{A}}, \mathbf{X}}^{dac}(\mathcal{H})$, where every function $h \in T_{\widetilde{\mathcal{A}}, \mathbf{X}}^{dac}(\mathcal{H})$ gives consistent representation for all samples and their augmentations (i.e., $\phi_h(\mathbf{x}_i) = \phi_h(\mathbf{x}_{i,j})$). It is now clear that with DAC regularization, Equation (2) is effectively learning in the function class $T_{\widetilde{\mathcal{A}}, \mathbf{X}}^{dac}(\mathcal{H})$, which is a subset of $\mathcal{H}$. As we will show in the next subsection, the function class size reduction is the key for efficient learning with data augmentations.

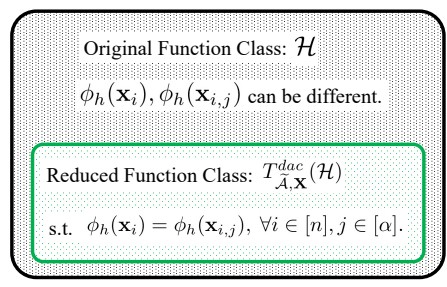

$\phi_h$ is the feature mapping of $h \in \mathcal{H}$

Figure 1: The DAC regularization reduces function class $\mathcal{H}$ to $T_{\widetilde{\mathcal{A}}, \mathbf{X}}^{dac}(\mathcal{H})$.

One of the contributions of our paper is to view consistency regularization as function class complexity reduction. Our next proposition connects the generalization bound to the Rademacher complexity $T_{\widetilde{\mathcal{A}}, \mathbf{X}}^{dac}(\mathcal{H})$ via standard analysis. We further provide various instantiations in the rest of our paper.

Let $L(h)$ denote the population loss induced by $h \in \mathcal{H}$, i.e., $L(h) \triangleq \mathbb{E}_{(\mathbf{x}, y) \sim P^*}[l(h(\mathbf{x}), y)]$.

**Proposition 1** (formally in Proposition 8). *With Equation (2), if $l$ is $B$-bounded and $C_l$-Lipschitz, then for a fixed $T_{\widetilde{\mathcal{A}}, \mathbf{X}}^{dac}(\mathcal{H})$ and any $\delta \in (0, 1)$, with probability at least $1 - \delta$, we have*

$$L(\widehat{h}^{dac}) - L(h^*) \leq 4C_l \cdot \mathfrak{R}_N\left(T_{\widetilde{\mathcal{A}}, \mathbf{X}}^{dac}(\mathcal{H})\right) + \sqrt{\frac{2B^2 \log(2/\delta)}{N}}.$$

**Remark 1.** *Our results can be immediately extended to use unlabeled data. Notice that the DAC regularization only enforces the same predictions for the original and augmented samples, where no ground truth label is needed. For clarity of exposition, we focus on the labeled dataset in the main text and defer discussions on unlabeled data to the appendix.*

### 3.3 EFFICACY OF DAC REGULARIZATION: A GENTLE START WITH LINEAR REGRESSION

To see the efficacy of DAC regularization (i.e., Equation (2)), we revisit a more commonly adopted training method here - empirical risk minimization (ERM) on augmented data:

$$\widehat{h}^{da-erm} \triangleq \underset{h \in \mathcal{H}}{\mathrm{argmin}} \sum_{i=1}^{N} l(h(\mathbf{x}_i), y_i) + \sum_{i=1}^{N} \sum_{j=1}^{\alpha} l(h(\mathbf{x}_{i,j}), y_i). \tag{3}$$

Now we show that the DAC regularization learns more efficiently than ERM. Consider the following setting: given $N$ observations $\mathbf{X} \in \mathbb{R}^{N \times d}$, the responses $\mathbf{y} \in \mathbb{R}^N$ are generated from a linear model $\mathbf{y} = \mathbf{X}\theta^* + \epsilon$, where $\epsilon \in \mathbb{R}^N$ is zero-mean noise with $\mathbb{E}\left[\epsilon\epsilon^\top\right] = \sigma^2 \mathbf{I}_N$. Recall that $\widetilde{\mathcal{A}}(\mathbf{X})$ is the entire augmented dataset, and $\widetilde{\mathbf{M}}\mathbf{y}$ corresponds to the labels. Our next result characterizes the fixed design excess risk of $\theta$ on $\widetilde{\mathcal{A}}(\mathbf{X})$, which is defined as $L(\theta) \triangleq \frac{1}{(1+\alpha)N} \left\| \widetilde{\mathcal{A}}(\mathbf{X})\theta - \widetilde{\mathcal{A}}(\mathbf{X})\theta^* \right\|_2^2$. Under regularity conditions (e.g., $\mathbf{x}$ is sub-Gaussian and $N$ is not too small), it is not hard to extend to random design, i.e., the more commonly acknowledged generalization bound with the same order.

Given $\widetilde{\mathcal{A}}(\mathbf{X})$, notice that by Definition 2, $d_{aug} = \mathrm{rank}\left(\widetilde{\mathcal{A}}(\mathbf{X}) - \widetilde{\mathbf{M}}\mathbf{X}\right)$ since there is no randomness in $\widetilde{\mathcal{A}}, \mathbf{X}$ in fixed design setting. For a linear regression model to be identifiable (i.e., having an unique solution), we assume that $\widetilde{\mathcal{A}}(\mathbf{X})$ has full column rank. We then have the following theorem on the excess risks of learning by DAC regularization and by ERM on the augmented dataset.

**Theorem 2** (Informal result on linear regression (formally in Theorem 6))**.** *Learning with DAC regularization, we have* $\mathbb{E}\left[L(\widehat{\theta}^{dac}) - L(\theta^*)\right] = \frac{(d-d_{aug})\sigma^2}{N}$, *while learning with ERM directly on the augmented dataset, we have* $\mathbb{E}\left[L(\widehat{\theta}^{da-erm}) - L(\theta^*)\right] = \frac{(d-d_{aug}+d')\sigma^2}{N}$, *where* $d' \in [0, d_{aug}]$.

Formally, $d'$ is defined as $d' \triangleq \frac{\mathrm{tr}\left((H_{\widetilde{\mathcal{A}}(\mathbf{X})} - \mathbf{P}_\mathcal{S})\widetilde{\mathbf{M}}\widetilde{\mathbf{M}}^\top\right)}{1+\alpha}$, where $H_{\widetilde{\mathcal{A}}(\mathbf{X})} \triangleq \widetilde{\mathcal{A}}(\mathbf{X})\widetilde{\mathcal{A}}(\mathbf{X})^\dagger$, and $\mathbf{P}_\mathcal{S}$ is the orthogonal projector onto $\mathcal{S} \triangleq \left\{\widetilde{\mathbf{M}}\mathbf{X}\theta \mid \forall \theta \in \mathbb{R}^d, s.t. \left(\widetilde{\mathcal{A}}(\mathbf{X}) - \widetilde{\mathbf{M}}\mathbf{X}\right)\theta = 0\right\}$.

Here we present an explanation for $d'$. Note that $\sigma^2 \cdot \widetilde{\mathbf{M}}\widetilde{\mathbf{M}}^\top$ is the noise covariance matrix of the augmented dataset. We have $\mathrm{tr}\left(\mathbf{P}_\mathcal{S}\widetilde{\mathbf{M}}\widetilde{\mathbf{M}}^\top\right)$ being the variance of $\widehat{\theta}^{dac}$, and $\mathrm{tr}\left(H_{\widetilde{\mathcal{A}}(\mathbf{X})}\widetilde{\mathbf{M}}\widetilde{\mathbf{M}}^\top\right)$ being the variance of $\widehat{\theta}^{da-erm}$. Therefore, $d' \propto \mathrm{tr}\left(\left(H_{\widetilde{\mathcal{A}}(\mathbf{X})} - \mathbf{P}_\mathcal{S}\right)\widetilde{\mathbf{M}}\widetilde{\mathbf{M}}^\top\right)$ is used to measure the difference. When $H_{\widetilde{\mathcal{A}}(\mathbf{X})} \neq \mathbf{P}_\mathcal{S}$ (a common scenario as instantiated in Example 1), we have DAC being strictly better than ERM on augmented data.

**Example 1.** *Consider a 30-dimensional linear regression. The original training set contains 50 samples. The inputs $\mathbf{x}_i$s are generated independently from $\mathcal{N}(0, \mathbf{I}_{30})$ and we set $\theta^* = [\theta_c^*; \mathbf{0}]$ with $\theta_c^* \sim \mathcal{N}(0, \mathbf{I}_5)$ and $\mathbf{0} \in \mathbb{R}^{25}$.*

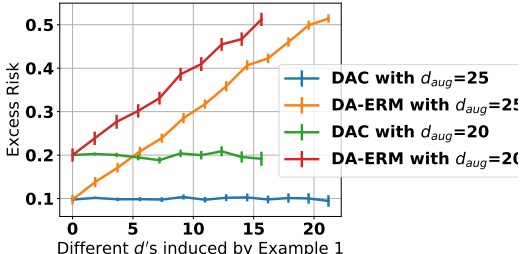

Figure 2: Comparison of DAC regularization and DA-ERM (Example 1). The results precisely match Theorem 2. DA-ERM depends on the $d'$ induced by different augmentations, while the DAC regularization works equally well and better than the DA-ERM. Further, both DAC and DA-ERM are affected by the "strength of augmentation" $d_{aug}$.

*The noise variance $\sigma$ is set to 1. We break $\mathbf{x}$ into 3 parts $[x_{c1}, x_{e1}, x_{e2}]$ and take the following augmentations: $A([x_{c1}; x_{e1}; x_{e2}]) = [x_{c1}; 2x_{e1}; -x_{e2}]$, $x_{c1} \in \mathbb{R}^{d_{c1}}, x_{e1} \in \mathbb{R}^{d_{e1}}, x_{e2} \in \mathbb{R}^{d_{e2}}$, where $d_{c1} + d_{e1} + d_{e2} = 30$.*

*Notice that the augmentation perturbs $x_{e1}$ and $x_{e2}$ and leaving $x_{c1}$ unchanged, we therefore have $d_{aug} = 30 - d_{c1}$. By changing $d_{c1}$ and $d_{e1}$, we can have different augmentations with different $d_{aug}, d'$. The results for $d_{aug} \in \{20, 25\}$ and various $d's$ induced by different $d_{e1}s$ are presented in Figure 2. The excess risks precisely match Theorem 2. It confirms that the DAC regularization is strictly better than ERM on an augmented dataset for a wide variety of augmentations.*

## 4 VARIOUS APPLICATIONS

Now we ground our general result on the DAC regularization with a set of common applications, including the logistic regression, two-layer neural networks, and expansion-based data augmentations. For each of the applications, we first specify the ground truth distribution $P^*$, the function class $\mathcal{H}$, the loss function $l$, as well as the augmented dataset $\widetilde{\mathcal{A}}(\mathbf{X})$. Then, we discuss the corresponding excess risk $L(\widehat{h}^{dac}) - L(h^*)$ for the DAC regularization (Equation (2)). We abridge the analysis in a set of concrete examples with concise arguments while deferring the complete assumptions and theorems to Appendix B. As a supplementary remark, in addition to the popular in-distribution setting where we consider a unique distribution $P^*$ for both training and testing, DAC regularization is also known to improve out-of-distribution generalization for settings like domain adaptation. We defer detailed discussion on such advantage of DAC regularization for linear regression in the domain adaptation setting to Appendix B.4.

### 4.1 LOGISTIC REGRESSION

Here we consider logistic regressions with $\mathcal{X} = \{\mathbf{x} \in \mathbb{R}^d \mid \|\mathbf{x}\|_2 \le D\}$, $\mathcal{Y} = \{0, 1\}$. For some unkown $\boldsymbol{\theta}^* \in \mathbb{R}^d$ with $\|\boldsymbol{\theta}^*\| \le C_0$, we have $P^* (y = 1|\mathbf{x}) = \sigma(\boldsymbol{\theta}^{*\top}\mathbf{x})$, where $\sigma(\cdot)$ is the sigmoid function $\sigma(z) \triangleq 1/(1 + \exp(-z))$. The function class $\mathcal{H}$ is $\mathcal{H} = \{h(\mathbf{x}) = \boldsymbol{\theta}^\top\mathbf{x} \mid \boldsymbol{\theta} \in \mathbb{R}^d, \|\boldsymbol{\theta}\|_2 \le C_0\}$, such that predictions are given by $\widehat{y} = \sigma(\boldsymbol{\theta}^\top\mathbf{x})$. For binary classification, we use the logistic loss $l(\boldsymbol{\theta}^\top\mathbf{x}, y) = -y \log(\sigma(\boldsymbol{\theta}^\top\mathbf{x})) - (1 - y) \log(1 - \sigma(\boldsymbol{\theta}^\top\mathbf{x}))$.

Recall the strength of augmentations $d_{aug}$ from Definition 2. Under proper regularity conditions (i.e., $\mathbf{x}$ is sub-Gaussian and $N$ is not too small, see Appendix B), we have the following generalization bound for learning logistic regression under DAC regularization.

**Theorem 3** (Informal result on logistic regression with DAC (formally in Theorem 9)). *Learning logistic regression with the DAC regularization $h(\mathbf{x}_i) = h(\mathbf{x}_{i,j})$, with high probability:*

$$L(\widehat{h}^{dac}) - L(h^*) \lesssim \sqrt{\frac{d - d_{aug} + \log N}{N}},$$

*where the strength of augmentation $d_{aug} \in [0, d - 1]$.*

Intuitively, with high probability, minimizing empirical loss on augmented samples gives a generalization bound of $L(\widehat{h}^{da-erm}) - L(h^*) \lesssim \max\left(\sqrt{\frac{d}{(\alpha+1)N}}, \sqrt{\frac{d-d_{aug}}{N}}\right)$ at best, where the first term corresponds to the generalization bound for a $d$-dimensional logistic regression with $(\alpha + 1)N$ samples, and the second term follows as the augmentations fail to perturb a $(d - d_{aug})$-dimensional sub-space (and in which ERM can only rely on the $N$ original samples for learning). The first term will dominate the $\max$ when there is limited augmented data (i.e., $\alpha$ is small).

Comparing the two, we see that DAC is more efficient than ERM. In particular, consider the scenario that the limited data augmentations well perturb the data (e.g., $\alpha = 1$ and $d_{aug} = d - 1$). The ERM gives a generalization error that scales as $\sqrt{d/N}$, while DAC yields a dimension-free $\sqrt{1/N}$ error. Please see Appendix E for a supporting numerical example, which demonstrates the benefits of DAC over DA-ERM for logistic regression, and empirically verifies the impact of $d_{aug}$ and $\alpha$ that matches with our theoretical analysis.

### 4.2 TWO-LAYER NEURAL NETWORK

In this section, we discuss a special case where the result for bounded losses in Proposition 8 can be extended to the unbounded square loss. With $\mathcal{X} = \mathbb{R}^d$ and $\mathcal{Y} = \mathbb{R}$, we consider a ground truth

distribution $P^*(y|\mathbf{x})$ induced by a two-layer ReLU network: $y = \left(\mathbf{x}^\top \mathbf{B}^*\right)_+ \mathbf{w}^* + z$, $\mathbf{B}^*_{d \times q} = [\mathbf{b}_1^* \ldots \mathbf{b}_k^* \ldots \mathbf{b}_d^*]$, for some unknown $h^*(\mathbf{x}) \triangleq \left(\mathbf{x}^\top \mathbf{B}^*\right)_+ \mathbf{w}^*$ where $(\cdot)_+ \triangleq \max(0, \cdot)$ element-wisely denotes the ReLU function, $\mathbf{b}_k^* \in \mathbb{S}^{d-1}$ for all $k \in [q]$, and $z \sim \mathcal{N}\left(0, \sigma^2\right)$ is the Gaussian noise. In terms of the function class $\mathcal{H}$, for some constant $C_w \geq \|\mathbf{w}^*\|_1$, we have:

$$\mathcal{H} = \left\{ h(\mathbf{x}) = (\mathbf{x}^\top \mathbf{B})_+ \mathbf{w} \mid \mathbf{B} = [\mathbf{b}_1 \ldots \mathbf{b}_q] \in \mathbb{R}^{d \times q}, \|\mathbf{b}_k\|_2 = 1 \, \forall \, j \in [q], \|\mathbf{w}\|_1 \leq C_w \right\},$$

such that $h^* \in \mathcal{H}$. We use the standard square loss $l(h(\mathbf{x}), y) = \frac{1}{2}(h(\mathbf{x}) - y)^2$.

Let $\mathbf{P}_\mathcal{N}$ be the projector onto the null space of $\widetilde{\mathcal{A}}(\mathbf{X}) - \widetilde{\mathbf{M}}\mathbf{X}$. When the augmented sample set $\widetilde{\mathcal{A}}(\mathbf{X})$ is reasonably diverse (see Appendix B.2), regression over two-layer ReLU networks with the DAC regularization generalizes as following:

**Theorem 4** (Informal result on two-layer neural network with DAC (formally in Theorem 10)). *Assuming* $\mathbb{E}\left[\frac{1}{n}\sum_{i=1}^n \|\mathbf{P}_\mathcal{N}\mathbf{x}_i\|_2^2 \Big| \mathbf{P}_\mathcal{N}\right] \leq C_\mathcal{N}^2$ *for some* $C_\mathcal{N} > 0$, *learning the two-layer ReLU network with the DAC regularization on* $\left(\mathbf{x}_i^\top \mathbf{B}\right)_+ = \left(\mathbf{x}_{i,j}^\top \mathbf{B}\right)_+$ *gives, with high probability:*

$$L\left(\widehat{h}^{dac}\right) - L\left(h^*\right) \lesssim \frac{\sigma C_w C_\mathcal{N}}{\sqrt{N}}.$$

Recall the strength of augmentations $d_{aug}$ from Definition 2. Under proper regularity conditions (*i.e.*, $\mathbf{x}$ is sub-Gaussian and $N$ is reasonably large, see Appendix B.2), we have $C_\mathcal{N} \lesssim \sqrt{d - d_{aug}}$ with high probability. Analogous to the logistic regression example, applying the ERM directly on the augmented samples achieves no better than $L(\widehat{h}^{da-erm}) - L(h^*) \lesssim \sigma C_w \max\left(\sqrt{\frac{d}{(\alpha+1)N}}, \sqrt{\frac{d-d_{aug}}{N}}\right)$ with high probability. The comparison again illustrates the advantage of DAC regularization over ERM. The advantage is large when the limited data augmentations are strong (i.e., large $d_{aug}$ and small $\alpha$).

## 4.3 EXPANSION-BASED DATA AUGMENTATIONS

In this section, we demonstrate that for the classification problems, enforcing consistency on a different notion of data augmentations based on expansion also brings a considerable reduction in complexity of the feasible function class.

Concretely, we consider a multi-class classification problem: for an arbitrary set $\mathcal{X}$, let $\mathcal{Y} = [K]$, and $h^* : \mathcal{X} \to [K]$ be the ground truth classifier that partitions $\mathcal{X}$: for each $k \in [K]$, let $\mathcal{X}_k \triangleq \{\mathbf{x} \in \mathcal{X} \mid h^*(\mathbf{x}) = k\}$, with $\mathcal{X}_i \cap \mathcal{X}_j = \emptyset, \forall i \neq j$. Here we focus on the expansion-based data augmentations defined as following:

**Definition 4** (Expansion-based data augmentations). *Let* $\mathcal{A} : \mathcal{X} \to 2^\mathcal{X}$ *be a function generating a set of augmentations* $\mathcal{A}(\mathbf{x})$ *from a given* $\mathbf{x} \in \mathcal{X}$ *that satisfies the following:*

*(a) Nontrivial augmentation and class invariant:* $\{\mathbf{x}\} \subsetneq \mathcal{A}(\mathbf{x}) \subseteq \{\mathbf{x}' \in \mathcal{X} \mid h^*(\mathbf{x}) = h^*(\mathbf{x}')\}$ *for all* $\mathbf{x} \in \mathcal{X}$; *and*

*(b) Non-trivial expansion: for all* $k \in [K]$, *given any* $\emptyset \subsetneq S \subsetneq \mathcal{X}_k$, *there exists some* $\mathbf{x}' \notin S$ *such that* $\mathcal{A}(\mathbf{x}) \cap \mathcal{A}(\mathbf{x}') \neq \emptyset$.

We consider a general class of classifiers $\mathcal{H} \subseteq \{h : \mathcal{X} \to [K]\}$ where the ground truth classifier is realizable, $h^* \in \mathcal{H}$. With the zero-one loss $l_{01}(h(\mathbf{x}), y) = \mathbf{1}\{h(\mathbf{x}) \neq y\}$, and the corresponding population loss $L_{01}(h) \triangleq \mathbb{E}_\mathbf{x}[\mathbf{1}\{h(\mathbf{x}) \neq h^*(\mathbf{x})\}]$, we learn a classifier from $\mathcal{H}$ with the DAC regularization, $h(\mathbf{x}) = h(\mathbf{x}')$, where $\mathbf{x}'$s are expansion-based data augmentations generated by an $\mathcal{A}$ (Definition 4). As a warm-up, we begin with a simplified setting where we enforce consistency over the population in lieu of a finite set of training samples as in practice. Then, learning with the DAC regularization yields a classifier $\widehat{h}^{dac} \in T_{\mathcal{A},\mathcal{X}}^{dac}(\mathcal{H}) \triangleq \{h \in \mathcal{H} \mid h(\mathbf{x}) = h(\mathbf{x}') \, \forall \, \mathbf{x} \in \mathcal{X}, \, \mathbf{x}' \in \mathcal{A}(\mathbf{x})\}$.

**Theorem 5** (DAC with expansion-based data augmentations over population). *Learning the classifier with DAC regularization over population, with high probability,*

$$L_{01}\left(\widehat{h}^{dac}\right) - L_{01}\left(h^*\right) \leq \frac{K \log K + \log N}{N}.$$

Particularly, with the DAC regularization, the generalization bound of the $K$-class classification problem is dimension-independent but only scales with the number of classes, $\widetilde{O}(K)$.

Furthermore, in Appendix B.3, we extend the result to a more practical setting where the consistency is enforced over a finite training set. Notably, Theorem 5, along with the finite train set case in Appendix B.3, is a reminiscence of Theorem 3.6 and 3.7 by Wei et al. (2021), as well as Theorem 2.1, 2.2, and 2.3 by Cai et al. (2021). We adapt these existing theories and provide a unified analysis under our setting, which demonstrates the generality of our framework.

## 5 EXPERIMENTS

In this section, we empirically verify that training with DAC learns more efficiently than empirical risk minimization on an augmented dataset. The dataset is derived from CIFAR-100, where we randomly select 10,000 labeled data as the training set (i.e., 100 labeled samples per class). During the training time, given a training batch, we generate augmentations by RandAugment (Cubuk et al., 2020). We set the number of augmentations per sample to 7 unless otherwise mentioned.

The experiments focus on comparisons of 1) training with consistency regularization, and 2) empirical risk minimization on the augmented dataset (DA-ERM). In particular, we use the same network architecture (a WideResNet-28-2 (Zagoruyko & Komodakis, 2016)) and the same training settings (e.g., optimizer, learning rate schedule, etc) for both methods. We defer the detailed experiment settings to Appendix D. Our test set is the standard CIFAR-100 test set, and we report the average and standard deviation of the testing accuracy of 5 independent runs. The consistency regularizer is implemented as the $l_2$ distance of the model's predictions on the original and augmented samples.

**Efficacy of DAC regularization.** We first show that the DAC regularization learns more efficiently than ERM on the augmented dataset. The results are listed in Table 1. Notice that with proper choice of $\lambda$ (i.e., the multiplicative coefficient before the DAC regularization, see Equation (1)), training with DAC regularization significantly improves over DA-ERM.

| DA-ERM | DAC Regularization | | | | |
| --- | --- | --- | --- | --- | --- |
| | $\lambda = 0$ | $\lambda = 1$ | $\lambda = 5$ | $\lambda = 10$ | $\lambda = 20$ |
| $69.40 \pm 0.05$ | $62.82 \pm 0.21$ | $68.63 \pm 0.11$ | $\mathbf{70.56 \pm 0.07}$ | $\mathbf{70.52 \pm 0.14}$ | $68.65 \pm 0.27$ |

Table 1: Testing accuracy of ERM and DAC regularization with different $\lambda$'s (regularization coeff.).

**DAC regularization helps more when data is scarce.** Our theoretical results suggest that the DAC regularization brings more benefits when the training data is scarce. The data scarcity can be interpreted in two ways:

*(1) Labeled samples are scarce.* We conduct experiments with different numbers of labeled samples, ranging from 1,000 (i.e., 10 images per class) to 20,000 samples (i.e., 200 images per class). We generate 3 augmentations for each of the samples during the training time, and the results are presented in Table 2. Notice that the DAC regularization gives a bigger improvement over DA-ERM when the labeled samples are scarce. This matches the intuition that when there are sufficient training samples, data augmentation is less necessary. Therefore, the difference between different ways of utilizing the augmented samples becomes diminishing.

| Number of Labeled Data | 1000 | 10000 | 20000 |
| --- | --- | --- | --- |
| DA-ERM | $31.11 \pm 0.30$ | $68.89 \pm 0.07$ | $\mathbf{76.79 \pm 0.13}$ |
| DAC ($\lambda = 10$) | $\mathbf{33.59 \pm 0.41}$ | $\mathbf{70.71 \pm 0.10}$ | $\mathbf{76.86 \pm 0.16}$ |

Table 2: Testing accuracy of ERM and DAC regularization with different numbers of labeled data.

*(2) Augmented samples are scarce.* While keeping the number of labeled samples to be 10,000, we evaluate the performance of the DAC regularization and DA-ERM with different numbers of augmentations. The number of augmentations for each training sample ranges from 1 to 15, and the results are listed in Table 3. The DAC regularization offers a more significant improvement when the number of augmentations is small. This clearly demonstrates that the DAC regularization learns more efficiently from the limited number of augmentations.

| Number of Augmentations | 1 | 3 | 7 | 15 |
|---|---|---|---|---|
| DA-ERM | $67.92 \pm 0.08$ | $69.04 \pm 0.05$ | $69.25 \pm 0.16$ | $69.30 \pm 0.11$ |
| DAC ($\lambda = 10$) | $\mathbf{70.06 \pm 0.08}$ | $\mathbf{70.77 \pm 0.20}$ | $\mathbf{70.74 \pm 0.11}$ | $\mathbf{70.31 \pm 0.12}$ |

Table 3: Testing accuracy of ERM and DAC regularization with different numbers of augmentations.

**Proper augmentation brings good performance.** To achieve good performance, it is important to have proper data augmentation - it needs to be strong such that it well perturbs the input features, but it should also leave the label distribution unchanged. Here we experiment with different augmentation strengths, which is the number of different random transformations (e.g., random cropping, flipping, etc.) applied to the training samples sequentially. More transformations imply stronger augmentations. The number of transformations ranges from 1 to 10, and the results are listed in Table 4. We see that both DA-ERM and the DAC regularization benefits from a proper augmentation. When the augmentation is too strong (e.g., Augmentation Strength 10, as shown in Figure 3), the DAC regularization gives a worse performance. It might be explained by falsely enforcing DAC regularization where the labels of the augmented samples have changed.

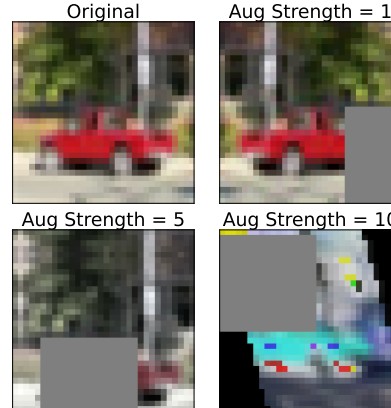

Figure 3: Examples of different augmentation strengths.

| Augmentation Strength | 1 | 2 | 5 | 10 |
|---|---|---|---|---|
| DA-ERM | $68.56 \pm 0.12$ | $69.32 \pm 0.11$ | $\mathbf{69.97 \pm 0.14}$ | $\mathbf{69.66 \pm 0.16}$ |
| DAC ($\lambda = 10$) | $\mathbf{70.66 \pm 0.14}$ | $\mathbf{70.65 \pm 0.07}$ | $70.01 \pm 0.10$ | $68.95 \pm 0.27$ |

Table 4: Testing accuracy of EMR and DAC regularization with various augmentation strengths.

**Combining with a semi-supervised learning algorithm.** Here we show that the DAC regularization can be easily extended to the semi-supervised learning setting. We take the previously established semi-supervised learning method Fix-

| Number of Unlabeled Data | 5000 | 10000 | 20000 |
|---|---|---|---|
| FixMatch | 67.74 | 69.23 | 70.76 |
| FixMatch + DAC ($\lambda = 1$) | $\mathbf{71.24}$ | $\mathbf{72.7}$ | $\mathbf{74.04}$ |

Table 5: DAC regularization helps FixMatch when the unlabeled data is scarce.

Match (Sohn et al., 2020) as the baseline and adapt the FixMatch by combining it with the DAC regularization. Namely, besides using FixMatch to learn from the unlabeled data, we additionally generate augmentations for the labeled samples and apply the DAC regularization. In particular, we focus on the data-scarce regime by only keeping 10,000 labeled samples and at most 20,000 unlabeled samples. Results are listed in Table 5. We see that the DAC regularization also improves the performance of FixMatch when the unlabeled samples are scarce. This again demonstrates the efficiency of learning with DAC regularization.

## 6 CONCLUSION

We present a simple new theoretical framework for understanding the statistical efficiency of consistency regularization with limited data augmentations. In particular, our proposed framework gives a generalization bound for consistency regularization in general cases. We also provide instantiations for linear regression, logistic regression, two-layer neural networks, and expansion-based data augmentations. When specialized to linear regression/logistic regression, it shows that consistency regularization yields a strictly smaller generalization error than ERM with augmented data. We also provide apples-to-apples empirical comparisons between augmented ERM and consistency regularization. These together demonstrate the superior efficacy of consistency regularization.

**Reproducibility Statement** Our main theoretical results, including problem setup and (simplified) theorem statements are in Sections 3 and 4. The formal theorem statements along with the proofs can be found in Appendices A and B. We present empirical results in Section 5, and the detailed experiment setup is in Appendix D. We also submit our experiment code in the supplementary material.

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

## A  PROOFS FOR BOUNDED LOSS AND LINEAR REGRESSION

### A.1  PROOF FOR DAC ON LINEAR REGRESSION

For fixed $\widetilde{\mathcal{A}}(\mathbf{X})$, we recall that $d_{aug} = \mathrm{rank}\left(\widetilde{\mathcal{A}}(\mathbf{X}) - \widetilde{\mathbf{M}}\mathbf{X}\right)$ since there is no randomness in $\widetilde{\mathcal{A}}, \mathbf{X}$ in fix design setting. Assuming that $\widetilde{\mathcal{A}}(\mathbf{X})$ admits full column rank, we have the following theorem on the excess risk of DAC and ERM:

**Theorem 6** (Formal restatement of Theorem 2 on linear regression.). *Learning with DAC regularization, we have* $\mathbb{E}\left[L(\widehat{\theta}^{dac}) - L(\theta^*)\right] = \frac{(d-d_{aug})\sigma^2}{N}$, *while learning with ERM directly on the augmented dataset, we have* $\mathbb{E}\left[L(\widehat{\theta}^{da-erm}) - L(\theta^*)\right] = \frac{(d-d_{aug}+d')\sigma^2}{N}$. $d'$ *is defined as*

$$d' \triangleq \frac{\mathrm{tr}\left(\widetilde{\mathbf{M}}^\top \left(H_{\widetilde{\mathcal{A}}(\mathbf{X})} - \mathbf{P}_{\mathcal{S}}\right)\widetilde{\mathbf{M}}\right)}{1+\alpha},$$

*where* $d' \in [0, d_{aug}]$ *with* $H_{\widetilde{\mathcal{A}}(\mathbf{X})} = \widetilde{\mathcal{A}}(\mathbf{X})\left(\widetilde{\mathcal{A}}(\mathbf{X})^\top \widetilde{\mathcal{A}}(\mathbf{X})\right)^{-1}\widetilde{\mathcal{A}}(\mathbf{X})^\top$ *and* $\mathbf{P}_{\mathcal{S}} \in \mathbb{R}^{(\alpha+1)N \times (\alpha+1)N}$ *is the projection matrix to* $\mathcal{S}$, *and* $\mathcal{S}$ *is defined to be* $\mathcal{S} \triangleq \left\{\widetilde{\mathbf{M}}\mathbf{X}\theta \mid \forall \theta \in \mathbb{R}^d, s.t. \left(\widetilde{\mathcal{A}}(\mathbf{X}) - \widetilde{\mathbf{M}}\mathbf{X}\right)\theta = 0\right\}$.

*Proof.* With $L(\theta) \triangleq \frac{1}{(1+\alpha)N}\left\|\widetilde{\mathcal{A}}(\mathbf{X})\theta - \widetilde{\mathcal{A}}(\mathbf{X})\theta^*\right\|_2^2$, the excess risk of ERM on the augmented training set satisfies that:

$$
\begin{aligned}
\mathbb{E}\left[L(\widehat{\theta}^{da-erm})\right] &= \frac{1}{(1+\alpha)N}\mathbb{E}\left[\left\|\widetilde{\mathcal{A}}(\mathbf{X})\widehat{\theta}^{da-erm} - \widetilde{\mathcal{A}}(\mathbf{X})\theta^*\right\|_2^2\right] \\
&= \frac{1}{(1+\alpha)N}\mathbb{E}\left[\left\|\widetilde{\mathcal{A}}(\mathbf{X})(\widetilde{\mathcal{A}}(\mathbf{X})^\top\widetilde{\mathcal{A}}(\mathbf{X}))^{-1}\widetilde{\mathcal{A}}(\mathbf{X})^\top(\widetilde{\mathcal{A}}(\mathbf{X})\theta^* + \widetilde{\mathbf{M}}\epsilon) - \widetilde{\mathcal{A}}(\mathbf{X})\theta^*\right\|_2^2\right] \\
&= \frac{1}{(1+\alpha)N}\mathbb{E}\left[\left\|H_{\widetilde{\mathcal{A}}(\mathbf{X})}\widetilde{\mathcal{A}}(\mathbf{X})\theta^* + H_{\widetilde{\mathcal{A}}(\mathbf{X})}\widetilde{\mathbf{M}}\epsilon - \widetilde{\mathcal{A}}(\mathbf{X})\theta^*\right\|_2^2\right] \\
&= \frac{1}{(1+\alpha)N}\mathbb{E}\left[\left\|H_{\widetilde{\mathcal{A}}(\mathbf{X})}\widetilde{\mathbf{M}}\epsilon\right\|_2^2\right] \\
&= \frac{1}{(1+\alpha)N}\mathbb{E}\left[\mathrm{tr}(\epsilon^\top\widetilde{\mathbf{M}}^\top H_{\widetilde{\mathcal{A}}(\mathbf{X})}\widetilde{\mathbf{M}}\epsilon)\right] \\
&= \frac{\sigma^2}{(1+\alpha)N}\mathrm{tr}\left(\widetilde{\mathbf{M}}^\top H_{\widetilde{\mathcal{A}}(\mathbf{X})}\widetilde{\mathbf{M}}\right).
\end{aligned}
$$

Let $\mathcal{C}_{\widetilde{\mathcal{A}}(\mathbf{X})}$ and $\mathcal{C}_{\widetilde{\mathbf{M}}}$ denote the column space of $\widetilde{\mathcal{A}}(\mathbf{X})$ and $\widetilde{\mathbf{M}}$, respectively. Notice that $\mathcal{S}$ is a subspace of both $\mathcal{C}_{\widetilde{\mathcal{A}}(\mathbf{X})}$ and $\mathcal{C}_{\widetilde{\mathbf{M}}}$. Observing that $d_{aug} = \mathrm{rank}\left(\widetilde{\mathcal{A}}(\mathbf{X}) - \widetilde{\mathbf{M}}\mathbf{X}\right) = \mathrm{rank}\left(\mathbf{P}_{\mathcal{S}}\right)$, we have

$$
\begin{aligned}
\mathbb{E}\left[L(\widehat{\theta}^{da-erm})\right] &= \frac{\sigma^2}{(1+\alpha)N}\mathbb{E}\left[\mathrm{tr}(\widetilde{\mathbf{M}}^\top H_{\widetilde{\mathcal{A}}(\mathbf{X})}\widetilde{\mathbf{M}})\right] \\
&= \frac{\sigma^2}{(1+\alpha)N}\mathbb{E}\left[\mathrm{tr}(\widetilde{\mathbf{M}}^\top\mathbf{P}_{\mathcal{S}}\widetilde{\mathbf{M}})\right] + \frac{\sigma^2}{(1+\alpha)N}\mathbb{E}\left[\mathrm{tr}(\widetilde{\mathbf{M}}^\top(H_{\widetilde{\mathcal{A}}(\mathbf{X})} - \mathbf{P}_{\mathcal{S}})\widetilde{\mathbf{M}})\right] \\
&= \frac{\sigma^2}{(1+\alpha)N}\mathbb{E}\left[\mathrm{tr}(\widetilde{\mathbf{M}}^\top\mathbf{P}_{\mathcal{S}}\widetilde{\mathbf{M}})\right] + \frac{\sigma^2}{N}\cdot\mathbb{E}\left[\frac{\mathrm{tr}(\widetilde{\mathbf{M}}^\top(H_{\widetilde{\mathcal{A}}(\mathbf{X})} - \mathbf{P}_{\mathcal{S}})\widetilde{\mathbf{M}})}{1+\alpha}\right]
\end{aligned}
$$

By the data augmentation consistency constraint, we are essentially solving the linear regression on the $(d - d_{aug})$-dimensional space $\left\{\theta \mid (\widetilde{\mathcal{A}}(\mathbf{X}) - \widetilde{\mathbf{M}}\mathbf{X})\theta = 0\right\}$. The rest of proof is identical to standard regression analysis, with features first projected to $\mathcal{S}$:

$$
\begin{aligned}
\mathbb{E}\left[L(\widehat{\theta}^{dac})\right] &= \frac{1}{(1+\alpha)N}\mathbb{E}\left[\left\|\widetilde{\mathcal{A}}(\mathbf{X})\widehat{\theta}^{dac} - \widetilde{\mathcal{A}}(\mathbf{X})\theta^*\right\|_2^2\right] \\
&= \frac{1}{(1+\alpha)N}\mathbb{E}\left[\left\|\widetilde{\mathcal{A}}(\mathbf{X})(\widetilde{\mathcal{A}}(\mathbf{X})^\top\widetilde{\mathcal{A}}(\mathbf{X}))^{-1}\widetilde{\mathcal{A}}(\mathbf{X})^\top\mathbf{P}_{\mathcal{S}}(\widetilde{\mathcal{A}}(\mathbf{X})\theta^* + \widetilde{\mathbf{M}}\epsilon) - \widetilde{\mathcal{A}}(\mathbf{X})\theta^*\right\|_2^2\right] \\
&= \frac{1}{(1+\alpha)N}\mathbb{E}\left[\left\|H_{\widetilde{\mathcal{A}}(\mathbf{X})}\mathbf{P}_{\mathcal{S}}\widetilde{\mathcal{A}}(\mathbf{X})\theta^* + H_{\widetilde{\mathcal{A}}(\mathbf{X})}\mathbf{P}_{\mathcal{S}}\widetilde{\mathbf{M}}\epsilon - \widetilde{\mathcal{A}}(\mathbf{X})\theta^*\right\|_2^2\right] \\
&\quad \left(\because \widetilde{\mathcal{A}}(\mathbf{X})\theta^* \in \mathcal{S}, \text{ and } H_{\widetilde{\mathcal{A}}(\mathbf{X})}\mathbf{P}_{\mathcal{S}} = \mathbf{P}_{\mathcal{S}} \text{ since } \mathcal{S} \subseteq \mathcal{C}_{\widetilde{\mathcal{A}}(\mathbf{X})}\right) \\
&= \frac{1}{(1+\alpha)N}\mathbb{E}\left[\left\|\mathbf{P}_{\mathcal{S}}\widetilde{\mathbf{M}}\epsilon\right\|_2^2\right] \\
&= \frac{\sigma^2}{(1+\alpha)N}\mathbb{E}\left[\mathrm{tr}(\widetilde{\mathbf{M}}^\top\mathbf{P}_{\mathcal{S}}\widetilde{\mathbf{M}})\right] \\
&= \frac{(d - d_{aug})\sigma^2}{N}.
\end{aligned}
$$

∎

## B  APPLICATIONS OF LEARNING WITH DAC

To instantiate our general result on the DAC regularization, we start with a concrete application of Proposition 8 on the logistic regression problem with bounded loss (Section 4.1, Appendix B.1), followed by an extension to the unbounded square loss (Section 4.2, Appendix B.2). Subsequently, we discuss an alternative notion of data augmentations based on expansion for the classification problems (Section 4.3, Appendix B.3). Finally, we present a supplementary example for the domain adaptation in linear regression (Appendix B.4).

Before diving into the concrete applications, we recall the general setting, and introduce some additional notations used throughout the formal analysis.

**DAC regularization with unlabeled data.**  Since the DAC regularization leverages only on the unlabeled observable features but not their labels, the unlabeled training set in the DAC regularizer and the labeled training set in the first term of Equation (1) for label learning can be considered separately. For clarification, in the formal analysis, we distinguish the labeled training set $(\mathbf{X}, \mathbf{y}) \in \mathcal{X}^n \times \mathcal{Y}^n$ from the unlabeled samples $\mathbf{X}^u \in \mathcal{X}^N$ (possibly with different sizes $N \geq n$), while both sample sets are drawn *i.i.d.* from $P^*$. Particularly, in the **supervised learning setting**, the DAC regularization is based on the observables from the labeled training set such that $\mathbf{X}^u = \mathbf{X}$ ($N = n$). While in the **semi-supervised learning setting**, $\mathbf{X}^u$ can be $N$ unlabeled observations drawn *i.i.d.* from the marginal distribution $P^*(\mathbf{x})$.

For the DAC regularization, we denote the augmentation of the unlabeled samples (excluding the original samples $\mathbf{X}^u$, in contrast to the augmented sample $\widetilde{\mathcal{A}}(\mathbf{X})$ in the main text) as

$$
\widehat{\mathcal{A}}(\mathbf{X}^u) = \left[\mathbf{x}_{1,1}^u; \cdots; \mathbf{x}_{N,1}^u; \cdots; \mathbf{x}_{1,\alpha}^u; \cdots; \mathbf{x}_{N,\alpha}^u\right] \in \mathcal{X}^{\alpha N},
$$

where for each $i \in [N]$, $\left\{\mathbf{x}_{i,j}^u\right\}_{j\in[\alpha]}$ is a set of $\alpha$ augmentations generated from $\mathbf{x}_i^u$, and let $\mathbf{M} \in \mathbb{R}^{\alpha N \times N}$ be the vertical stack of $\alpha$ $N \times N$ identity matrices. Then analogously, with the unlabeled training set $\mathbf{X}^u$, we can quantify the strength of the data augmentations $\widehat{\mathcal{A}}$ with

$$
\widehat{d}_{aug} \triangleq \mathrm{rank}\left(\widehat{\mathcal{A}}(\mathbf{X}^u) - \mathbf{M}\mathbf{X}^u\right) = \mathrm{rank}\left(\widetilde{\mathcal{A}}(\mathbf{X}^u) - \widetilde{\mathbf{M}}\mathbf{X}^u\right),
$$

such that $0 \leq \widehat{d}_{aug} \leq \min(d, \alpha N)$ can be intuitively interpreted as the number of dimensions in the span of the unlabeled samples, $\mathrm{Row}(\mathbf{X}^u)$, perturbed by $\widehat{\mathcal{A}}$. Moreover, we denote $\mathcal{N}$ as the $(d - \widehat{d}_{aug})$-dimensional null space of $\widehat{\mathcal{A}}(\mathbf{X}^u) - \mathbf{M}\mathbf{X}^u$, and $\mathbf{P}_{\mathcal{N}}$ as the projection onto $\mathcal{N}$ such that

$$
\mathbf{P}_{\mathcal{N}} \triangleq \mathbf{I}_d - \left(\widehat{\mathcal{A}}(\mathbf{X}^u) - \mathbf{M}\mathbf{X}^u\right)^\dagger \left(\widehat{\mathcal{A}}(\mathbf{X}^u) - \mathbf{M}\mathbf{X}^u\right).
$$

Correspondingly, let $\mathcal{N}^\perp$ be the orthogonal complement of $\mathcal{N}$, and $\mathbf{P}_{\mathcal{N}}^\perp$ be the projection onto $\mathcal{N}^\perp$ such that $\mathbf{P}_{\mathcal{N}}^\perp = \mathbf{I}_d - \mathbf{P}_{\mathcal{N}}$. Then by recalling Definition 2, we have the following:

**Proposition 7.** *For any $\delta \in (0, 1)$, with probability at least $1 - \delta$,*

$$\text{rank}\left(\mathbf{P}_{\mathcal{N}}^\perp\right) = \dim\left(\mathcal{N}^\perp\right) = \widehat{d}_{aug} \geq d_{aug}(\delta),$$

$$\text{rank}\left(\mathbf{P}_{\mathcal{N}}\right) = \dim\left(\mathcal{N}\right) = d - \widehat{d}_{aug} \leq d - d_{aug}(\delta).$$

*Furthermore, for $d_{aug} \triangleq d_{aug}\left(1/n\right)$, with high probability, $\widehat{d}_{aug} \geq d_{aug}$ and $d - \widehat{d}_{aug} \leq d - d_{aug}$.*

In terms of $(\mathbf{X}, \mathbf{y})$ and $\mathbf{X}^u$, the regularization formulation in Equation (1) can be restated as,

$$\underset{h \in \mathcal{H}}{\text{argmin}} \sum_{i=1}^{n} l(h(\mathbf{x}_i), y_i) + \lambda \underbrace{\sum_{j=1}^{\alpha} \sum_{i=1}^{N} \varrho\left(\phi_h(\mathbf{x}_i^u), \phi_h(\mathbf{x}_{i,j}^u)\right)}_{DAC\ regularizer},$$

and with the notation of $\widehat{\mathcal{A}}$, the corresponding DAC operator is given by

$$T_{\widehat{\mathcal{A}}, \mathbf{X}^u}^{dac}(\mathcal{H}) \triangleq \left\{ h \in \mathcal{H} \mid \phi_h(\mathbf{x}_i^u) = \phi_h(\mathbf{x}_{i,j}^u), \ \forall i \in [N], j \in [\alpha] \right\},$$

where $T_{\widehat{\mathcal{A}}, \mathbf{X}^u}^{dac}$ and $T_{\widehat{\mathcal{A}}, \mathbf{X}^u}^{dac}$ are equivalent by construction.

**Marginal distribution and training set.** Here we formally state the common regularity conditions for the logistic regression (Section 4.1) and the two-layer neural network (Section 4.2) examples in Assumption 1 and Assumption 2, where we assume $\mathcal{X} \subseteq \mathbb{R}^d$.

**Assumption 1** (Regularity of marginal distribution). *Let $\mathbf{x} \sim P^*(\mathbf{x})$ be zero-mean $\mathbb{E}[\mathbf{x}] = \mathbf{0}$, with the covairance $\mathbb{E}[\mathbf{x}\mathbf{x}^\top] = \mathbf{\Sigma_x} \succ 0$. We assume that $(\mathbf{\Sigma_x}^{-1/2}\mathbf{x})$ is $\rho^2$-subgaussian [1], and there exist constants $C \geq c = \Theta(1)$ such that $c\mathbf{I}_d \preccurlyeq \mathbf{\Sigma_x} \preccurlyeq C\mathbf{I}_d$.*

**Assumption 2** (Sufficient labeled data). *For $(\mathbf{X}, \mathbf{y}) \in \mathcal{X}^n \times \mathcal{Y}^n$, with respect to $\mathbf{X}^u$ and $\widehat{\mathcal{A}}\left(\mathbf{X}^u\right)$ that characterize $\widehat{d}_{aug}$, we assume $n \gg \rho^4\left(d - \widehat{d}_{aug}\right)$.*

For a $B$-bounded loss function $l : \mathcal{Y} \times \mathcal{Y} \to \mathbb{R}$ (*i.e.*, $0 \leq l \leq B$) such that for all $y \in \mathcal{Y}$, $l(\cdot, y) : \mathcal{Y} \to \mathbb{R}$ is $C_l$-Lipschitz, we have the following:

**Proposition 8** (Formal restatement of Proposition 1 on DAC with bounded loss). *Learning with DAC regularization (Equation (2)), for a fixed $T_{\widehat{\mathcal{A}}, \mathbf{X}^u}^{dac}(\mathcal{H})$ and any $\delta \in (0, 1)$, with probability at least $1 - \delta$, we have*

$$L(\widehat{h}^{dac}) - L(h^*) \leq 4C_l \cdot \mathfrak{R}_n(T_{\widehat{\mathcal{A}}, \mathbf{X}^u}^{dac}(\mathcal{H})) + \sqrt{\frac{2B^2 \log(2/\delta)}{n}},$$

*where $\mathfrak{R}_n(T_{\widehat{\mathcal{A}}, \mathbf{X}^u}^{dac}(\mathcal{H}))$ represents the Rademacher complexity of $T_{\widehat{\mathcal{A}}, \mathbf{X}^u}^{dac}(\mathcal{H})$.*

*Proof of Proposition 8.* We first decompose the expected excess risk as

$$L(\widehat{h}^{dac}) - L(h^*) = \left(L(\widehat{h}^{dac}) - \widehat{L}(\widehat{h}^{dac})\right) + \left(\widehat{L}(\widehat{h}^{dac}) - \widehat{L}(h^*)\right) + \left(\widehat{L}(h^*) - L(h^*)\right),$$

where $\widehat{L}(\widehat{h}^{dac}) - \widehat{L}(h^*) \leq 0$ by the basic inequality, and as a consequence, for a fixed $\mathcal{H}_o = T_{\widehat{\mathcal{A}}, \mathbf{X}^u}^{dac}(\mathcal{H})$,

$$L(\widehat{h}^{dac}) - L(h^*) \leq 2 \sup_{h \in T_{\widehat{\mathcal{A}}, \mathbf{X}^u}^{dac}(\mathcal{H})} \left| L(h) - \widehat{L}(h) \right| = 2 \sup_{h \in \mathcal{H}_o} \left| L(h) - \widehat{L}(h) \right|.$$

We denote $g^+(\mathbf{X}, \mathbf{y}) = \sup_{h \in \mathcal{H}_o} : L(h) - \widehat{L}(h)$ and $g^-(\mathbf{X}, \mathbf{y}) = \sup_{h \in \mathcal{H}_o} : -L(h) + \widehat{L}(h)$. Then,

$$\mathbb{P}\left[L(\widehat{h}^{dac}) - L(h^*) \geq \epsilon\right] \leq \mathbb{P}\left[g^+(\mathbf{X}, \mathbf{y}) \geq \frac{\epsilon}{2}\right] + \mathbb{P}\left[g^-(\mathbf{X}, \mathbf{y}) \geq \frac{\epsilon}{2}\right].$$

---

[1] A random vector $\mathbf{v} \in \mathbb{R}^d$ is $\rho^2$-subgaussian if for any unit vector $\mathbf{u} \in \mathbb{S}^{d-1}$, $\mathbf{u}^\top \mathbf{v}$ is $\rho^2$-subgaussian, $\mathbb{E}\left[\exp(s \cdot \mathbf{u}^\top \mathbf{v})\right] \leq \exp\left(s^2 \rho^2/2\right)$.

We will derive a tail bound for $g^+(\mathbf{X}, \mathbf{y})$ with the standard inequalities and symmetrization argument Wainwright (2019); Bartlett & Mendelson (2003), while the analogous statement holds for $g^-(\mathbf{X}, \mathbf{y})$.

Let $(\mathbf{X}^{(1)}, \mathbf{y}^{(1)})$ be a sample set generated by replacing an arbitrary observation in $(\mathbf{X}, \mathbf{y})$ with any $(\mathbf{x}, y) \in \mathcal{X} \times \mathcal{Y}$. Since $l$ is $B$-bounded, we have $\left| g^+(\mathbf{X}, \mathbf{y}) - g^+(\mathbf{X}^{(1)}, \mathbf{y}^{(1)}) \right| \le B/n$. Leveraging the McDiarmid's inequality Bartlett & Mendelson (2003),

$$\mathbb{P}\left[ g^+(\mathbf{X}, \mathbf{y}) \ge \mathbb{E}[g^+(\mathbf{X}, \mathbf{y})] + t \right] \le \exp\left( -\frac{2nt^2}{B^2} \right).$$

For an arbitrary sample set $(\mathbf{X}, \mathbf{y})$, let $\widehat{L}_{(\mathbf{X}, \mathbf{y})}(h) = \frac{1}{n} \sum_{i=1}^{n} l\left( h(\mathbf{x}_i), y_i \right)$ be the empirical risk of $h$ with respect to $(\mathbf{X}, \mathbf{y})$. Then, by a classical symmetrization argument (e.g., proof of Wainwright (2019) Theorem 4.10), we can bound the expectation: for an independent sample set $(\mathbf{X}', \mathbf{y}') \in \mathcal{X}^n \times \mathcal{Y}^n$ drawn *i.i.d.* from $P^*$,

$$
\begin{aligned}
\mathbb{E}\left[ g^+(\mathbf{X}, \mathbf{y}) \right] &= \mathbb{E}_{(\mathbf{X}, \mathbf{y})} \left[ \sup_{h \in \mathcal{H}_o} L(h) - \widehat{L}_{(\mathbf{X}, \mathbf{y})}(h) \right] \\
&= \mathbb{E}_{(\mathbf{X}, \mathbf{y})} \left[ \sup_{h \in \mathcal{H}_o} \mathbb{E}_{(\mathbf{X}', \mathbf{y}')} \left[ \widehat{L}_{(\mathbf{X}', \mathbf{y}')}(h) \right] - \widehat{L}_{(\mathbf{X}, \mathbf{y})}(h) \right] \\
&= \mathbb{E}_{(\mathbf{X}, \mathbf{y})} \left[ \sup_{h \in \mathcal{H}_o} \mathbb{E}_{(\mathbf{X}', \mathbf{y}')} \left[ \widehat{L}_{(\mathbf{X}', \mathbf{y}')}(h) - \widehat{L}_{(\mathbf{X}, \mathbf{y})}(h) \,\Big|\, (\mathbf{X}, \mathbf{y}) \right] \right] \\
&\le \mathbb{E}_{(\mathbf{X}, \mathbf{y})} \left[ \mathbb{E}_{(\mathbf{X}', \mathbf{y}')} \left[ \sup_{h \in \mathcal{H}_o} \widehat{L}_{(\mathbf{X}', \mathbf{y}')}(h) - \widehat{L}_{(\mathbf{X}, \mathbf{y})}(h) \,\Big|\, (\mathbf{X}, \mathbf{y}) \right] \right] \\
&\quad \text{(Law of iterated conditional expectation)} \\
&= \mathbb{E}_{(\mathbf{X}, \mathbf{y}, \mathbf{X}', \mathbf{y}')} \left[ \sup_{h \in \mathcal{H}_o} \widehat{L}_{(\mathbf{X}', \mathbf{y}')}(h) - \widehat{L}_{(\mathbf{X}, \mathbf{y})}(h) \right]
\end{aligned}
$$

Since $(\mathbf{X}, \mathbf{y})$ and $(\mathbf{X}', \mathbf{y}')$ are independent and identically distributed, we can introduce *i.i.d.* Rademacher random variables $\mathbf{r} = \{ r_i \in \{-1, +1\} \mid i \in [n] \}$ (independent of both $(\mathbf{X}, \mathbf{y})$ and $(\mathbf{X}', \mathbf{y}')$) such that

$$
\begin{aligned}
\mathbb{E}\left[ g^+(\mathbf{X}, \mathbf{y}) \right] &\le \mathbb{E}_{(\mathbf{X}, \mathbf{y}, \mathbf{X}', \mathbf{y}', \mathbf{r})} \left[ \sup_{h \in \mathcal{H}_o} \frac{1}{n} \sum_{i=1}^{n} r_i \cdot \left( l\left( h(\mathbf{x}_i'), y_i' \right) - l\left( h(\mathbf{x}_i), y_i \right) \right) \right] \\
&\le 2 \, E_{(\mathbf{X}, \mathbf{y}, \mathbf{r})} \left[ \sup_{h \in \mathcal{H}_o} \frac{1}{n} \sum_{i=1}^{n} r_i \cdot l\left( h(\mathbf{x}_i), y_i \right) \right] \\
&\le 2 \, \mathfrak{R}_n(l \circ \mathcal{H}_o)
\end{aligned}
$$

where $l \circ \mathcal{H}_o = \{ l(h(\cdot), \cdot) : \mathcal{X} \times \mathcal{Y} \to \mathbb{R} : h \in \mathcal{H}_o \}$ is the loss function class. Analogously, $\mathbb{E}[g^-(\mathbf{X}, \mathbf{y})] \le 2\mathfrak{R}_n(l \circ \mathcal{H}_o)$. Therefore, with probability at least $1 - \delta$,

$$L(\widehat{h}^{dac}) - L(h^*) \le 4\mathfrak{R}_n(l \circ \mathcal{H}_o) + \sqrt{\frac{2B^2 \log(2/\delta)}{n}}$$

Finally, since $l(\cdot, y)$ is $C_l$-Lipschitz for all $y \in \mathcal{Y}$, by Ledoux & Talagrand (2013) Theorem 4.12, we have $\mathfrak{R}_n(l \circ \mathcal{H}_o) \le C_l \cdot \mathfrak{R}_n(\mathcal{H}_o)$. ∎

### B.1 FORMAL RESULT ON LOGISTIC REGRESSION

For the logistic regression problem in Section 4.1, with $\mathcal{X} = \left\{ \mathbf{x} \in \mathbb{R}^d \mid \|\mathbf{x}\|_2 \le D \right\}$, $\mathcal{Y} = \{0, 1\}$, training with the DAC regularization can be formulated explicitly as

$$\widehat{\boldsymbol{\theta}}^{dac} = \operatorname*{argmin}_{\boldsymbol{\theta} \in \mathbb{R}^d} \frac{1}{n} \sum_{i=1}^{n} l\left( \boldsymbol{\theta}^\top \mathbf{x}_i, y_i \right)$$

$$\text{s.t.} \quad \|\boldsymbol{\theta}\|_2 \le C_0, \quad \widehat{\mathcal{A}}(\mathbf{X}^u)\boldsymbol{\theta} = \mathbf{M}\mathbf{X}^u\boldsymbol{\theta},$$

and yields $\widehat{h}^{dac}(\mathbf{x}) \triangleq \mathbf{x}^\top \widehat{\boldsymbol{\theta}}^{dac}$.

**Theorem 9** (Formal restatement of Theorem 3 on logistic regression with DAC). *Let* $\mathbb{E}\left[\frac{1}{n}\sum_{i=1}^{n}\|\mathbf{P}_{\mathcal{N}}\mathbf{x}_i\|_2^2\Big|\mathbf{P}_{\mathcal{N}}\right] \leq C_{\mathcal{N}}^2$ *for some* $C_{\mathcal{N}} > 0$. *Then, learning logistic regression with the DAC regularization* $h(\mathbf{x}_i^u) = h(\mathbf{x}_{i,j}^u)$ *yields that, for any* $\delta \in (0,1)$, *with probability at least* $1 - \delta$:

$$L(\widehat{h}^{dac}) - L(h^*) \lesssim \frac{C_0 C_{\mathcal{N}} + C_0 D\sqrt{\log\left(1/\delta\right)}}{\sqrt{n}},$$

*where under Assumption 1 and Assumption 2,* $C_{\mathcal{N}} \lesssim \sqrt{d - d_{aug}\left(\delta/2\right)}$.

*Proof of Theorem 9.* Notice that the logistic loss $l(\cdot, y)$ is 2-Lipschitz for all $y \in \mathcal{Y}$, as $\frac{\partial l(z,y)}{\partial z} = \sigma(z) - y$, whose absolute value is bounded by 2. Meanwhile, with $\|\boldsymbol{\theta}\|_2 \leq C_0$ and $\|\mathbf{x}\|_2 \leq D$ for all $\boldsymbol{\theta}$ and $\mathbf{x}$, the logistic loss is also bounded: $l\left(\boldsymbol{\theta}^\top\mathbf{x}, y\right) \leq \log 2 + C_0 D$. Then, applying Proposition 8, we have that for any $\delta \in (0,1)$, with probability at least $1 - \delta/2$,

$$L(\widehat{h}^{dac}) - L(h^*) \lesssim \mathfrak{R}_n\left(T_{\widehat{\mathcal{A}},\mathbf{X}^u}^{dac}(\mathcal{H})\right) + C_0 D\sqrt{\frac{\log(1/\delta)}{n}}.$$

It left to bound the Rademacher complexity of $T_{\widehat{\mathcal{A}},\mathbf{X}^u}^{dac}(\mathcal{H})$. First notice that the sigmoid function $\sigma(\cdot)$ is a bijective function, the data augmentation consistency constraints are equivalent to

$$\left(\widehat{\mathcal{A}}(\mathbf{X}^u) - \mathbf{M}\mathbf{X}^u\right)\boldsymbol{\theta} = 0.$$

Thus, we have

$$T_{\widehat{\mathcal{A}},\mathbf{X}^u}^{dac}(\mathcal{H}) = \left\{h(\mathbf{x}) = \boldsymbol{\theta}^\top\mathbf{x} \mid \boldsymbol{\theta} \in \mathcal{N}, \|\boldsymbol{\theta}\|_2 \leq C_0\right\},$$

with which we are ready to bound the Rademacher complexity of $T_{\widehat{\mathcal{A}},\mathbf{X}^u}^{dac}(\mathcal{H})$.

For the empirical Rademacher complexity, we have

$$
\begin{aligned}
\widehat{\mathfrak{R}}_{\mathbf{X}}\left(T_{\widehat{\mathcal{A}},\mathbf{X}^u}^{dac}(\mathcal{H})\right) &= \frac{1}{n}\mathbb{E}_{\epsilon_i \sim Rad(\frac{1}{2})}\left[\sup_{\boldsymbol{\theta} \in T_{\widehat{\mathcal{A}},\mathbf{X}^u}^{dac}(\mathcal{H})}\sum_{i=1}^{n}\epsilon_i\boldsymbol{\theta}^\top\mathbf{x}_i\right] \\
&= \frac{1}{n}\mathbb{E}_{\epsilon_i \sim Rad(\frac{1}{2})}\left[\sup_{\boldsymbol{\theta} \in \mathcal{H}}\sum_{i=1}^{n}\epsilon_i\boldsymbol{\theta}^\top\mathbf{P}_{\mathcal{N}}\mathbf{x}_i\right] \\
&= \frac{C_0}{n}\mathbb{E}_{\epsilon_i \sim Rad(\frac{1}{2})}\left[\left\|\sum_{i=1}^{n}\epsilon_i\mathbf{P}_{\mathcal{N}}\mathbf{x}_i\right\|_2\right] \\
&\leq \frac{C_0}{n}\sqrt{\mathbb{E}_{\epsilon_i \sim Rad(\frac{1}{2})}\sum_{i=1}^{n}\|\epsilon_i\mathbf{P}_{\mathcal{N}}\mathbf{x}_i\|_2^2} \\
&= \frac{C_0}{n}\sqrt{\sum_{i=1}^{n}\|\mathbf{P}_{\mathcal{N}}\mathbf{x}_i\|_2^2} \\
&= \frac{C_0}{\sqrt{n}}\sqrt{\text{tr}\left(\frac{1}{n}\mathbf{P}_{\mathcal{N}}\mathbf{X}^\top\mathbf{X}\mathbf{P}_{\mathcal{N}}\right)}.
\end{aligned}
$$

Converting this to the population Rademacher complexity, we take expectation over $\mathbf{X}$, while conditioned on $\mathbf{P}_{\mathcal{N}}$, and recall Jensen's inequality,

$$
\begin{aligned}
\mathfrak{R}_n\left(T_{\widehat{\mathcal{A}},\mathbf{X}^u}^{dac}(\mathcal{H})\right) &\leq \frac{C_0}{\sqrt{n}}\mathbb{E}\left[\sqrt{\text{tr}\left(\frac{1}{n}\mathbf{P}_{\mathcal{N}}\mathbf{X}^\top\mathbf{X}\mathbf{P}_{\mathcal{N}}\right)}\ \Big|\ \mathbf{P}_{\mathcal{N}}\right] \\
&\leq \frac{C_0}{\sqrt{n}}\sqrt{\mathbb{E}\left[\text{tr}\left(\frac{1}{n}\mathbf{P}_{\mathcal{N}}\mathbf{X}^\top\mathbf{X}\mathbf{P}_{\mathcal{N}}\right)\ \Big|\ \mathbf{P}_{\mathcal{N}}\right]} \\
&= \frac{C_0 C_{\mathcal{N}}}{\sqrt{n}}.
\end{aligned}
$$

Leveraging Lemma 5, we have that under Assumption 1 and Assumption 2, conditioned on $\mathbf{P}_{\mathcal{N}}$, with high probability over $\mathbf{X} \sim P^*$,

$$\left\| \frac{1}{n} \mathbf{P}_{\mathcal{N}} \mathbf{X}^\top \mathbf{X} \mathbf{P}_{\mathcal{N}} \right\|_2 \leq 1.1 C \lesssim 1$$

Then we can find some $C_{\mathcal{N}} > 0$ with $\frac{1}{n} \sum_{i=1}^n \|\mathbf{P}_{\mathcal{N}} \mathbf{x}_i\|_2^2 \leq C_{\mathcal{N}}^2$ such that,

$$C_{\mathcal{N}}^2 \leq \left( d - \widehat{d}_{aug} \right) \cdot \left\| \frac{1}{n} \mathbf{P}_{\mathcal{N}} \mathbf{X}^\top \mathbf{X} \mathbf{P}_{\mathcal{N}} \right\|_2 \lesssim d - \widehat{d}_{aug}.$$

By Proposition 7, we have $C_{\mathcal{N}} \leq \sqrt{d - d_{aug}(\delta/2)}$ with probability at least $1 - \delta/2$.

To obtain Theorem 3, we leverage the union bound and suppress the constants such that, with probability at least $1 - \delta$,

$$L(\widehat{h}^{dac}) - L(h^*) \lesssim \sqrt{\frac{d - d_{aug}(\delta/2)}{n}} + \sqrt{\frac{\log(1/\delta)}{n}}. \tag{4}$$

Finally, taking $\delta = 2/n$ and applying the Jensen's inequality to the two terms in Equation (4) complete the proof. ∎

## B.2 FORMAL RESULT ON TWO-LAYER NEURAL NETWORK

In the two-layer neural network regression setting described in Section 4.2, training with the DAC regularization can be formulated explicitly as

$$\widehat{\mathbf{B}}^{dac}, \widehat{\mathbf{w}}^{dac} = \underset{\mathbf{B} \in \mathbb{R}^{d \times q}, \mathbf{w} \in \mathbb{R}^q}{\operatorname{argmin}} \frac{1}{2n} \left\| \mathbf{y} - (\mathbf{X}\mathbf{B})_+ \mathbf{w} \right\|_2^2$$

$$\text{s.t.} \quad \mathbf{B} = [\mathbf{b}_1 \ldots \mathbf{b}_k \ldots \mathbf{b}_q], \quad \mathbf{b}_k \in \mathbb{S}^{d-1} \ \forall \ k \in [q], \quad \|\mathbf{w}\|_1 \leq C_w$$

$$\left( \widehat{\mathcal{A}}(\mathbf{X}^u) \mathbf{B} \right)_+ = (\mathbf{M} \mathbf{X}^u \mathbf{B})_+,$$

and yields $\widehat{h}^{dac}(\mathbf{x}) \triangleq (\mathbf{x}^\top \widehat{\mathbf{B}}^{dac})_+ \widehat{\mathbf{w}}^{dac}$.

**Theorem 10** (Formal restatement of Theorem 4 on two-layer neural network with DAC). *Sampling from $P^*$ that follows Assumption 1, let the augmented training set $\widehat{\mathcal{A}}(\mathbf{X}^u)$ satisfy (a) $\alpha N \geq 4\widehat{d}_{aug}$, (b) $\widehat{\mathcal{A}}(\mathbf{X}^u) - \mathbf{M}\mathbf{X}^u$ admits an absolutely continuous distribution over $\mathcal{N}^\perp$, and $P^*$ fulfills (c) $\mathbb{E}\left[ \frac{1}{n} \sum_{i=1}^n \|\mathbf{P}_{\mathcal{N}} \mathbf{x}_i\|_2^2 \Big| \mathbf{P}_{\mathcal{N}} \right] \leq C_{\mathcal{N}}^2$ for some $C_{\mathcal{N}} > 0$. Then, learning the two-layer ReLU network with the DAC regularization on $\left( (\mathbf{x}_i^u)^\top \mathbf{B} \right)_+ = \left( (\mathbf{x}_{i,j}^u)^\top \mathbf{B} \right)_+$ gives that, with high probability,*

$$L\left( \widehat{h}^{dac} \right) - L(h^*) \lesssim \frac{\sigma C_w C_{\mathcal{N}}}{\sqrt{n}},$$

*where under Assumption 1 and Assumption 2, $C_{\mathcal{N}} \lesssim \sqrt{d - d_{aug}(\delta)}$ with probability at least $1 - \delta$.*

**Lemma 1.** *Under the assumptions in Theorem 10, every size-$\widehat{d}_{aug}$ subset of rows in $\widehat{\mathcal{A}}(\mathbf{X}^u) - \mathbf{M}\mathbf{X}^u$ is linearly independent almost surely.*

*Proof of Lemma 1.* Since $\alpha N > \widehat{d}_{aug}$, it is sufficient to show that a random matrix with an absolutely continuous distribution is totally invertible [2] almost surely.

It is known that for any dimension $m \in \mathbb{N}$, an $m \times m$ square matrix $\mathbf{S}$ is singular if $\det(\mathbf{S}) = 0$ where entries of $\mathbf{S}$ lie within the roots of the polynomial equation specified by the determinant. Therefore, the set of all singular matrices in $\mathbb{R}^{m \times m}$ has Lebesgue measure zero, $\lambda\left( \{ \mathbf{S} \in \mathbb{R}^{m \times m} \mid \det(\mathbf{S}) = 0 \} \right) = 0$. Then, for an absolutely continuous probability measure $\mu$ with respect to $\lambda$, we also have

$$\mathbb{P}_\mu \left[ \mathbf{S} \in \mathbb{R}^{m \times m} \text{ is singular} \right] = \mu\left( \{ \mathbf{S} \in \mathbb{R}^{m \times m} \mid \det(\mathbf{S}) = 0 \} \right) = 0.$$

Since a general matrix $\mathbf{R}$ contains only finite number of submatrices, when $\mathbf{R}$ is drawn from an absolutely continuous distribution, by the union bound, $\mathbb{P}[\mathbf{R}$ cotains a singular submatrix$] = 0$. That is, $\mathbf{R}$ is totally invertible almost surely. ∎

---

[2] A matrix is totally invertible if all its square submatrices are invertible.

**Lemma 2.** *Under the assumptions in Theorem 10, the hidden layer in the two-layer ReLU network learns $\mathcal{N}$, the invariant subspace under data augmentations : with high probability,*

$$\left(\mathbf{x}^\top \widehat{\mathbf{B}}^{dac}\right)_+ = \left(\mathbf{x}^\top \mathbf{P}_\mathcal{N} \widehat{\mathbf{B}}^{dac}\right)_+ \quad \forall\, \mathbf{x} \in \mathcal{X}.$$

*Proof of Lemma 2.* We will show that for all $\mathbf{b}_k = \mathbf{P}_\mathcal{N}\mathbf{b}_k + \mathbf{P}_\mathcal{N}^\perp\mathbf{b}_k$, $k \in [q]$, $\mathbf{P}_\mathcal{N}^\perp\mathbf{b}_k = \mathbf{0}$ with high probability, which then implies that given any $\mathbf{x} \in \mathcal{X}$, $(\mathbf{x}^\top\mathbf{b}_k)_+ = (\mathbf{x}^\top\mathbf{P}_\mathcal{N}\mathbf{b}_k)_+$ for all $k \in [q]$.

For any $k \in [q]$ associated with an arbitrary fixed $\mathbf{b}_k \in \mathbb{S}^{d-1}$, let $\mathbf{X}_k^u \triangleq \mathbf{X}_k^u \mathbf{P}_\mathcal{N} + \mathbf{X}_k^u \mathbf{P}_\mathcal{N}^\perp \in \mathcal{X}^{N_k}$ be the inclusion-wisely maximum row subset of $\mathbf{X}^u$ such that $\mathbf{X}_k^u \mathbf{b}_k > \mathbf{0}$ element-wisely. Meanwhile, we denote $\widehat{\mathcal{A}}(\mathbf{X}_k^u) = \mathbf{M}_k \mathbf{X}_k^u \mathbf{P}_\mathcal{N} + \widehat{\mathcal{A}}(\mathbf{X}_k^u)\mathbf{P}_\mathcal{N}^\perp \in \mathcal{X}^{\alpha N_k}$ as the augmentation of $\mathbf{X}_k^u$ where $\mathbf{M}_k \in \mathbb{R}^{\alpha N_k \times N_k}$ is the vertical stack of $\alpha$ identity matrices with size $N_k \times N_k$. Then the DAC constraint implies that $(\widehat{\mathcal{A}}(\mathbf{X}_k^u) - \mathbf{M}_k \mathbf{X}_k^u)\mathbf{P}_\mathcal{N}^\perp\mathbf{b}_k = \mathbf{0}$.

With Assumption 1, for a fixed $\mathbf{b}_k \in \mathbb{S}^{d-1}$, $\mathbb{P}[\mathbf{x}^\top\mathbf{b}_k > 0] = \frac{1}{2}$. Then, with the Chernoff bound,

$$\mathbb{P}\left[N_k < \frac{N}{2} - t\right] \le e^{-\frac{2t^2}{N}},$$

which implies that, $N_k \ge \frac{N}{4}$ with high probability.

Leveraging the assumptions in Theorem 10, $\alpha N \ge 4\widehat{d}_{aug}$ implies that $\alpha N_k \ge \widehat{d}_{aug}$. Therefore by Lemma 1, $\text{Row}\left(\widehat{\mathcal{A}}(\mathbf{X}_k^u) - \mathbf{M}_k \mathbf{X}_k^u\right) = \mathcal{N}^\perp$ with probability 1. Thus, $(\widehat{\mathcal{A}}(\mathbf{X}_k^u) - \mathbf{M}_k \mathbf{X}_k^u)\mathbf{P}_\mathcal{N}^\perp\mathbf{b}_k = \mathbf{0}$ enforces that $\mathbf{P}_\mathcal{N}^\perp\mathbf{b}_k = \mathbf{0}$. ∎

*Proof of Theorem 10.* Lemma 2 implies that $(\mathbf{X}\widehat{\mathbf{B}}^{dac})_+ = (\mathbf{X}\mathbf{P}_\mathcal{N}\widehat{\mathbf{B}}^{dac})_+$, and therefore,

$$T_{\widehat{\mathcal{A}},\mathbf{X}^u}^{dac}(\mathcal{H}) = \left\{h(\mathbf{x}) = (\mathbf{x}^\top\mathbf{B})_+\mathbf{w} \mid \mathbf{B} = [\mathbf{b}_1\dots\mathbf{b}_q], \mathbf{b}_k \in \mathbb{S}^{d-1}, (\mathbf{XB})_+ = (\mathbf{XP}_\mathcal{N}\mathbf{B})_+, \|\mathbf{w}\|_1 \le C_w\right\}.$$

In other words, the DAC regularization reduces the feasible set for $\mathbf{B}$ such that

$$\mathbf{B} \in \mathcal{B} \triangleq \{\mathbf{B} = [\mathbf{b}_1\dots\mathbf{b}_q] \mid \|\mathbf{b}_k\| = 1 \,\forall\, k \in [q], (\mathbf{XB})_+ = (\mathbf{XP}_\mathcal{N}\mathbf{B})_+\}, \quad \|\mathbf{w}\|_1 \le C_w.$$

Meanwhile, for the square loss, the corresponding excess risk is given by

$$L\left(\widehat{h}^{dac}\right) - L\left(h^*\right) = \mathbb{E}_\mathbf{X}\left[\frac{1}{2n}\left\|(\mathbf{X}\widehat{\mathbf{B}}^{dac})_+\widehat{\mathbf{w}}^{dac} - (\mathbf{XB}^*)_+\mathbf{w}^*\right\|_2^2\right].$$

Leveraging Equation (21) and (22) in Du et al. (2020), since $(\mathbf{B}^*, \mathbf{w}^*)$ is feasible under the constraint, by the basic inequality,

$$\left\|\mathbf{y} - (\mathbf{X}\widehat{\mathbf{B}}^{dac})_+\widehat{\mathbf{w}}^{dac}\right\|_2^2 \le \|\mathbf{y} - (\mathbf{XB}^*)_+\mathbf{w}^*\|_2^2, \tag{5}$$

where $\widehat{\mathbf{B}}^{dac}, \mathbf{B}^* \in \mathcal{B}$ and $\left\|\widehat{\mathbf{w}}^{dac}\right\|_1 \le C_w$.

Analogously to the Rademacher complexities for the bounded losses, for the square loss, we recall the definitions of the Gaussian complexities of a vector-valued function class $\Phi = \{\phi : \mathcal{X} \to \mathbb{R}^q\}$,

$$\widehat{\mathfrak{G}}_\mathbf{X}(\Phi) \triangleq \mathop{\mathbb{E}}_{g_{ik}\sim\mathcal{N}(0,1)\ i.i.d.}\left[\sup_{\phi\in\Phi}\frac{1}{n}\sum_{i=1}^n\sum_{k=1}^q g_{ik}\phi_k(\mathbf{x}_i)\right], \quad \mathfrak{G}_n(\Phi) \triangleq \mathbb{E}\left[\widehat{\mathfrak{G}}_\mathbf{X}(\Phi)\right]. \tag{6}$$

In terms of the Gaussian complexity, for a fixed $T_{\widehat{\mathcal{A}},\mathbf{X}^u}^{dac}(\mathcal{H})$,

$$\mathbb{E}_\mathbf{X}\left[\frac{1}{2n}\left\|(\mathbf{X}\widehat{\mathbf{B}}^{dac})_+\widehat{\mathbf{w}}^{dac} - (\mathbf{XB}^*)_+\mathbf{w}^*\right\|_2^2 \,\Big|\, \mathbf{X}^u\right]$$

$$\le \mathbb{E}_\mathbf{X}\left[\mathbf{z}^\top\left(\frac{1}{n}\left((\mathbf{X}\widehat{\mathbf{B}}^{dac})_+\widehat{\mathbf{w}}^{dac} - (\mathbf{XB}^*)_+\mathbf{w}^*\right)\right) \,\Big|\, \mathbf{X}^u\right]$$

$$\lesssim \sigma \cdot \mathbb{E}_\mathbf{X}\left[\widehat{\mathfrak{G}}_\mathbf{X}\left(T_{\widehat{\mathcal{A}},\mathbf{X}^u}^{dac}(\mathcal{H})\right) \,\Big|\, \mathbf{X}^u\right]$$

where the empirical Gaussian complexity can be upper bounded by

$$
\begin{aligned}
\widehat{\mathfrak{G}}_{\mathbf{X}}\left(T_{\widehat{\mathcal{A}},\mathbf{X}^u}^{dac}(\mathcal{H})\right) &= \mathop{\mathbb{E}}_{\mathbf{g}\sim\mathcal{N}(\mathbf{0},\mathbf{I}_n)}\left[\sup_{\mathbf{B}\in\mathcal{B},\|\mathbf{w}\|_1\leq R}\frac{1}{n}\mathbf{g}^\top(\mathbf{XB})_+\mathbf{w}\right] \\
&\leq\frac{C_w}{n}\,\mathop{\mathbb{E}}_{\mathbf{g}}\left[\sup_{\mathbf{B}\in\mathcal{B}}\left\|(\mathbf{XB})_+^\top\mathbf{g}\right\|_\infty\right] \\
&=\frac{C_w}{n}\,\mathop{\mathbb{E}}_{\mathbf{g}}\left[\sup_{\mathbf{b}\in\mathbb{S}^{d-1}}\mathbf{g}^\top\left(\mathbf{XP}_\mathcal{N}\mathbf{b}\right)_+\right] \qquad\text{(Lemma 6, $(\cdot)_+$ is 1-Lipschitz)} \\
&\leq\frac{C_w}{n}\,\mathop{\mathbb{E}}_{\mathbf{g}}\left[\sup_{\mathbf{b}\in\mathbb{S}^{d-1}}\mathbf{g}^\top\mathbf{XP}_\mathcal{N}\mathbf{b}\right] \\
&=\frac{C_w}{n}\,\mathop{\mathbb{E}}_{\mathbf{g}}\left[\left\|\mathbf{P}_\mathcal{N}\mathbf{X}^\top\mathbf{g}\right\|_2\right] \\
&\leq\frac{C_w}{n}\,\left(\mathop{\mathbb{E}}_{\mathbf{g}}\left[\left\|\mathbf{P}_\mathcal{N}\mathbf{X}^\top\mathbf{g}\right\|_2^2\right]\right)^{1/2} \\
&=\frac{C_w}{n}\,\sqrt{\mathrm{tr}(\mathbf{P}_\mathcal{N}\mathbf{X}^\top\mathbf{XP}_\mathcal{N})} \\
&=\frac{C_w}{\sqrt{n}}\,\sqrt{\frac{1}{n}\sum_{i=1}^n\|\mathbf{P}_\mathcal{N}\mathbf{x}_i\|_2^2}.
\end{aligned}
$$

Taking expectation over $\mathbf{X}\sim P^*$, we have that for a fixed $T_{\widehat{\mathcal{A}},\mathbf{X}^u}^{dac}(\mathcal{H})$ (*i.e.*, conditioned on $\mathbf{P}_\mathcal{N}$),

$$
\mathfrak{G}_n\left(T_{\widehat{\mathcal{A}},\mathbf{X}^u}^{dac}(\mathcal{H})\right)\leq\frac{C_w C_\mathcal{N}}{\sqrt{n}}.
$$

Leveraging Lemma 5, we have that under Assumption 1 and Assumption 2, conditioned on $\mathbf{P}_\mathcal{N}$, with high probability over $\mathbf{X}\sim P^*$,

$$
\left\|\frac{1}{n}\mathbf{P}_\mathcal{N}\mathbf{X}^\top\mathbf{XP}_\mathcal{N}\right\|_2\leq 1.1C\lesssim 1
$$

Then we can find some $C_\mathcal{N}>0$ with $\frac{1}{n}\sum_{i=1}^n\|\mathbf{P}_\mathcal{N}\mathbf{x}_i\|_2^2\leq C_\mathcal{N}^2$ such that,

$$
C_\mathcal{N}^2\leq\left(d-\widehat{d}_{aug}\right)\cdot\left\|\frac{1}{n}\mathbf{P}_\mathcal{N}\mathbf{X}^\top\mathbf{XP}_\mathcal{N}\right\|_2\lesssim d-\widehat{d}_{aug}.
$$

By Proposition 7, we have $C_\mathcal{N}\leq\sqrt{d-d_{aug}(\delta)}$ with probability at least $1-\delta$. ∎

### B.3 FORML RESULTS ON EXPANSION-BASED DATA AUGMENTATION

For the formal analysis of the expansion-based data augmentations, we first introduce some helpful notations.

With respect to an augmentation function $\mathcal{A}:\mathcal{X}\to 2^\mathcal{X}$ fulfilling the conditions in Definition 4, we define the neighborhood for an arbitrary $\mathbf{x}\in\mathcal{X}$, as well as for any $S\subseteq\mathcal{X}$, such that,

$$
NB(\mathbf{x})\triangleq\left\{\mathbf{x}'\in\mathcal{X}\mid\mathcal{A}(\mathbf{x})\cap\mathcal{A}(\mathbf{x}')\neq\emptyset\right\},\quad NB(S)\triangleq\cup_{\mathbf{x}\in S}NB(\mathbf{x}).
$$

Then Definition 4(b) implies that the corresponding neighborhood function $NB:\mathcal{X}\to 2^\mathcal{X}$ of $\mathcal{A}$ has **non-trivial expansion**:

$$
\emptyset\subsetneq S\cap\mathcal{X}_k\subsetneq\mathcal{X}_k\Rightarrow(NB(S)\backslash S)\cap\mathcal{X}_k\neq\emptyset\quad\forall\,k\in[K],\,S\subseteq\mathcal{X}.
$$

Intuitively, the non-trivial expansion guarantees that the neighborhood of any proper subset of a class always enlarges the subset within the class.

**Remark 2.** *The ground truth classifier is **invariant throughout the neighborhood**: for any augmentation function $\mathcal{A}:\mathcal{X}\to 2^\mathcal{X}$ satisfying the conditions in Definition 4, the corresponding neighborhood function satisfies that, given any $\mathbf{x}\in\mathcal{X}$, $h^*(\mathbf{x})=h^*(\mathbf{x}')$ for all $\mathbf{x}'\in NB(\mathbf{x})$.*

Notice that the previous data augmentation strength $\widehat{d}_{aug}$ (Definition 2) is not well defined for Definition 4 (since $\mathcal{X}$ needs not be a subset of $\mathbb{R}^d$). Therefore we need a new notion of data augmentation strength.

**Remark 3.** *The strength of expansion-based data augmentations (Definition 4) is characterized by the size of $\mathcal{A}(\mathbf{x})$. Intuitively, for any $\mathbf{x} \in \mathcal{X}$, the augmentations can be stronger (i.e., there exists $\mathbf{x}' \in \mathcal{A}(\mathbf{x})$ that is more different from $\mathbf{x}$) if $\mathcal{A}(\mathbf{x})$ is inclusion-wisely larger.*

In addition, for an arbitrary classifier $h \in \mathcal{H}$, we denote the majority label with respect to $h$ for each class,

$$\widehat{y}_k \triangleq \operatorname*{argmax}_{y \in [K]} \mathbb{P}_\mathbf{x} \left[ h(\mathbf{x}) = y \mid \mathbf{x} \in \mathcal{X}_k \right] \quad \forall\, k \in [K],$$

along with the respective class-wise local and global minority sets,

$$M_k \triangleq \left\{ \mathbf{x} \in \mathcal{X}_k \mid h(\mathbf{x}) \neq \widehat{y}_k \right\} \subsetneq \mathcal{X}_k \quad \forall\, k \in [K], \quad M \triangleq \bigcup_{k=1}^{K} M_k.$$

**Infinite unlabeled data.** As a warm-up, we simplify the setting by enforcing consistency over the population such that learning with the DAC constraints can be expressed as,

$$\widehat{h}_\mathcal{X}^{dac} \triangleq \operatorname*{argmin}_{h \in \mathcal{H}} \widehat{L}_{01}^{dac}(h) = \frac{1}{n} \sum_{i=1}^{n} \mathbf{1}\left\{ h\left(\mathbf{x}_i\right) \neq h^*\left(\mathbf{x}_i\right) \right\}$$

$$\text{s.t.} \quad h(\mathbf{x}) = h\left(\mathbf{x}'\right) \quad \forall\, \mathbf{x} \in \mathcal{X},\ \mathbf{x}' \in \mathcal{A}(\mathbf{x}).$$

We can also formulate algorithm in terms of the DAC operator,

$$\widehat{h}_\mathcal{X}^{dac} \triangleq \operatorname*{argmin}_{h \in T_{\mathcal{A},\mathcal{X}}^{dac}(\mathcal{H})} \widehat{L}_{01}^{dac}(h), \quad T_{\mathcal{A},\mathcal{X}}^{dac} \triangleq \left\{ h \in \mathcal{H} \mid h(\mathbf{x}) = h(\mathbf{x}') \,\forall\, \mathbf{x} \in \mathcal{X},\ \mathbf{x}' \in \mathcal{A}(\mathbf{x}) \right\}.$$

**Lemma 3.** *Given any $h \in T_{\mathcal{A},\mathcal{X}}^{dac}(\mathcal{H})$, for each $k \in [K]$, there exists some $\widehat{y}_k \in [K]$ such that $h(\mathbf{x}) = \widehat{y}_k$ for all $\mathbf{x} \in \mathcal{X}_k$.*

*Proof of Lemma 3.* It is sufficient to show that $M_k = \emptyset$ for all $k \in [K]$. By contradiction, suppose $M_k \neq \emptyset$. Then by the non-trivial expansion assumption in Definition 4, $(NB(M_k)\backslash M_k) \cap \mathcal{X}_k \neq \emptyset$. That is, there exists some $\mathbf{x} \in (NB(M_k)\backslash M_k) \cap \mathcal{X}_k$. But on one hand, $\mathbf{x} \in NB(M_k)$ implies that we can find some $\mathbf{x}' \in M_k$ and $\mathbf{x}'' \in \mathcal{A}(\mathbf{x}) \cap \mathcal{A}(\mathbf{x}')$. Therefore by the consistency constraint $h \in T_{\mathcal{A},\mathcal{X}}^{dac}(\mathcal{H})$, $h(\mathbf{x}'') = h(\mathbf{x}) = h(\mathbf{x}')$. On the other hand, $\mathbf{x} \in \mathcal{X}_k\backslash M_k$ implies that $h(\mathbf{x}) = \widehat{y}_k \neq h(\mathbf{x}')$, and leads to a contradiction. ∎

Restating Theorem 5 with the formal notations in the appendix: with high probability,

$$L_{01}\left(\widehat{h}_\mathcal{X}^{dac}\right) - L_{01}\left(h^*\right) \leq \frac{K \log K + \log n}{n}.$$

*Proof of Theorem 5.* By Lemma 3, since

$$T_{\mathcal{A},\mathcal{X}}^{dac}(\mathcal{H}) \subseteq \left\{ h \in \mathcal{H} \mid h(\mathbf{x}) = \widehat{y}_k \,\forall\, \mathbf{x} \in \mathcal{X}_k \right\} \quad \text{where} \quad \left\{ \widehat{y}_k \in [K] \right\}_{k \in [K]},$$

the consistency constraint yields a finite feasible classifier class with $\left| T_{\mathcal{A},\mathcal{X}}^{dac}(\mathcal{H}) \right| = K! \leq K^K$.

Meanwhile, since $\mathcal{A}(\mathbf{x}) \subseteq \left\{ \mathbf{x}' \in \mathcal{X} \mid h^*(\mathbf{x}) = h^*(\mathbf{x}') \right\}$, by construction, we know that $T_{\mathcal{A},\mathcal{X}}^{dac}(\mathcal{H})$ is realizable (*i.e.*, $h^* \in T_{\mathcal{A},\mathcal{X}}^{dac}(\mathcal{H})$ such that $\widehat{L}_{01}^{dac}(h^*) = 0$). Therefore,

$$\mathbb{P}\left[ L_{01}\left(\widehat{h}_\mathcal{X}^{dac}\right) > \epsilon \right] \leq \mathbb{P}\left[ \exists\, h \in T_{\mathcal{A},\mathcal{X}}^{dac}(\mathcal{H}) \,\Big|\, L_{01}(h) > \epsilon \,\wedge\, \widehat{L}_{01}^{dac}(h) = 0 \right],$$

where for every classifier $h \in T_{\mathcal{A},\mathcal{X}}^{dac}(\mathcal{H})$,

$$\mathbb{P}\left[ L_{01}(h) > \epsilon \,\wedge\, \widehat{L}_{01}^{dac}(h) = 0 \right] \leq (1 - \epsilon)^n \leq e^{-\epsilon n}.$$

Then by the union bound, we have

$$\mathbb{P}\left[L_{01}\left(\widehat{h}_{\mathcal{X}}^{dac}\right) > \epsilon\right] \leq \sum_{h \in T_{\mathcal{A},\mathcal{X}}^{dac}(\mathcal{H})} \mathbb{P}\left[L_{01}(h) > \epsilon \, \wedge \, \widehat{L}_{01}^{dac}(h) = 0\right]$$

$$\leq \left|T_{\mathcal{A},\mathcal{X}}^{dac}(\mathcal{H})\right| \cdot e^{-\epsilon n}.$$

The proof is completed by upper bounding the above, $\mathbb{P}\left[L_{01}\left(\widehat{h}_{\mathcal{X}}^{dac}\right) > \epsilon\right] \leq 1/n$, when taking

$$\epsilon = \frac{\log\left|T_{\mathcal{A},\mathcal{X}}^{dac}(\mathcal{H})\right| + \log n}{n} \leq \frac{K \log K + \log n}{n},$$

and observing that $L_{01}(h^*) = 0$.

$\blacksquare$

**Finite unlabeled data.** For the finite unlabeled data case, we start by introducing some additional concepts and notations. We concretize the classifier class $\mathcal{H}$ with a proper function class $\mathcal{F} \subseteq \left\{f : \mathcal{X} \to \mathbb{R}^K\right\}$ such that $\mathcal{H} = \left\{h(\mathbf{x}) \triangleq \operatorname{argmax}_{k \in [K]} f(\mathbf{x})_k \,\middle|\, f \in \mathcal{F}\right\}$. Specifically, we adapt the existing setting in Wei et al. (2021); Cai et al. (2021), and consider $\mathcal{F}$ as a class of fully connected neural networks. To constrain the feasible hypothesis class through the DAC regularization with finite unlabeled observations $\mathbf{X}^u = [\mathbf{x}_1^u, \dots, \mathbf{x}_N^u]^\top$, we leverage, from Wei & Ma (2021), the notion of all-layer-margin, $m : \mathcal{F} \times \mathcal{X} \times \mathcal{Y} \to \mathbb{R}_{\geq 0}$, that measures the maximum possible perturbation in all layers of $f$ while maintaining the prediction $y$. Precisely, given any $f \in \mathcal{F}$ such that $f(\mathbf{x}) = \mathbf{W}_p \varphi(\dots \varphi(\mathbf{W}_1 \mathbf{x})\dots)$ for some activation function $\varphi : \mathbb{R} \to \mathbb{R}$ and parameters $\left\{\mathbf{W}_\iota \in \mathbb{R}^{d_\iota \times d_{\iota-1}}\right\}_{\iota=1}^p$, we can write $f = f_{2p-1} \circ \cdots \circ f_1$ where $f_{2\iota-1}(\mathbf{x}) = \mathbf{W}_\iota \mathbf{x}$ for all $\iota \in [p]$ and $f_{2\iota}(\mathbf{z}) = \varphi(\mathbf{z})$ for $\iota \in [p-1]$. For an arbitrary set of perturbation vectors $\boldsymbol{\delta} = (\boldsymbol{\delta}_1, \dots, \boldsymbol{\delta}_{2p-1})$ such that $\boldsymbol{\delta}_{2\iota-1}, \boldsymbol{\delta}_{2\iota} \in \mathbb{R}^{d_\iota}$ for all $\iota$, let $f(\mathbf{x}, \boldsymbol{\delta})$ be the perturbed neural network defined recursively such that

$$\widetilde{\mathbf{z}}_1 = f_1(\mathbf{x}) + \|\mathbf{x}\|_2 \, \boldsymbol{\delta}_1,$$
$$\widetilde{\mathbf{z}}_\iota = f_\iota(\widetilde{\mathbf{z}}_{\iota-1}) + \|\widetilde{\mathbf{z}}_{\iota-1}\|_2 \, \boldsymbol{\delta}_\iota \quad \forall \, \iota = 2, \dots, 2p-1,$$
$$f(\mathbf{x}, \boldsymbol{\delta}) = \widetilde{\mathbf{z}}_{2p-1}.$$

The all-layer-margin $m(f, \mathbf{x}, y)$ measures the minimum norm of the perturbation $\boldsymbol{\delta}$ such that $f(\mathbf{x}, \boldsymbol{\delta})$ fails to provide the classification $y$,

$$m(f, \mathbf{x}, y) \triangleq \min_{\boldsymbol{\delta} = (\boldsymbol{\delta}_1, \dots, \boldsymbol{\delta}_{2p-1})} \sqrt{\sum_{\iota=1}^{2p-1} \|\boldsymbol{\delta}_\iota\|_2^2} \quad \text{s.t.} \quad \operatorname*{argmax}_{k \in [K]} f(\mathbf{x}, \boldsymbol{\delta})_k \neq y. \tag{7}$$

With the notion of all-layer-margin established, for any $\mathcal{A} : \mathcal{X} \to 2^{\mathcal{X}}$ that satisfies conditions in Definition 4, the robust margin is defined as

$$m_{\mathcal{A}}(f, \mathbf{x}) \triangleq \sup_{\mathbf{x}' \in \mathcal{A}(\mathbf{x})} m\left(f, \mathbf{x}', \operatorname*{argmax}_{k \in [k]} f(\mathbf{x})_k\right).$$

With merely finite unlabeled data, comparing to enforcing consistency over population, we need stronger assumptions on data augmentations in addition to the ones in Definition 4, as specified in Definition 5 below.

**Definition 5** (($\mathcal{F}, \tau$)-expansion-based data augmentation). *With respect to the given function class $\mathcal{F}$ and some proper constant $\tau > 0$, Let $\mathcal{A} : \mathcal{X} \to 2^{\mathcal{X}}$ be a function that satisfies conditions in Definition 4, and additionally,*

$$\sup_{f \in \mathcal{F}} \inf_{\mathbf{x} \in \mathcal{X}} m_{\mathcal{A}}(f, \mathbf{x}) > \tau. \tag{8}$$

*Then for any sample $\mathbf{x} \in \mathcal{X}$, $\mathbf{x}' \in \mathcal{X}$ is a ($\mathcal{F}, \tau$)-expansion-based data augmentation of $\mathbf{x}$ induced by $\mathcal{A}$ if $\mathbf{x}' \in \mathcal{A}(\mathbf{x})$.*

**Remark 4.** *The existence of the $(\mathcal{F}, \tau)$-expansion-based data augmentations relies on the suitable choice of $\tau$ with respect to the given $\mathcal{F}$. For a fixed $\mathcal{F}$, as $\tau$ increases, Equation (8) imposes stronger assumption on $\mathcal{A}$ (i.e., $\mathcal{A}(\mathbf{x})$ is forced to be inclusion-wisely smaller, where $\mathcal{A}$ becomes weaker), while better theoretical guarantee (Theorem 12) can be achieved.*

We consider a class of $p$-layer fully connected neural networks with maximum width $q$,

$$\mathcal{F} = \left\{ f : \mathcal{X} \to \mathbb{R}^K \mid f = f_{2p-1} \circ \cdots \circ f_1, \ f_{2\iota-1}(\mathbf{x}) = \mathbf{W}_\iota \mathbf{x}, \ f_{2\iota}(\mathbf{z}) = \varphi(\mathbf{z}) \ \forall \ \iota = 1, \ldots, p \right\},$$

where with $\mathbf{W}_\iota \in \mathbb{R}^{d_\iota \times d_{\iota-1}}$ for all $\iota \in [p]$, $q \triangleq \max_{\iota=0,\ldots,p} d_\iota$. The goal is to learn a classifier $\widehat{h}_{\mathbf{X}^u}^{dac} = \mathrm{argmax}_{k \in [K]} \ \widehat{f}_{n,N}^{dac}(\mathbf{x})_k$ where $\widehat{f}_{n,N}^{dac} \in \mathcal{F}$ is a fully connected neural network. With respect to $\mathcal{F}$, we choose a suitably large $\tau > 0$ such that there exists a $\mathcal{A} : \mathcal{X} \to 2^\mathcal{X}$ fulfilling the conditions in Definition 5. To enforce sufficiently strong consistency of $f$ with finite unlabeled samples, we incorporate the DAC regularization with a positive robust margin such that,

$$\widehat{h}_{\mathbf{X}^u}^{dac} \triangleq \underset{h \in \mathcal{H}}{\mathrm{argmin}} \ \widehat{L}_{01}^{dac}(h) = \frac{1}{n} \sum_{i=1}^n \mathbf{1} \left\{ h\left(\mathbf{x}_i\right) \neq h^*\left(\mathbf{x}_i\right) \right\} \tag{9}$$

$$\text{s.t.} \quad m_\mathcal{A}(f, \mathbf{x}_i^u) > \tau \quad \forall \ i \in [N].$$

In terms of the DAC operator, the algorithm can be restated as $\widehat{h}_{\mathbf{X}^u}^{dac} \triangleq \mathrm{argmin}_{h \in T_{\mathcal{A},\mathbf{X}^u}^{dac}(\mathcal{H})} \ \widehat{L}_{01}^{dac}(h)$ with

$$T_{\mathcal{A},\mathbf{X}^u}^{dac}(\mathcal{H}) \triangleq \left\{ h \in \mathcal{H} \mid m_\mathcal{A}(f, \mathbf{x}_i^u) > \tau \quad \forall \ i \in [N] \right\}.$$

**Definition 6** (Expansion assumptions, Wei et al. (2021); Cai et al. (2021))**.** *The marginal distribution $P^*(\mathbf{x})$ satisfies*

(a) *$(q, \xi)$-constant expansion if given any $S \subseteq \mathcal{X}$ with $P^*(S) \geq q$ and $P^*(S \cap \mathcal{X}_k) \leq \frac{1}{2}$ for all $k \in [K]$, $P^*(NB(S)) \geq \min\{P^*(S), \xi\} + P^*(S)$;*

(b) *$(a, c)$-multiplicative expansion if for all $k \in [K]$, given any $S \subseteq \mathcal{X}$ with $P^*(S \cap \mathcal{X}_k) \leq a$, $P^*(NB(S) \cap \mathcal{X}_k) \geq \min\{c \cdot P^*(S \cap \mathcal{X}_k), 1\}$.*

**Proposition 11** (Wei et al. (2021) Theorem 3.7, Cai et al. (2021) Proposition 2.2)**.** *For any $\delta \in (0, 1)$, with probability at least $1 - \delta/2$, there exists some $\mu$ such that*

$$\mathbb{P}_{P^*}\left[\exists \ \mathbf{x}' \in \mathcal{A}(\mathbf{x}) \ s.t. \ h(\mathbf{x}) \neq h(\mathbf{x}')\right] \leq \mu \leq \widetilde{O}\left(\frac{\sum_{\iota=1}^p \sqrt{q} \|\mathbf{W}_\iota\|_F}{\tau \sqrt{N}} + \sqrt{\frac{\log(1/\delta) + p \log N}{N}}\right)$$

*for all $h \in T_{\mathcal{A},\mathbf{X}^u}^{dac}(\mathcal{H})$, where $\widetilde{O}(\cdot)$ hides polylogarithmic factors in $N$ and $d$.*

**Theorem 12** (DAC with expansion-based data augmentations over finite samples)**.** *With proper choices of $\tau > 0$, for any $\mathcal{A} : \mathcal{X} \to 2^\mathcal{X}$ inducing $(\mathcal{F}, \tau)$-expansion-based data augmentations, learning the classifier with DAC regularization, Equation (9), provides that, for any $\delta \in (0, 1)$, with probability at least $1 - \delta$,*

$$L_{01}\left(\widehat{h}_{\mathbf{X}^u}^{dac}\right) - L_{01}\left(h^*\right) \leq 4\mathfrak{R} + \sqrt{\frac{2\log(4/\delta)}{n}}, \tag{10}$$

*In specific, for some $q < \frac{1}{2}$, $c > 1$, with a sufficiently large unlabeled sample size $N$ such that $\mu \leq \frac{1}{4}\min\{c - 1, 1\}$,*

(a) *when $P^*(\mathbf{x})$ satisfies $(q, 2\mu)$-constant expansion,*

$$\mathfrak{R} \leq \sqrt{\frac{2K \log K}{n}} + 2K \max\{q, 2\mu\},$$

(b) *while when $P^*(\mathbf{x})$ satisfies $(\frac{1}{2}, c)$-multiplicative expansion,*

$$\mathfrak{R} \leq \sqrt{\frac{2K \log K}{n}} + \frac{4K\mu}{\min\{c - 1, 1\}},$$

*where we recall from Proposition 11 that,*

$$\mu \leq \widetilde{O} \left( \frac{\sum_{\iota=1}^p \sqrt{q} \, \|\mathbf{W}_\iota\|_F}{\tau \sqrt{N}} + \sqrt{\frac{\log(1/\delta) + p \log N}{N}} \right).$$

**Lemma 4** (Cai et al. (2021), Lemma A.1). *For any $h \in T_{\mathcal{A},\mathbf{X}^u}^{dac}(\mathcal{H})$, when $P^*$ satisfies*

(a) $(q, 2\mu)$-*constant expansion with* $q < \frac{1}{2}$, $P^*(M) \leq \max\{q, 2\mu\}$;

(b) $\left(\frac{1}{2}, c\right)$-*multiplicative expansion with* $c > 1 + 4\mu$, $P^*(M) \leq \max\left\{\frac{2\mu}{c-1}, 2\mu\right\}$.

*Proof of Lemma 4.* We start with the proof for Lemma 4 (a). By definition of $M_k$ and $\widehat{y}_k$, we know that $M_k = M \cap \mathcal{X}_k \leq \frac{1}{2}$. Therefore, for any $0 < q < \frac{1}{2}$, one of the following two cases holds:

(i) $P^*(M) < q$;

(ii) $P^*(M) \geq q$. Since $P^*(M \cap \mathcal{X}_k) < \frac{1}{2}$ for all $k \in [K]$ holds by construction, with the $(q, 2\mu)$-constant expansion, $P^*(NB(M)) \geq \min\{P^*(M), 2\mu\} + P^*(M)$.
Meanwhile, since the ground truth classifier $h^*$ is invariant throughout the neighborhoods, $NB(M_k) \cap NB(M_{k'}) = \emptyset$ for $k \neq k'$, and therefore $NB(M) \backslash M = \bigcup_{k=1}^K NB(M_k) \backslash M_k$ with each $NB(M_k) \backslash M_k$ disjoint. Then, we observe that for each $\mathbf{x} \in NB(M) \backslash M$, here exists some $k = h^*(\mathbf{x})$ such that $\mathbf{x} \in NB(M_k) \backslash M_k$. $\mathbf{x} \in \mathcal{X}_k \backslash M_k$ implies that $h(\mathbf{x}) = \widehat{y}_k$, while $\mathbf{x} \in NB(M_k)$ suggests that there exists some $\mathbf{x}' \in \mathcal{A}(\mathbf{x}) \cap \mathcal{A}(\mathbf{x}'')$ where $\mathbf{x}'' \in M_k$ such that either $h(\mathbf{x}') = \widehat{y}_k$ and $h(\mathbf{x}') \neq h(\mathbf{x}'')$ for $\mathbf{x}' \in \mathcal{A}(\mathbf{x}'')$, or $h(\mathbf{x}') \neq \widehat{y}_k$ and $h(\mathbf{x}') \neq h(\mathbf{x})$ for $\mathbf{x}' \in \mathcal{A}(\mathbf{x})$. Therefore, we have

$$P^*(NB(M) \backslash M) \leq 2\mathbb{P}_{P^*}[\exists \, \mathbf{x}' \in \mathcal{A}(\mathbf{x}) \text{ s.t. } h(\mathbf{x}) \neq h(\mathbf{x}')] \leq 2\mu.$$

Moreover, since $P^*(NB(M)) - P^*(M) \leq P^*(NB(M) \backslash M) \leq 2\mu$, we know that

$$\min\{P^*(M), 2\mu\} + P^*(M) \leq P^*(NB(M)) \leq P^*(M) + 2\mu.$$

That is, $P^*(M) \leq 2\mu$.

Overall, we have $P^*(M) \leq \max\{q, 2\mu\}$.

To show Lemma 4 (b), we recall from Wei et al. (2021) Lemma B.6 that for any $c > 1 + 4\mu$, $\left(\frac{1}{2}, c\right)$-multiplicative expansion implies $\left(\frac{2\mu}{c-1}, 2\mu\right)$-constant expansion. Then leveraging the proof for Lemma 4 (a), with $q = \frac{2\mu}{c-1}$, we have $P^*(M) \leq \max\left\{\frac{2\mu}{c-1}, 2\mu\right\}$. ∎

*Proof of Theorem 12.* To show Equation (10), we leverage the proof of Proposition 8, and observe that $B = 1$ with the zero-one loss. Therefore, when conditioned on $T_{\mathcal{A},\mathbf{X}^u}^{dac}(\mathcal{H})$, for any $\delta \in (0,1)$, with probability at least $1 - \delta/2$,

$$L_{01}\left(\widehat{h}_{\mathbf{X}^u}^{dac}\right) - L_{01}(h^*) \leq 4\mathfrak{R}_n\left(l_{01} \circ T_{\mathcal{A},\mathbf{X}^u}^{dac}(\mathcal{H})\right) + \sqrt{\frac{2\log(4/\delta)}{n}}.$$

For the upper bounds of the Rademacher complexity, let $\widetilde{\mu} \triangleq \sup_{h \in T_{\mathcal{A},\mathbf{X}^u}^{dac}(\mathcal{H})} P^*(M)$ where $M$ denotes the global minority set with respect to $h \in T_{\mathcal{A},\mathbf{X}^u}^{dac}(\mathcal{H})$. Lemma 4 suggests that

(a) when $P^*$ satisfies $(q, 2\mu)$-constant expansion for some $q < \frac{1}{2}$, $\widetilde{\mu} \leq \max\{q, 2\mu\}$; while

(b) when $P^*$ satisfies $\left(\frac{1}{2}, c\right)$-multiplicative expansion for some $c > 1 + 4\mu$, $\widetilde{\mu} \leq \frac{2\mu}{\min\{c-1,1\}}$.

Then, it is sufficient to show that, conditioned on $T_{\mathcal{A},\mathbf{X}^u}^{dac}(\mathcal{H})$,

$$\mathfrak{R}_n\left(l_{01} \circ T_{\mathcal{A},\mathbf{X}^u}^{dac}(\mathcal{H})\right) \leq \sqrt{\frac{2K \log K}{n}} + 2K\widetilde{\mu}. \tag{11}$$

To show this, we first consider a fixed set of $n$ observations in $\mathcal{X}$, $\mathbf{X} = [\mathbf{x}_1, \ldots, \mathbf{x}_n]^\top \in \mathcal{X}^n$. Let the number of distinct behaviors over $\mathbf{X}$ in $T_{\mathcal{A},\mathbf{X}^u}^{dac}(\mathcal{H})$ be

$$\mathfrak{s}\left(T_{\mathcal{A},\mathbf{X}^u}^{dac}(\mathcal{H}), \mathbf{X}\right) \triangleq \left| \left\{ [h(\mathbf{x}_1), \ldots, h(\mathbf{x}_n)] \mid h \in T_{\mathcal{A},\mathbf{X}^u}^{dac}(\mathcal{H}) \right\} \right|.$$

Then, by the Massart's finite lemma, the empirical rademacher complexity with respect to $\mathbf{X}$ is upper bounded by

$$\widehat{\mathfrak{R}}_{\mathbf{X}}\left(l_{01} \circ T_{\mathcal{A},\mathbf{X}^u}^{dac}(\mathcal{H})\right) \leq \sqrt{\frac{2 \log \mathfrak{s}\left(T_{\mathcal{A},\mathbf{X}^u}^{dac}(\mathcal{H}), \mathbf{X}\right)}{n}}.$$

By the concavity of $\sqrt{\log(\cdot)}$, we know that,

$$\mathfrak{R}_n\left(l_{01} \circ T_{\mathcal{A},\mathbf{X}^u}^{dac}(\mathcal{H})\right) = \mathbb{E}_{\mathbf{X}}\left[\widehat{\mathfrak{R}}_{\mathbf{X}}\left(l_{01} \circ T_{\mathcal{A},\mathbf{X}^u}^{dac}(\mathcal{H})\right)\right] \leq \mathbb{E}_{\mathbf{X}}\left[\sqrt{\frac{2 \log \mathfrak{s}\left(T_{\mathcal{A},\mathbf{X}^u}^{dac}(\mathcal{H}), \mathbf{X}\right)}{n}}\right]$$

$$\leq \sqrt{\frac{2 \log \mathbb{E}_{\mathbf{X}}\left[\mathfrak{s}\left(T_{\mathcal{A},\mathbf{X}^u}^{dac}(\mathcal{H}), \mathbf{X}\right)\right]}{n}}. \tag{12}$$

Since $P^*(M) \leq \widetilde{\mu} \leq \frac{1}{2}$ for all $h \in T_{\mathcal{A},\mathbf{X}^u}^{dac}(\mathcal{H})$, we have that, conditioned on $T_{\mathcal{A},\mathbf{X}^u}^{dac}(\mathcal{H})$,

$$\mathbb{E}_{\mathbf{X}}\left[\mathfrak{s}\left(T_{\mathcal{A},\mathbf{X}^u}^{dac}(\mathcal{H}), \mathbf{X}\right)\right] \leq \sum_{r=0}^n \binom{n}{r}\widetilde{\mu}^r (1-\widetilde{\mu})^{n-r} \cdot K^K \cdot K^r$$

$$\leq K^K \sum_{r=0}^n \binom{n}{r}(\widetilde{\mu}K)^r (1-\widetilde{\mu})^{n-r}$$

$$= K^K (1-\widetilde{\mu}+K\widetilde{\mu})^n$$

$$\leq K^K \cdot e^{Kn\widetilde{\mu}}.$$

Plugging this into Equation (12) yields Equation (11). Finally, the randomness in $T_{\mathcal{A},\mathbf{X}^u}^{dac}(\mathcal{H})$ is quantified by $\widetilde{\mu}, \mu$, and upper bounded by Proposition 11. ∎

### B.4 SUPPLEMENTARY EXAMPLE: DOMAIN ADAPTATION

As a supplementary example, we demonstrate the possible failure of the ERM on augmented training set, and how the DAC regularization can serve as a remedy, with an illustrative linear regression problem in the domain adaptation setting: where the training samples are drawn from some source distribution $P^s$, while the excess risk is tested over a related but different distribution $P^t$, known as the target distribution. Specifically, assuming that $\mathbb{E}_{P^s}[y|\mathbf{x}]$ and $\mathbb{E}_{P^t}[y|\mathbf{x}]$ are distinct, but there exists some invariant feature subspace $\mathcal{X}_r \subset \mathcal{X}$ such that $P^s[y|\mathbf{x} \in \mathcal{X}_r] = P^t[y|\mathbf{x} \in \mathcal{X}_r]$, we aim to demonstrate the advantage of the DAC regularization over the ERM on augmented training set, with a provable separation in the respective excess risks.

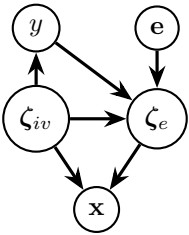

Figure 4: Causal graph shared by $P^s$ and $P^t$.

**Source and target distributions.** Formally, the source and target distributions are concretized with the causal graph in Figure 4. For both $P^s$ and $P^t$, the observable feature $\mathbf{x}$ is described via a linear generative model in terms of two latent features, the 'invariant' feature $\boldsymbol{\zeta}_{iv} \in \mathbb{R}^{d_{iv}}$ and the 'environmental' feature $\boldsymbol{\zeta}_e \in \mathbb{R}^{d_e}$:

$$\mathbf{x} = g(\boldsymbol{\zeta}_{iv}, \boldsymbol{\zeta}_e) \triangleq \mathbf{S}[\boldsymbol{\zeta}_{iv}; \boldsymbol{\zeta}_e] = \mathbf{S}_{iv}\boldsymbol{\zeta}_{iv} + \mathbf{S}_e\boldsymbol{\zeta}_e,$$

where $\mathbf{S} = [\mathbf{S}_{iv}, \mathbf{S}_e] \in \mathbb{R}^{d \times (d_{iv} + d_e)}$ ($d_{iv} + d_e \le d$) consists of orthonormal columns. Let the label $y$ depends only on the invariant feature $\boldsymbol{\zeta}_{iv}$ for both domains,

$$y = (\boldsymbol{\theta}^*)^\top \mathbf{x} + z = (\boldsymbol{\theta}^*)^\top \mathbf{S}_{iv} \boldsymbol{\zeta}_{iv} + z, \quad z \sim \mathcal{N}\left(0, \sigma^2\right), \quad z \perp \boldsymbol{\zeta}_{iv},$$

for some $\boldsymbol{\theta}^* \in \text{Range}\,(\mathbf{S}_{iv})$ such that $P^s[y|\boldsymbol{\zeta}_{iv}] = P^t[y|\boldsymbol{\zeta}_{iv}]$, while the environmental feature $\boldsymbol{\zeta}_e$ is conditioned on $y$, $\boldsymbol{\zeta}_{iv}$, (along with the Gaussian noise $z$), and varies across different domains $\mathbf{e}$ with $\mathbb{E}_{P^s}[y|\mathbf{x}] \ne \mathbb{E}_{P^t}[y|\mathbf{x}]$. In other words, with the square loss $l(h(\mathbf{x}), y) = \frac{1}{2}(h(\mathbf{x}) - y)^2$, the optimal hypotheses that minimize the expected excess risk over the source and target distributions are distinct. Therefore, learning via the ERM with training samples from $P^s$ can overfit the source distribution, in which scenario identifying the invariant feature subspace $\text{Range}\,(\mathbf{S}_{iv})$ becomes indispensable for achieving good generalization in the target domain.

Moreover, we assume the following regularity conditions on the source and target distributions:

**Assumption 3** (Regularity conditions for $P^s$ and $P^t$)**.** *Let $P^s$ satisfy Assumption 1, and $P^t$ satisfy the following: let $\mathbb{E}_{P^t}[\mathbf{x}\mathbf{x}^\top] \succ 0$, and*

*(a) for the invariant feature, $c_{t,iv}\mathbf{I}_{d_{iv}} \preccurlyeq \mathbb{E}_{P^t}[\boldsymbol{\zeta}_{iv}\boldsymbol{\zeta}_{iv}^\top] \preccurlyeq C_{t,iv}\mathbf{I}_{d_{iv}}$ for some $C_{t,iv} \ge c_{t,iv} = \Theta(1)$;*
*(b) for the environmental feature, $\mathbb{E}_{P^t}[\boldsymbol{\zeta}_e\boldsymbol{\zeta}_e^\top] \succcurlyeq c_{t,e}\mathbf{I}_{d_e}$ for some $c_{t,e} > 0$, and $\mathbb{E}_{P^t}[z \cdot \boldsymbol{\zeta}_e] = \mathbf{0}$.*

**Training samples and data augmentations.** For a fair comparison between learning with the DAC regularization and the ERM on augmented training set, we restrict to the supervised learning setting: $\mathbf{X}^u = \mathbf{X} \in \mathcal{X}^n$. Recall that we denote the augmented training sets, including and excluding the original samples, respectively with

$$\widetilde{\mathcal{A}}(\mathbf{X}) = [\mathbf{x}_1; \cdots ; \mathbf{x}_n; \mathbf{x}_{1,1}; \cdots ; \mathbf{x}_{n,1}; \cdots ; \mathbf{x}_{1,\alpha}; \cdots ; \mathbf{x}_{n,\alpha}] \in \mathcal{X}^{(1+\alpha)n},$$
$$\widehat{\mathcal{A}}(\mathbf{X}) = [\mathbf{x}_{1,1}; \cdots ; \mathbf{x}_{n,1}; \cdots ; \mathbf{x}_{1,\alpha}; \cdots ; \mathbf{x}_{n,\alpha}] \in \mathcal{X}^{\alpha n}.$$

In particular, we consider a set of augmentations that only perturb the environmental feature $\boldsymbol{\zeta}_e$, while keep the invariant feature $\boldsymbol{\zeta}_{iv}$ intact,

$$\mathbf{S}_{iv}^\top \mathbf{x}_i = \mathbf{S}_{iv}^\top \mathbf{x}_{i,j}, \quad \mathbf{S}_e^\top \mathbf{x}_i \ne \mathbf{S}_e^\top \mathbf{x}_{i,j} \quad \forall\, i \in [n],\, j \in [\alpha]. \tag{13}$$

We recall the notion $\widehat{d}_{aug} = \text{rank}\left(\widehat{\mathcal{A}}(\mathbf{X}) - \mathbf{M}\mathbf{X}\right) = \text{rank}\left(\widetilde{\mathcal{A}}(\mathbf{X}) - \widetilde{\mathbf{M}}\mathbf{X}\right)$ (notice that $0 \le \widehat{d}_{aug} \le d_e$), and assume that $\mathbf{X}$ and $\widehat{\mathcal{A}}(\mathbf{X})$ are representative enough:

**Assumption 4** (Diversity of $\mathbf{X}$ and $\widehat{\mathcal{A}}(\mathbf{X})$)**.** *$(\mathbf{X}, \mathbf{y}) \in \mathcal{X}^n \times \mathcal{Y}^n$ is sufficiently large with $n \gg \rho^4 d_{iv}$, $\boldsymbol{\theta}^* \in \text{span}\,\{\mathbf{x}_i \mid i \in [n]\}$, and $\widehat{d}_{aug} = d_e$.*

**Excess risks in target domain.** Learning from the linear hypothesis class $\mathcal{H} = \left\{h(\mathbf{x}) = \boldsymbol{\theta}^\top \mathbf{x} \mid \boldsymbol{\theta} \in \mathbb{R}^d\right\}$, with the DAC regularization on $h(\mathbf{x}_i) = h(\mathbf{x}_{i,j})$, we have

$$\widehat{\boldsymbol{\theta}}^{dac} = \underset{\boldsymbol{\theta} \in T_{\widehat{\mathcal{A}},\mathbf{X}}^{dac}(\mathcal{H})}{\text{argmin}} \frac{1}{2n} \|\mathbf{y} - \mathbf{X}\boldsymbol{\theta}\|_2^2, \quad T_{\widehat{\mathcal{A}},\mathbf{X}}^{dac}(\mathcal{H}) = \left\{h(\mathbf{x}) = \boldsymbol{\theta}^\top \mathbf{x} \mid \widehat{\mathcal{A}}(\mathbf{X})\boldsymbol{\theta} = \mathbf{M}\mathbf{X}\boldsymbol{\theta}\right\},$$

while with the ERM on augmented training set,

$$\widehat{\boldsymbol{\theta}}^{da-erm} = \underset{\boldsymbol{\theta} \in \mathbb{R}^d}{\text{argmin}} \frac{1}{2(1+\alpha)n} \left\|\widetilde{\mathbf{M}}\mathbf{y} - \widetilde{\mathcal{A}}(\mathbf{X})\boldsymbol{\theta}\right\|_2^2,$$

where $\mathbf{M}$ and $\widetilde{\mathbf{M}}$ denote the vertical stacks of $\alpha$ and $1+\alpha$ identity matrices of size $n \times n$, respectively as denoted earlier.

We are interested in the excess risk on $P^t$: $L_t(\boldsymbol{\theta}) - L_t(\boldsymbol{\theta}^*)$ where $L_t(\boldsymbol{\theta}) \triangleq \mathbb{E}_{P^t(\mathbf{x},y)}\left[\frac{1}{2}(y - \mathbf{x}^\top\boldsymbol{\theta})^2\right]$.

**Theorem 13** (Domain adaptation with DAC)**.** *Under Assumption 3(a) and Assumption 4, $\widehat{\boldsymbol{\theta}}^{dac}$ satisfies that, with constant probability,*

$$\mathbb{E}_{P^s}\left[L_t(\widehat{\boldsymbol{\theta}}^{dac}) - L_t(\boldsymbol{\theta}^*)\right] \lesssim \frac{\sigma^2 d_{iv}}{n}. \tag{14}$$

**Theorem 14** (Domain adaptation with ERM on augmented samples). *Under Assumption 3 and Assumption 4, $\widehat{\boldsymbol{\theta}}^{dac}$ and $\widehat{\boldsymbol{\theta}}^{da-erm}$ satisfies that,*

$$\mathbb{E}_{P^s}\left[L_t(\widehat{\boldsymbol{\theta}}^{da-erm}) - L_t(\boldsymbol{\theta}^*)\right] \geq \mathbb{E}_{P^s}\left[L_t(\widehat{\boldsymbol{\theta}}^{dac}) - L_t(\boldsymbol{\theta}^*)\right] + c_{t,e} \cdot EER_e, \tag{15}$$

*for some $EER_e > 0$.*

In contrast to $\widehat{\boldsymbol{\theta}}^{dac}$ where the DAC constraints enforce $\mathbf{S}_e^\top \widehat{\boldsymbol{\theta}}^{dac} = \mathbf{0}$ with a sufficiently diverse $\widehat{\mathcal{A}}(\mathbf{X})$ (Assumption 4), the ERM on augmented training set fails to filter out the environmental feature in $\widehat{\boldsymbol{\theta}}^{da-erm}$: $\mathbf{S}_e^\top \widehat{\boldsymbol{\theta}}^{da-erm} \neq \mathbf{0}$. As a consequence, the expected excess risk of $\widehat{\boldsymbol{\theta}}^{da-erm}$ in the target domain can be catastrophic when $c_{t,e} \to \infty$, as instantiated by Example 2.

**Proofs and instantiation.** We first recall from the beginning of Appendix B that

$$\mathbf{P}_{\mathcal{N}} \triangleq \mathbf{I}_d - \left(\widehat{\mathcal{A}}(\mathbf{X}) - \mathbf{M}\mathbf{X}\right)^\dagger \left(\widehat{\mathcal{A}}(\mathbf{X}) - \mathbf{M}\mathbf{X}\right)$$

is the orthogonal projector onto the dimension-$(d - \widehat{d}_{aug})$ null space of $\widehat{\mathcal{A}}(\mathbf{X}) - \mathbf{M}\mathbf{X}$. Furthermore, let $\mathbf{P}_{iv} \triangleq \mathbf{S}_{iv}\mathbf{S}_{iv}^\top$ and $\mathbf{P}_e \triangleq \mathbf{S}_e\mathbf{S}_e^\top$ be the orthogonal projectors onto the invariant and environmental feature subspaces, respectively, such that $\mathbf{x} = \mathbf{S}_{iv}\boldsymbol{\zeta}_{iv} + \mathbf{S}_e\boldsymbol{\zeta}_e = (\mathbf{P}_{iv} + \mathbf{P}_e)\mathbf{x}$ for all $\mathbf{x}$.

*Proof of Theorem 13.* By construction Equation (13), $\left(\widehat{\mathcal{A}}(\mathbf{X}) - \mathbf{M}\mathbf{X}\right)\mathbf{P}_{iv} = \mathbf{0}$, and it follows that $\mathbf{P}_{iv} \preccurlyeq \mathbf{P}_{\mathcal{N}}$. Meanwhile from Assumption 4, $\widehat{d}_{aug} = d_e$ implies that $\dim(\mathbf{P}_{\mathcal{N}}) = d_{iv}$. Therefore, $\mathbf{P}_{iv} = \mathbf{P}_{\mathcal{N}}$, and the data augmentation consistency constraints can be restated as

$$T_{\widehat{\mathcal{A}},\mathbf{X}}^{dac}(\mathcal{H}) \triangleq \left\{h(\mathbf{x}) = \boldsymbol{\theta}^\top \mathbf{x} \mid \mathbf{P}_{\mathcal{N}}\boldsymbol{\theta} = \boldsymbol{\theta}\right\} = \left\{h(\mathbf{x}) = \boldsymbol{\theta}^\top \mathbf{x} \mid \mathbf{P}_{iv}\boldsymbol{\theta} = \boldsymbol{\theta}\right\}$$

Then with $\boldsymbol{\theta}^* \in \text{span}\{\mathbf{x}_i \mid i \in [n]\}$ from Assumption 4,

$$\widehat{\boldsymbol{\theta}}^{dac} - \boldsymbol{\theta}^* = \frac{1}{n}\widehat{\boldsymbol{\Sigma}}_{\mathbf{X}_{iv}}^\dagger \mathbf{P}_{iv}\mathbf{X}^\top(\mathbf{X}\mathbf{P}_{iv}\boldsymbol{\theta}^* + \mathbf{z}) - \boldsymbol{\theta}^* = \frac{1}{n}\widehat{\boldsymbol{\Sigma}}_{\mathbf{X}_{iv}}^\dagger \mathbf{P}_{iv}\mathbf{X}^\top \mathbf{z},$$

where $\widehat{\boldsymbol{\Sigma}}_{\mathbf{X}_{iv}} \triangleq \frac{1}{n}\mathbf{P}_{iv}\mathbf{X}^\top\mathbf{X}\mathbf{P}_{iv}$. Since $\widehat{\boldsymbol{\theta}}^{dac} - \boldsymbol{\theta}^* \in \text{Col}(\mathbf{S}_{iv})$, we have $\mathbb{E}_{P^t}\left[z \cdot \mathbf{x}^\top \mathbf{P}_e(\widehat{\boldsymbol{\theta}}^{dac} - \boldsymbol{\theta}^*)\right] = 0$. Therefore, let $\boldsymbol{\Sigma}_{\mathbf{x},t} \triangleq \mathbb{E}_{P^t}[\mathbf{x}\mathbf{x}^\top]$, with high probability,

$$\begin{aligned}
E_{P^s}\left[L_t(\widehat{\boldsymbol{\theta}}^{dac}) - L_t(\boldsymbol{\theta}^*)\right] &= E_{P^s}\left[\frac{1}{2}\left\|\widehat{\boldsymbol{\theta}}^{dac} - \boldsymbol{\theta}^*\right\|_{\boldsymbol{\Sigma}_{\mathbf{x},t}}^2\right] \\
&= \text{tr}\left(\frac{1}{2n}\mathbb{E}_{P^s}\left[\mathbf{z}\mathbf{z}^\top\right]\mathbb{E}_{P^s}\left[\left(\frac{1}{n}\mathbf{P}_{iv}\mathbf{X}^\top\mathbf{X}\mathbf{P}_{iv}\right)^\dagger\right]\boldsymbol{\Sigma}_{\mathbf{x},t}\right) \\
&= \text{tr}\left(\frac{\sigma^2}{2n}\mathbb{E}_{P^s}\left[\widehat{\boldsymbol{\Sigma}}_{\mathbf{X}_{iv}}^\dagger\right]\boldsymbol{\Sigma}_{\mathbf{x},t}\right) \\
&\leq C_{t,iv}\frac{\sigma^2}{2n}\,tr\left(\mathbb{E}_{P^s}\left[\widehat{\boldsymbol{\Sigma}}_{\mathbf{X}_{iv}}^\dagger\right]\right) \qquad\text{(Lemma 5, \textit{w.h.p.})} \\
&\lesssim \frac{\sigma^2}{2n}\text{tr}\left(\left(\mathbb{E}_{P^s}\left[\mathbf{P}_{iv}\mathbf{x}\mathbf{x}^\top\mathbf{P}_{iv}\right]\right)^\dagger\right) \\
&\leq \frac{\sigma^2}{2nc}\text{tr}(\mathbf{P}_{iv}) \lesssim \frac{\sigma^2 d_{iv}}{2n}.
\end{aligned}$$

∎

*Proof of Theorem 14.* Let $\widehat{\boldsymbol{\Sigma}}_{\widetilde{\mathcal{A}}(\mathbf{X})} \triangleq \frac{1}{(1+\alpha)n}\widetilde{\mathcal{A}}(\mathbf{X})^\top \widetilde{\mathcal{A}}(\mathbf{X})$. Then with $\boldsymbol{\theta}^* \in \text{span}\{\mathbf{x}_i \mid i \in [n]\}$ from Assumption 4, we have $\boldsymbol{\theta}^* = \widehat{\boldsymbol{\Sigma}}_{\widetilde{\mathcal{A}}(\mathbf{X})}^\dagger \widehat{\boldsymbol{\Sigma}}_{\widetilde{\mathcal{A}}(\mathbf{X})}\boldsymbol{\theta}^*$. Since $\boldsymbol{\theta}^* \in \text{Col}(\mathbf{S}_{iv})$, $\widetilde{\mathbf{M}}\mathbf{X}\boldsymbol{\theta}^* = \widetilde{\mathbf{M}}\mathbf{X}\mathbf{P}_{iv}\boldsymbol{\theta}^* = \widetilde{\mathcal{A}}(\mathbf{X})\boldsymbol{\theta}^*$. Then, the ERM on the augmented training set yields

$$\begin{aligned}
\widehat{\boldsymbol{\theta}}^{da-erm} - \boldsymbol{\theta}^* &= \frac{1}{(1+\alpha)n}\widehat{\boldsymbol{\Sigma}}_{\widetilde{\mathcal{A}}(\mathbf{X})}^\dagger \widetilde{\mathcal{A}}(\mathbf{X})^\top \widetilde{\mathbf{M}}(\mathbf{X}\boldsymbol{\theta}^* + \mathbf{z}) - \widehat{\boldsymbol{\Sigma}}_{\widetilde{\mathcal{A}}(\mathbf{X})}^\dagger \widehat{\boldsymbol{\Sigma}}_{\widetilde{\mathcal{A}}(\mathbf{X})}\boldsymbol{\theta}^* \\
&= \frac{1}{(1+\alpha)n}\widehat{\boldsymbol{\Sigma}}_{\widetilde{\mathcal{A}}(\mathbf{X})}^\dagger \widetilde{\mathcal{A}}(\mathbf{X})^\top \widetilde{\mathbf{M}}\mathbf{z}.
\end{aligned}$$

Meanwhile with $\mathbb{E}_{P^t}\left[z \cdot \boldsymbol{\zeta}_e\right] = \mathbf{0}$ from Assumption 3, we have $\mathbb{E}_{P^t}\left[z \cdot \mathbf{P}_e \mathbf{x}\right] = \mathbf{0}$. Therefore, by recalling that $\boldsymbol{\Sigma}_{\mathbf{x},t} \triangleq \mathbb{E}_{P^t}[\mathbf{x}\mathbf{x}^\top]$,

$$L_t(\boldsymbol{\theta}) - L_t(\boldsymbol{\theta}^*) = \mathbb{E}_{P^t(\mathbf{x})}\left[\frac{1}{2}\left(\mathbf{x}^\top(\boldsymbol{\theta} - \boldsymbol{\theta}^*)\right)^2 + z \cdot \mathbf{x}^\top \mathbf{P}_e(\boldsymbol{\theta} - \boldsymbol{\theta}^*)\right] = \frac{1}{2}\|\boldsymbol{\theta}^* - \boldsymbol{\theta}\|^2_{\boldsymbol{\Sigma}_{\mathbf{x},t}},$$

such that the expected excess risk can be expressed as

$$\mathbb{E}_{P^s}\left[L_t(\widehat{\boldsymbol{\theta}}^{da-erm}) - L_t(\boldsymbol{\theta}^*)\right] = \frac{1}{2(1+\alpha)^2 n^2}\operatorname{tr}\left(\mathbb{E}_{P^s}\left[\widehat{\boldsymbol{\Sigma}}^\dagger_{\widetilde{\mathcal{A}}(\mathbf{X})}\left(\widetilde{\mathcal{A}}(\mathbf{X})^\top \widetilde{\mathbf{M}}\mathbf{z}\mathbf{z}^\top \widetilde{\mathbf{M}}^\top \widetilde{\mathcal{A}}(\mathbf{X})\right)\widehat{\boldsymbol{\Sigma}}^\dagger_{\widetilde{\mathcal{A}}(\mathbf{X})}\right]\boldsymbol{\Sigma}_{\mathbf{x},t}\right),$$

where let $\widehat{\boldsymbol{\Sigma}}_{\widetilde{\mathcal{A}}(\mathbf{X}_e)} \triangleq \mathbf{P}_e \widehat{\boldsymbol{\Sigma}}_{\widetilde{\mathcal{A}}(\mathbf{X})}\mathbf{P}_e$,

$$\mathbb{E}_{P^s}\left[\widehat{\boldsymbol{\Sigma}}^\dagger_{\widetilde{\mathcal{A}}(\mathbf{X})}\left(\widetilde{\mathcal{A}}(\mathbf{X})^\top \widetilde{\mathbf{M}}\mathbf{z}\mathbf{z}^\top \widetilde{\mathbf{M}}^\top \widetilde{\mathcal{A}}(\mathbf{X})\right)\widehat{\boldsymbol{\Sigma}}^\dagger_{\widetilde{\mathcal{A}}(\mathbf{X})}\right]$$
$$\succcurlyeq \mathbb{E}_{P^s}\left[\left(\mathbf{P}_{iv}\widehat{\boldsymbol{\Sigma}}^\dagger_{\widetilde{\mathcal{A}}(\mathbf{X})}\mathbf{P}_{iv} + \mathbf{P}_e\widehat{\boldsymbol{\Sigma}}^\dagger_{\widetilde{\mathcal{A}}(\mathbf{X})}\mathbf{P}_e\right)\widetilde{\mathcal{A}}(\mathbf{X})^\top \widetilde{\mathbf{M}}\mathbf{z}\mathbf{z}^\top \widetilde{\mathbf{M}}^\top \widetilde{\mathcal{A}}(\mathbf{X})\left(\mathbf{P}_{iv}\widehat{\boldsymbol{\Sigma}}^\dagger_{\widetilde{\mathcal{A}}(\mathbf{X})}\mathbf{P}_{iv} + \mathbf{P}_e\widehat{\boldsymbol{\Sigma}}^\dagger_{\widetilde{\mathcal{A}}(\mathbf{X})}\mathbf{P}_e\right)\right]$$
$$\succcurlyeq \sigma^2(1+\alpha)^2 n \cdot \mathbb{E}_{P^s}\left[\widehat{\boldsymbol{\Sigma}}^\dagger_{\mathbf{X}_{iv}}\right] + \mathbb{E}_{P^s}\left[\widehat{\boldsymbol{\Sigma}}^\dagger_{\widetilde{\mathcal{A}}(\mathbf{X}_e)}\widetilde{\mathcal{A}}(\mathbf{X}_e)^\top \widetilde{\mathbf{M}}\mathbf{z}\mathbf{z}^\top \widetilde{\mathbf{M}}^\top \widetilde{\mathcal{A}}(\mathbf{X}_e)\widehat{\boldsymbol{\Sigma}}^\dagger_{\widetilde{\mathcal{A}}(\mathbf{X}_e)}\right].$$

We denote

$$\mathrm{EER}_e \triangleq \operatorname{tr}\left(\mathbb{E}_{P^s}\left[\frac{1}{2(1+\alpha)^2 n^2}\widehat{\boldsymbol{\Sigma}}^\dagger_{\widetilde{\mathcal{A}}(\mathbf{X}_e)}\widetilde{\mathcal{A}}(\mathbf{X}_e)^\top \widetilde{\mathbf{M}}\mathbf{z}\mathbf{z}^\top \widetilde{\mathbf{M}}^\top \widetilde{\mathcal{A}}(\mathbf{X}_e)\widehat{\boldsymbol{\Sigma}}^\dagger_{\widetilde{\mathcal{A}}(\mathbf{X}_e)}\right]\right),$$

and observe that

$$\mathrm{EER}_e = \mathbb{E}_{P^s}\left[\frac{1}{2}\left\|\frac{1}{(1+\alpha)n}\widehat{\boldsymbol{\Sigma}}^\dagger_{\widetilde{\mathcal{A}}(\mathbf{X}_e)}\widetilde{\mathcal{A}}(\mathbf{X}_e)^\top \widetilde{\mathbf{M}}\mathbf{z}\right\|^2_2\right] > 0.$$

Finally, we complete the proof by partitioning the lower bound for the target expected excess risk of $\widehat{\boldsymbol{\theta}}^{da-erm}$ into the invariantand environmental parts such that

$$\mathbb{E}_{P^s}\left[L_t(\widehat{\boldsymbol{\theta}}^{da-erm}) - L_t(\boldsymbol{\theta}^*)\right]$$
$$\geq \underbrace{\operatorname{tr}\left(\frac{\sigma^2}{2n}\mathbb{E}_{P^s}\left[\widehat{\boldsymbol{\Sigma}}^\dagger_{\mathbf{X}_{iv}}\right]\boldsymbol{\Sigma}_{\mathbf{x},t}\right)}_{=\mathbb{E}\left[L_t(\widehat{\boldsymbol{\theta}}^{dac}) - L_t(\boldsymbol{\theta}^*)\right]} + \underbrace{\operatorname{tr}\left(\mathbb{E}_{P^s}\left[\frac{1}{2(1+\alpha)^2 n^2}\widehat{\boldsymbol{\Sigma}}^\dagger_{\widetilde{\mathcal{A}}(\mathbf{X}_e)}\widetilde{\mathcal{A}}(\mathbf{X}_e)^\top \widetilde{\mathbf{M}}\mathbf{z}\mathbf{z}^\top \widetilde{\mathbf{M}}^\top \widetilde{\mathcal{A}}(\mathbf{X}_e)\widehat{\boldsymbol{\Sigma}}^\dagger_{\widetilde{\mathcal{A}}(\mathbf{X}_e)}\right]\boldsymbol{\Sigma}_{\mathbf{x},t}\right)}_{\text{expected excess risk from environmental feature subspace} \geq c_{t,e}\cdot \mathrm{EER}_e}$$
$$\geq \mathbb{E}_{P^s}\left[L_t(\widehat{\boldsymbol{\theta}}^{dac}) - L_t(\boldsymbol{\theta}^*)\right] + c_{t,e} \cdot \mathrm{EER}_e.$$

∎

Now we construct a specific domain adaptation example with a large separation (*i.e.*, proportional to $d_e$) in the target excess risk between learning with the DAC regularization (*i.e.*, $\widehat{\boldsymbol{\theta}}^{dac}$) and with the ERM on augmented training set (*i.e.*, $\widehat{\boldsymbol{\theta}}^{da-erm}$).

**Example 2.** *We consider $P^s$ and $P^t$ that follow the same set of relations in Figure 4, except for the distributions over $\mathbf{e}$ where $P^s(\mathbf{e}) \neq P^t(\mathbf{e})$. Precisely, let the environmental feature $\boldsymbol{\zeta}_e$ depend on $(\boldsymbol{\zeta}_{iv}, y, \mathbf{e})$:*

$$\boldsymbol{\zeta}_e = \operatorname{sign}\left(y - (\boldsymbol{\theta}^*)^\top \mathbf{S}_{iv}\boldsymbol{\zeta}_{iv}\right)\mathbf{e} = \operatorname{sign}(z)\mathbf{e}, \quad z \sim \mathcal{N}(0, \sigma^2), \quad z \perp \mathbf{e},$$

*where $\mathbf{e} \sim \mathcal{N}(\mathbf{0}, \mathbf{I}_{d_e})$ for $P^s(\mathbf{e})$ and $\mathbf{e} \sim \mathcal{N}(\mathbf{0}, \sigma_t \mathbf{I}_{d_e})$ for $P^t(\mathbf{e})$, $\sigma_t \geq c_{t,e}$ (recall $c_{t,e}$ from Assumption 3). Assume that the training set $\mathbf{X}$ is sufficiently large, $n \gg d_e + \log(1/\delta)$ for some given $\delta \in (0, 1)$. Augmenting $\mathbf{X}$ with a simple by common type of data augmentations – the linear transforms, we let*

$$\widetilde{\mathcal{A}}(\mathbf{X}) = [\mathbf{X}; (\mathbf{X}\mathbf{A}_1); \ldots; (\mathbf{X}\mathbf{A}_\alpha)], \quad \mathbf{A}_j = \mathbf{P}_{iv} + \mathbf{u}_j \mathbf{v}_j^\top, \quad \mathbf{u}_j, \mathbf{v}_j \in \operatorname{Col}(\mathbf{S}_e) \quad \forall j \in [\alpha],$$

*and define*

$$\nu_1 \triangleq \max\{1\} \cup \{\sigma_{\max}(\mathbf{A}_j) \mid j \in [\alpha]\} \quad and \quad \nu_2 \triangleq \sigma_{\min}\left(\frac{1}{1+\alpha}\left(\mathbf{I}_d + \sum_{j=1}^{\alpha}\mathbf{A}_k\right)\right),$$

where $\sigma_{\min}(\cdot)$ and $\sigma_{\max}(\cdot)$ refer to the minimum and maximum singular values, respectively. Then under Assumption 3 and Assumption 4, with constant probability,

$$\mathbb{E}_{P^s}\left[L_t(\widehat{\boldsymbol{\theta}}^{da-erm}) - L_t(\boldsymbol{\theta}^*)\right] \gtrsim \mathbb{E}_{P^s}\left[L_t(\widehat{\boldsymbol{\theta}}^{dac}) - L_t(\boldsymbol{\theta}^*)\right] + c_{t,e} \cdot \frac{\sigma^2 d_e}{2n}.$$

*Proof of Example 2.* With the specified distribution, for $\mathbf{E} = [\mathbf{e}_1; \ldots; \mathbf{e}_n] \in \mathbb{R}^{n \times d_e}$,

$$\widehat{\boldsymbol{\Sigma}}_{\widetilde{\mathcal{A}}(\mathbf{X}_e)} = \frac{1}{(1+\alpha)n}\mathbf{S}_e\left(\mathbf{E}^\top\mathbf{E} + \sum_{j=1}^\alpha \mathbf{A}_j^\top\mathbf{E}^\top\mathbf{E}\mathbf{A}_j\right)\mathbf{S}_e^\top \preccurlyeq \frac{\nu_1^2}{n}\mathbf{S}_e\mathbf{E}^\top\mathbf{E}\mathbf{S}_e^\top,$$

$$\frac{1}{(1+\alpha)n}\widetilde{\mathcal{A}}(\mathbf{X}_e)^\top\widetilde{\mathbf{M}}\mathbf{z} = \left(\frac{1}{1+\alpha}\left(\mathbf{I}_d + \sum_{j=1}^\alpha \mathbf{A}_j\right)\right)^\top \frac{1}{n}\mathbf{S}_e\mathbf{E}^\top |\mathbf{z}|.$$

By Lemma 5, under Assumption 3 and Assumption 4, we have that with high probability, $0.9\mathbf{I}_{d_e} \preccurlyeq \frac{1}{n}\mathbf{E}^\top\mathbf{E} \preccurlyeq 1.1\mathbf{I}_{d_e}$. Therefore with $\mathbf{E}$ and $\mathbf{z}$ being independent,

$$\begin{aligned}
\mathrm{EER}_e &= \mathbb{E}_{P^s}\left[\frac{1}{2}\left\|\frac{1}{(1+\alpha)n}\widehat{\boldsymbol{\Sigma}}_{\widetilde{\mathcal{A}}(\mathbf{X}_e)}^\dagger\widetilde{\mathcal{A}}(\mathbf{X}_e)^\top\widetilde{\mathbf{M}}\mathbf{z}\right\|_2^2\right]\\
&\geq \frac{\sigma^2}{2n}\frac{\nu_2^2}{\nu_1^4}\ \mathrm{tr}\left(\mathbb{E}_{P^s}\left[\left(\frac{1}{n}\mathbf{S}_e\mathbf{E}^\top\mathbf{E}\mathbf{S}_e^\top\right)^\dagger\right]\right)\\
&\gtrsim \frac{\sigma^2}{2n}\frac{\nu_2^2}{\nu_1^4}\ \mathrm{tr}\left(\mathbf{S}_e\mathbf{S}_e^\top\right)\\
&\gtrsim \frac{\sigma^2 d_e}{2n},
\end{aligned}$$

and the rest follows from Theorem 14. ■

## C  Technical Lemmas

**Lemma 5.** *We consider a random vector $\mathbf{x} \in \mathbb{R}^d$ with $\mathbb{E}[\mathbf{x}] = \mathbf{0}$, $\mathbb{E}[\mathbf{x}\mathbf{x}^\top] = \boldsymbol{\Sigma}$, and $\overline{\mathbf{x}} = \boldsymbol{\Sigma}^{-1/2}\mathbf{x}$ [3] being $\rho^2$-subgaussian. Given an i.i.d. sample of $\mathbf{x}$, $\mathbf{X} = [\mathbf{x}_1, \ldots, \mathbf{x}_n]^\top$, for any $\delta \in (0,1)$, if $n \gg \rho^4 d$, then $0.9\boldsymbol{\Sigma} \preccurlyeq \frac{1}{n}\mathbf{X}^\top\mathbf{X} \preccurlyeq 1.1\boldsymbol{\Sigma}$ with probability high probability.*

*Proof.* We first denote $\mathbf{P}_{\mathcal{X}} \triangleq \boldsymbol{\Sigma}\boldsymbol{\Sigma}^\dagger$ as the orthogonal projector onto the subspace $\mathcal{X} \subseteq \mathbb{R}^d$ supported by the distribution of $\mathbf{x}$. With the assumptions $\mathbb{E}[\mathbf{x}] = \mathbf{0}$ and $\mathbb{E}[\mathbf{x}\mathbf{x}^\top] = \boldsymbol{\Sigma}$, we observe that $\mathbb{E}[\overline{\mathbf{x}}] = \mathbf{0}$ and $\mathbb{E}[\overline{\mathbf{x}}\overline{\mathbf{x}}^\top] = \mathbb{E}[\mathbf{x}\boldsymbol{\Sigma}^{-1}\mathbf{x}^\top] = \mathbf{P}_{\mathcal{X}}$. Given the sample set $\mathbf{X}$ of size $n \gg \rho^4(d + \log(1/\delta))$ for some $\delta \in (0,1)$, we let $\mathbf{U} = \frac{1}{n}\sum_{i=1}^n \mathbf{x}_i\boldsymbol{\Sigma}^{-1}\mathbf{x}_i^\top - \mathbf{P}_{\mathcal{X}}$. Then the problem can be reduced to showing that, with probability at least $1 - \delta$, $\|\mathbf{U}\|_2 \leq 0.1$. For this, we leverage the $\epsilon$-net argument as following.

For an arbitrary $\mathbf{v} \in \mathcal{X} \cap \mathbb{S}^{d-1}$, we have

$$\mathbf{v}^\top\mathbf{U}\mathbf{v} = \frac{1}{n}\sum_{i=1}^n\left(\mathbf{v}^\top\mathbf{x}_i\boldsymbol{\Sigma}^{-1}\mathbf{x}_i^\top\mathbf{v} - 1\right) = \frac{1}{n}\sum_{i=1}^n\left(\left(\mathbf{v}^\top\overline{\mathbf{x}}_i\right)^2 - 1\right),$$

where, given $\overline{\mathbf{x}}_i$ being $\rho^2$-subgaussian, $\mathbf{v}^\top\overline{\mathbf{x}}_i$ is $\rho^2$-subgaussian. Since

$$\mathbb{E}\left[\left(\mathbf{v}^\top\overline{\mathbf{x}}_i\right)^2\right] = \mathbf{v}^\top\mathbb{E}\left[\overline{\mathbf{x}}_i\overline{\mathbf{x}}_i^\top\right]\mathbf{v} = 1,$$

we know that $\left(\mathbf{v}^\top\overline{\mathbf{x}}_i\right)^2 - 1$ is $16\rho^2$-subexponential. Then, we recall the Bernstein's inequality,

$$\mathbb{P}\left[\left|\mathbf{v}^\top\mathbf{U}\mathbf{v}\right| > \epsilon\right] \leq 2\exp\left(-\frac{n}{2}\min\left(\frac{\epsilon^2}{(16\rho^2)^2}, \frac{\epsilon}{16\rho^2}\right)\right).$$

---

[3]In the case where $\boldsymbol{\Sigma}$ is rank-deficient, we slightly abuse the notation such that $\boldsymbol{\Sigma}^{-1/2}$ and $\boldsymbol{\Sigma}^{-1}$ refer to the respective pseudo-inverses.

Let $N \subset \mathcal{X} \cap \mathbb{S}^{d-1}$ be an $\epsilon_1$-net such that $|N| = e^{O(d)}$. Then for some $0 < \epsilon_2 \leq 16\rho^2$, by the union bound,

$$\mathbb{P}\left[\max_{\mathbf{v} \in N} : \left|\mathbf{v}^\top \mathbf{U}\mathbf{v}\right| > \epsilon_2\right] \leq 2|N| \exp\left(-\frac{n}{2} \min\left(\frac{\epsilon_2^2}{(16\rho^2)^2}, \frac{\epsilon_2}{16\rho^2}\right)\right)$$

$$\leq \exp\left(O(d) - \frac{n}{2} \cdot \frac{\epsilon_2^2}{(16\rho^2)^2}\right) \leq \delta$$

whenever $n > \frac{2(16\rho^2)^2}{\epsilon_2^2}\left(\Theta(d) + \log\frac{1}{\delta}\right)$. By taking $\delta = \exp\left(-\frac{1}{4}\left(\frac{\epsilon_2}{16\rho^2}\right)^2 n\right)$, we have that $\max_{\mathbf{v} \in N}\left|\mathbf{v}^\top \mathbf{U}\mathbf{v}\right| \leq \epsilon_2$ with high probability when $n > 4\left(\frac{16\rho^2}{\epsilon_2}\right)^2 \Theta(d)$, and taking $n \gg \rho^4 d$ is sufficient.

Now for any $\mathbf{v} \in \mathcal{X} \cap \mathbb{S}^{d-1}$, there exists some $\mathbf{v}' \in N$ such that $\|\mathbf{v} - \mathbf{v}'\|_2 \leq \epsilon_1$. Therefore,

$$\left|\mathbf{v}^\top \mathbf{U}\mathbf{v}\right| = \left|\mathbf{v}'^\top \mathbf{U}\mathbf{v}' + 2\mathbf{v}'^\top \mathbf{U}(\mathbf{v} - \mathbf{v}') + (\mathbf{v} - \mathbf{v}')^\top \mathbf{U}(\mathbf{v} - \mathbf{v}')\right|$$

$$\leq \left(\max_{\mathbf{v} \in N} : \left|\mathbf{v}^\top \mathbf{U}\mathbf{v}\right|\right) + 2\|\mathbf{U}\|_2 \|\mathbf{v}'\|_2 \|\mathbf{v} - \mathbf{v}'\|_2 + \|\mathbf{U}\|_2 \|\mathbf{v} - \mathbf{v}'\|_2^2$$

$$\leq \left(\max_{\mathbf{v} \in N} : \left|\mathbf{v}^\top \mathbf{U}\mathbf{v}\right|\right) + \|\mathbf{U}\|_2\left(2\epsilon_1 + \epsilon_1^2\right).$$

Taking the supremum over $\mathbf{v} \in \mathbb{S}^{d-1}$, with probability at least $1 - \delta$,

$$\max_{\mathbf{v} \in \mathcal{X} \cap \mathbb{S}^{d-1}} : \left|\mathbf{v}^\top \mathbf{U}\mathbf{v}\right| = \|\mathbf{U}\|_2 \leq \epsilon_2 + \|\mathbf{U}\|_2\left(2\epsilon_1 + \epsilon_1^2\right), \qquad \|\mathbf{U}\|_2 \leq \frac{\epsilon_2}{2 - (1 + \epsilon_1)^2}.$$

With $\epsilon_1 = \frac{1}{3}$ and $\epsilon_2 = \frac{1}{45}$, we have $\frac{\epsilon_2}{2 - (1 + \epsilon_1)^2} = \frac{1}{10}$.

Overall, if $n \gg \rho^4 d$, then with high probability, we have $\|\mathbf{U}\|_2 \leq 0.1$. ∎

**Lemma 6.** *Let $U \subseteq \mathbb{R}^d$ be an arbitrary non-trivial subspace in $\mathbb{R}^d$, and $\mathbf{g} \sim \mathcal{N}(\mathbf{0}, \mathbf{I}_d)$ be a Gaussian random vector. Then for any continuous and $C_l$-Lipschitz function $\varphi : \mathbb{R} \to \mathbb{R}$ (i.e., $|\varphi(u) - \varphi(u')| \leq C_l \cdot |u - u'|$ for all $u, u' \in \mathbb{R}$),*

$$\mathbb{E}_{\mathbf{g}}\left[\sup_{\mathbf{u} \in U} \mathbf{g}^\top \varphi(\mathbf{u})\right] \leq C_l \cdot \mathbb{E}_{\mathbf{g}}\left[\sup_{\mathbf{u} \in U} \mathbf{g}^\top \mathbf{u}\right],$$

*where $\varphi$ acts on $\mathbf{u}$ entry-wisely, $(\varphi(\mathbf{u}))_j = \varphi(u_j)$. In other words, the Gaussian width of the image set $\varphi(U) \triangleq \{\varphi(\mathbf{u}) \in \mathbb{R}^d \mid \mathbf{u} \in U\}$ is upper bounded by that of $U$ scaled by the Lipschitz constant.*

*Proof.*

$$
\begin{aligned}
\mathbb{E}_{\mathbf{g}}\left[\sup_{\mathbf{u}\in U}\mathbf{g}^\top\varphi(\mathbf{u})\right] =& \frac{1}{2}\mathbb{E}_{\mathbf{g}}\left[\sup_{\mathbf{u}\in U}\mathbf{g}^\top\varphi(\mathbf{u}) + \sup_{\mathbf{u}'\in U}\mathbf{g}^\top\varphi(\mathbf{u})\right]\\
=& \frac{1}{2}\mathbb{E}_{\mathbf{g}}\left[\sup_{\mathbf{u},\mathbf{u}'\in U}\mathbf{g}^\top\left(\varphi(\mathbf{u}) - \varphi(\mathbf{u}')\right)\right]\\
\leq& \frac{1}{2}\mathbb{E}_{\mathbf{g}}\left[\sup_{\mathbf{u},\mathbf{u}'\in U}\sum_{j=1}^{d}|g_j|\left|\varphi(u_j) - \varphi(u_j')\right|\right] \quad \because \varphi \text{ is } C_l\text{-Lipschitz}\\
\leq& \frac{C_l}{2}\mathbb{E}_{\mathbf{g}}\left[\sup_{\mathbf{u},\mathbf{u}'\in U}\sum_{j=1}^{d}|g_j|\left|u_j - u_j'\right|\right]\\
=& \frac{C_l}{2}\mathbb{E}_{\mathbf{g}}\left[\sup_{\mathbf{u},\mathbf{u}'\in U}\mathbf{g}^\top\left(\mathbf{u} - \mathbf{u}'\right)\right]\\
=& \frac{C_l}{2}\mathbb{E}_{\mathbf{g}}\left[\sup_{\mathbf{u}\in U}\mathbf{g}^\top\mathbf{u} + \sup_{\mathbf{u}'\in U}\mathbf{g}^\top\left(-\mathbf{u}'\right)\right]\\
=& C_l\cdot\mathbb{E}_{\mathbf{g}}\left[\sup_{\mathbf{u}\in U}\mathbf{g}^\top\mathbf{u}\right]
\end{aligned}
$$

∎

## D  EXPERIMENT DETAILS

In this section, we provide the details of our experiments. Our code is adapted from the publicly released repo: https://github.com/kekmodel/FixMatch-pytorch.

**Dataset:** Our training dataset is derived from CIFAR-100, where the original dataset contains 50,000 training samples of 100 different classes. Out of the original 50,000 samples, we randomly select 10,000 labeled data as training set (i.e., 100 labeled samples per class). To see the impact of different training samples, we also trained our model with dataset that contains 1,000 and 20,000 samples. Evaluations are done on standard test set of CIFAR-100, which contains 10,000 testing samples.

**Data Augmentation:** During the training time, given a training batch, we generate corresponding augmented samples by RandAugment (Cubuk et al., 2020). We set the number of augmentations per sample to 7, unless otherwise mentioned.

To generate an augmented image, the RandAugment draws $n$ transformations uniformaly at random from 14 different augmentations, namely {identity, autoContrast, equalize, rotate, solarize, color, posterize, contrast, brightness, sharpness, shear-x, shear-y, translate-x, translate-y}. The RandAugment provides each transformation with a single scalar (1 to 10) to control the strength of each of them, which we always set to 10 for all transformations. By default, we set $n = 2$ (i.e., using 2 random transformations to generate an augmented sample). To see the impact of different augmentation strength, we choose $n \in \{1, 2, 5, 10\}$. Examples of augmented samples are shown in Figure 3.

**Parameter Setting:** The batch size is set to 64 and the entire training process takes $2^{15}$ steps. During the training, we adopt the SGD optimizer with momentum set to 0.9, with learning rate for step $i$ being $0.03 \times \cos\left(\frac{i \times 7\pi}{2^{15} \times 16}\right)$.

**Additional Settings for the semi-supervised learning results:** For the results on semi-supervised learning, besides the 10,000 labeled samples, we also draw additionally samples (ranging from 5,000 to 20,000) from the training set of the original CIFAR-100. We remove the labels of those additionally sampled images, as they serve as "unlabeled" samples in the semi-supervised learning setting. The FixMatch implementation follows the publicly available on in https://github.com/kekmodel/FixMatch-pytorch.

## E  ILLUSTRATIVE EXAMPLE

Here we present an illustrative example for logistic regression, which numerically shows the benefits of DAC over ERM, and also clearly demonstrates the impact of different $d_{aug}$ and $\alpha$.

**Example 3.** *Consider a 30-dimensional logistic regression. The original training set contains 50 samples. The inputs $\mathbf{x}_i$s are generated independently from $\mathcal{N}(0, \mathbf{I}_{30})$ and we set $\theta^* = [\theta_c^*; \mathbf{0}]$ with $\theta_c^* \sim \mathcal{N}(0, \mathbf{I}_3)$ and $\mathbf{0} \in \mathbb{R}^{27}$.*

*To generate the augmentations, we first specify a parameter $d_{aug}$ and leave the first $30 - d_{aug}$ elements of each $\mathbf{x}_i$ unchanged, and replace the later $d_{aug}$ elements of each $\mathbf{x}_i$ with a new vector randomly generated from $\mathcal{N}(0, \mathbf{I}_{d_{aug}})$. Further, we generate $\alpha$ augmentations for each of the $\mathbf{x}_i$. For any $\alpha \geq 1$, the augmentation will perturb $d_{aug}$ coordinates with probability 1.*

*The results for $d_{aug} \in \{20, 25\}$ and various $\alpha$s are presented in Figure 5. The test error clearly matches Theorem 3 and the discussion afterward: 1) The generalization performance of DAC only relies on $d_{aug}$ but not $\alpha$, and larger $d_{aug}$ gives smaller testing error. 2) The performance of DA-ERM crucially depends on $\alpha$, when $\alpha$ is small (i.e., limited augmentations), the performance between DAC and DA-ERM is very significant. And when we further increases $\alpha$, DA-ERM can only approach to DAC but not out-perform DAC.*

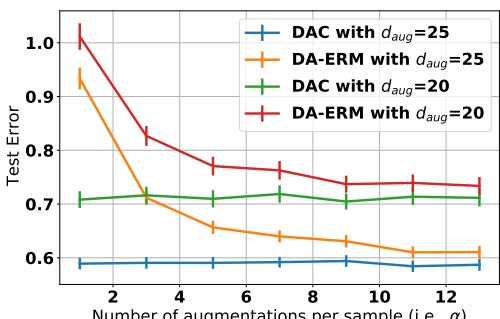

Figure 5: Comparison of DAC regularization and DA-ERM for logistic regression (Example 3). The results precisely match Theorem 3 and the discussion afterward. As suggested by the theoretical analysis, the performance of DAC only depends on the $d_{aug}$, but not $\alpha$. Further, given the same augmented dataset, DA-ERM is always worse than DAC. The gap is particularly significant when the number of augmentations is small (i.e., a small $\alpha$).

