# OpenReview forum: "Theoretical Analysis of Consistency Regularization with Limited Augmented Data"
_ICLR.cc/2022/Conference — ICLR 2022 Submitted_

### Official Review · Reviewer_TPuR · 2021-11-01

**Correctness:** 3
**Technical Novelty And Significance:** 2
**Empirical Novelty And Significance:** 2
**Recommendation:** 5
**Confidence:** 4

**Main Review:**

- What would be the difference between data augmentation and consistency regularization? This is unclear from the introduction and it serves as a motivation for this work. Previous work has demonstrated that regularized risk minimization problems with augmented training samples (the setting known as vicinal risk minimization) can have identical effect as "consistency regularization", i.e., low variance between predictions over original and augmented samples. Please check out the work on vicinal risk minimization and references further developing that line of work.
[1] O Chapelle, J Weston, L Bottou, V Vapnik (NIPS). Vicinal risk minimization.

- Definition 1: This definition is overly restrictive. For some instances it is natural to have the conditional distribution with larger entropy and yet the instance is a proper augmentation. There are definitely transformations of images that are more confusing relative to labels of interest, when compared to the notion of original instance. This assumption is, thus, not motivated by practical considerations and might be an artefact required for theoretical analysis.
Here, it is also important to properly introduce "original", because in different worlds pairs (x, y) sampled from a data generating distribution are different. This would then imply that the notion of original is not unique.

- Definition 2: It is common actually that the data lies on a low-dimensional manifold and augmentation is in a number of cases just a walk over that manifold. Thus, I see no reason why this metric would be of any use in assessing the informativeness of data augmentation. Also, the difference is the perturbation over the input space which might have nothing to do with the manifold that we would like to learn and over which the data generating distribution actually operates.

- Consistency regularization: What is so special about Eq. (1) in the context of prior work (not discussed nor covered in details)? Is this not equivalent to learning with empirical risk minimization and augmented samples (known as vicinal risk minimization for 20+ years)? For the latter, it can be demonstrated theoretically that it minimizes the variance over the neighborhood in the instance space defined by augmented samples. I can see that Eq. (1) might provide more flexibility when it comes to the enforcement of label alignment over such neighborhoods but that needs to be demonstrated. The vicinal setting is also quite flexible.

- The theoretical results are under restrictive assumptions and in my understanding apply only to linear models. Theorem 4 is an attempt to obtain a theoretical result on the effectiveness for two layer neural network, but the assumption $(x_i^{\top}B)_{+} = (x_{i,j}^{\top}B)_{+}$ actually restricts this to linear models as well.

- Experiments: The performance improvements are rather modest, 1% relative on CIFAR-100. In the training setting, were the batches ordered in the same way? The problem is non-convex and that might impact the generalization performance.
- Table 2: The experiment with different number of samples 1,000 - 20,000: Why only 3 augmented samples per training instance? In such cases, augmentation typically adds many more samples and it would be nice to see the results with 10+ additional samples per training instance, as in the case for the experiment in Table 1.
- Table 3: How can the performance be consistently the same with 3, 7, and 15 augmented samples? This might be an indication that the augmentation scheme does not do enough to describe the local neighborhoods around training samples. This, in turn, might then not be an objective assessment of the effectiveness.
Table 4: The same argument as for Table 3, one would expect much more differentiation between numbers for "strength" 2, 5, and 10. In fact, this experiment increases my scepticism in the utility of the $d_{aug}$ as a measure of augmentation informativeness.

**Summary Of The Paper:**

The paper proposes a regularization approach based on data augmentation and develops learning bounds for that setting. The main motivation is to provide means to characterize reduction in sample complexity as a result of employing data augmentation techniques during training.

**Summary Of The Review:**

The assumptions under which the paper operates are quite restrictive and do not see that they hold in practice, and thus apply to widely used data augmentation schemes along with deep learning models. The crux of the theoretical contributions is restricted to linear models, which further limits their scope. The paper does not also cover well the prior work on empirical risk minimization problems, especially the line of work on vicinal risk minimization. Experiments show rather incremental improvement (1% relative) and it is quite strange that there is no substantially larger effect of augmentation when the replication factor goes from 3 to 15.

---

> ### Author Response · Authors · 2021-11-17
> **Response to Reviewer TPuR - Detailed Responses**
>
> *[1. VRM v.s. DAC]*
>
> We hope to kindly emphasize that our comparison is NOT between data augmentation (DA) and consistency regularization (DAC), but to show that DAC is a better way to utilize the augmented samples than just doing empirical risk minimization (ERM) on the augmented dataset.
>
> The vicinal risk minimization (VRM) is more similar to ERM on the augmented dataset (if we view the neighbor of original samples as augmentations). We agree with the reviewer that VRM and ERM both implicitly reduce the variance of predictions on the original sample and augmented ones. On the other hand, DAC explicitly penalizes the prediction discrepancy between those predictions, and our main contribution is theoretically showing that DAC is statistically more efficient because of this explicit consistency regularization.
>
> Therefore, the specialty of Equation (1) is the regularization term “DAC regularizer”, which penalizes the prediction discrepancy explicitly; while ERM on the augmented training set (Eq(3), or VRM) merges the augmented data into the training set. As a further illustration, Example 1 and Figure 2 show the difference of ERM with augmented samples and DAC when given the same augmented dataset - depending on the augmentation transformation, ERM with augmented samples (or VRM in this form) can be significantly worse than DAC.
>
> *[2. Data augmentation and strength]*
>
> This is a well-established assumption in most related work studying data augmentation (see e.g. [1][2][3]). Intuitively, the observation is considered to be a mixture of content information that infers the label and environmental information that will be modified by data augmentation. Further, when the original sample and augmented one have the same label distribution, there is no need to have a unique notion of which one is the “original” sample.
>
> We agree with the reviewer that we can view the data as lying in a low-dimensional manifold. When the manifold is contained in a lower-dimensional subspace $S_M$, then the $d_{aug}$ can be intuitively understood as the dimension of the subspace orthogonal to $S_M$ that is perturbed by augmentations. Therefore $d_{aug}$ shows how many “irrelevant” dimensions can be eliminated by the augmentation, and it is, therefore, a measure of augmentation strength. Besides $d_{aug}$, we also present a generalization in Section 4.3, where we have a different notion of augmentation strength.
>
> *[3. Limitations to linear models]*
>
> We do not require either the model or the data augmentation map to be linear. Concerning Theorem 4, although we characterize the benefits of DAC with a linear reduction, both the data augmentation and the model (i.e., two-layer ReLU network) can be non-linear. From the representation learning viewpoint, the first layer learns a nonlinear representation of the data with consistency regularization. Therefore we do not see our results being limited to linear models. Furthermore, in Section 4.3, for classification problems, we unify the existing theories and provide even more general notions for both the data augmentations and the model.
>
> *[4. Clarification on Experiments]*
>
> We would kindly emphasize that the experiments are to present apples-to-apples comparisons between ERM and DAC. For all experiments for DAC and ERM, we kept the random seed the same and repeated over 5 different seeds to report the average and standard deviation. The 1% improvement on CIFAR-100 is statistically significant and warrants the benefits of DAC.
>
> Clarification on using limited augmented samples: one of the focuses of our work is to show the benefits of DAC with limited augmentations. The impact of different numbers of augmentations is studied separately in Table 3.
>
> For the question in Table 3: Notice that the performance of ERM indeed increases as it has more augmented samples. This matches the intuition in the review. On the other hand, the performance of DAC is more stable when it sees less augmented samples, as the regularization is still effectively regulating the model to generate consistent predictions for the original sample and augmented ones. This “counterintuitive” observation verifies our theoretical results that DAC is more statistically efficient (i.e., requires fewer samples to learn).
>
> For the question in Table 4: for DAC, we see a decreasing trend of performance as we use more augmentation transformation. It shows that when too many transformations are applied to an image, it will eventually change the label distribution (as suggested in Figure 3). Therefore, the decreasing trend of DAC in Table 4 is due to the violation of Definition 1 (i.e., augmented samples do not have the same label distribution as the original one) and does not correspond to the change of $d_{aug}$.
>
> [1] Self-Supervised Learning with Data Augmentations Provably Isolates Content from Style
>
> [2] Learning with invariances in random features and kernel models
>
> [3] A Group-Theoretic Framework for Data Augmentation

---

> ### Author Response · Authors · 2021-11-17
> **Response to Reviewer TPuR - Highlight for the Summary of the Review**
>
> We appreciate the reviewer’s questions and comments on both the theory and experiments. We would like to start our response by kindly highlighting several points which hopefully can address the concerns in the summary of the review:
>
> 1. The main objective of our work is to demonstrate that DAC is a better way to leverage data augmentations than applying the ERM directly on the augmented training set, in lieu of studying variations of the ERM. In fact, the VRM is more related to the ERM with augmented samples, instead of the consistency regularization that we are focusing on. (Response 1 in the detailed responses)
> 2. Neither data augmentations nor models are required to be linear in our general framework. Moreover, when concretizing our general framework in Section 4, we discuss applications of our framework to nonlinear models, including two-layer neural networks and general classifiers, with non-parametric notions of data augmentations such as the expansion-based data augmentations. (Response 3 in the detailed responses)
> 3. The goal of our experiments is to present apples-to-apples comparisons between ERM and DAC, instead of leveraging other techniques for further improvements in the final prediction accuracy. (Response 4 in the detailed responses)
>
> We sincerely hope that the reviewer will find these explanations helpful, and will reevaluate our work based on these clarifications.

---

### Official Review · Reviewer_fxf3 · 2021-11-01

**Correctness:** 2
**Technical Novelty And Significance:** 3
**Empirical Novelty And Significance:** 2
**Recommendation:** 5
**Confidence:** 4

**Main Review:**

Data augmentation (DA) is a common technique to improve generalization, especially when data is scarce. This paper introduces a theoretical framework for analyzing the effectiveness of consistency regularization when data augmentation is employed. In the limit, consistency regularization is akin to solving a constrained optimization problem with consistency constraints.
- While data augmentation is used for better generalization with less data, it is also used to suppress spurious features and generalize to unseen distributions. A discussion around that is warranted. Also, the theory unfortunately is not applicable to this other major usecase of DA.
- While this paper introduces a regularizer $\rho$ in Eq. (1), the exact functional form of the regularizer is of no use as long as it is a proper divergence between the original samples and the augmented samples. However, this is practically not true as the functional form of the regularizer can be make or break in obtaining results. This is a shortcoming of the analysis in this paper, which needs to be clearly reflected.
- Eq. (2) can be obtained as a limit of Eq. (1) when $\lambda \to \infty$ only if $\rho$ satisfies certain regularity conditions (e.g., being a divergence). No such conditions are stated in the paper, so the deduction is wrong as stated.
- The mathematical model for DA that is described in Definition 1 is limited in scope. For example, it cannot capture some of the motivating works that the paper cites, such as mixup. This needs to be clearly stated.
- The operational meaning of the DA strength in Definition 2 is unclear beyond the linear examples that are provided, as this does not capture any nonlinear relationship between features.
- As far as I followed, from a theoretical perspective, the data augmentation in this paper $\widetilde{\mathcal{A}}(X)$ is always linear for deriving the main Theorems (2-4).
- It is not clear how Definition 2 does or can capture randomness in training examples or the augmentation function as $d_\text{aug}$ itself becomes a random variable.
- Another important relationship that is missing is the relationship between $d_\text{aug}$ and $\alpha$ (which is the number of times data is augmented). Can you at least say what happens when $\alpha \to \infty$?
- One major issue that has been ignored is that $\widehat{h}^{\text{dac}}$ as defined in Eq. (2) is not directly solvable through the empirical risk minimization framework. Even worse, it is unclear whether there is any way to devise a stochastic solver for it. Hence, the theoretical study of the paper is only applicable to a solver that is not practically existing.
- In Theorem 1, it is unclear how one would be able to relate the Rademacher complexity of $T^{\text{dac}}_{\widetilde{\mathcal{A}}, X}$ to $\alpha$ and $d_{\text{aug}}$. Henceforth, it is not clear how to interpret the improvement beyond **there is some improvement associated with dac compared to vanilla DA,** which we already knew.
- In Theorem 1, what is $C_l$?
- In Theorem 1 and its proof, the data augmentation seem to have been assumed to be a fixed linear transformation. That should be clearly communicated. In particular, in the second line of the proof on Page 13 in Appendix A.1, there is no expectation with respect to the randomness associated with DA. Even if the DA function is deterministic, the set $T^{\text{dac}}_{\widetilde{\mathcal{A}}, X}$ is random so I don’t know how to interpret the inequality.


The paper theoretically studies this limit for linear regression, logistic regression, and a two-layer perceptron with ReLU activation, and tries to characterize the benefits of consistency regularization beyond that of vanilla data augmentation.
- Please change the notation of $\widehat{h}^{erm}$ in Eq. (3). I suggest calling this DA-ERM rather than ERM not to confuse with ERM w/o/ augmentation.
- The benefit of the consistency regularizer in LR is characterized in Theorem 2 through a quantity called d’. It is not clear to me how to interpret that even after reading Remark 2 and Example 1. Can you please provide a more intuitive explanation?

- Can you please better explain Figure 2? Is the x-axis d’? How is it calculated?

- Why is the bound in Theorem 3 presented in this obscure form rather than the more commonly used high probability bound?
- Wouldn’t data augmentation without consistency regularization also admit the same form as Theorem 3? If so, then how does Theorem 3 characterize benefits of consistency regularization? In particular, I am not sure if I can follow the argument about the generalization properties of $\widehat{h}^{erm}$ stated without proof after Theorem 3, and I suspect it to be incorrect.
- The proof of Theorem 4 is inscrutable. I could not make my way through it although I consider myself to be on the more theoretical side of the spectrum. So, I would say the mathematical exposition of the paper is not accessible to the ICLR general audience.
- I could not follow what the goal of Section 4.3 is.
- Theorem 5 is stated for realizable model classes. It is known that in this case, the generalization bounds admit “fast” $O(1/n)$ rates rather than the usual $O(1/\sqrt{n})$ rates; see Tsybakov (2004); Steinke and Zakynthinou (2020). Hence, the upper bound in Theorem 5 is order-wise loose, and I am not sure how to interpret it.


The paper then continues to experiments where it is shown that consistency regularization outperforms data augmentation on three benchmarks, and the benefits are significant especially when labeled data is scarce. While I like the results, there are several major concerns here.
- First, it is unclear what form of regularizer function $\rho$ has been used to produce the results in the experiments section. Can you please clearly explain the regularizer for all experiments.
- As is well known and can also be seen in the experiments, when dealing with overparameterized non-convex neural models, choosing $\lambda \to \infty$ practically results in convergence to poor local minima that do not generalize. For example, this can be seen in Table 1 for $\lambda = 20$ and if $\lambda$ is even further increased, the resulting model would achieve the performance of a random classifier. This is a big discrepancy between the theoretical setup of this paper and the empirical setup, and a discussion around this shortcoming is warranted.
- In Table 1, can you please explain the drop of the performance seen for $\lambda =1$? One would intuitively expect that increasing $\lambda$ from 0, the performance would increase, and then it will plateau and then start to deteriorate. Hence, $lambda =1$ performance is counter-intuitive.
- It is claimed that “the DAC regularization gives a worse performance as it falsely enforces DAC regularization where the label of the augmented sample may have changed.” However, this is not substantiated through sufficient evidence. Either change the language to “might be explained” or provide more evidence.


Typos:
- pg 4, after (1): as an regularizer -> as a regularizer
- pg 17, Statement of Thm 8: satisfies -> satisfy

References
- Alexander B Tsybakov. Optimal aggregation of classifiers in statistical learning. The Annals of Statistics, 32(1):135–166, 2004.
- Thomas Steinke and Lydia Zakynthinou. Reasoning about generalization via conditional mutual information. In Proceedings of Thirty Third Conference on Learning Theory, volume 125, pages 3437–3452, 09–12 Jul 2020.


**Summary Of The Paper:**

Data augmentation is a common technique to improve generalization, especially when data is scarce. This paper introduces a theoretical framework for analyzing the effectiveness of consistency regularization when data augmentation is employed. In the limit, consistency regularization is akin to solving a constrained optimization problem with consistency constraints. The paper theoretically studies this limit for linear regression, logistic regression, and a two-layer perceptron with ReLU activation, and tries to characterize the benefits of consistency regularization beyond that of vanilla data augmentation. The paper then continues to experiments where it is shown that consistency regularization outperforms data augmentation on three benchmarks, and the benefits are significant especially when labeled data is scarce.

**Summary Of The Review:**

This paper develops a mathematical framework for consistency regularization. Several theoretical results and empirical results are provided. Overall, I like the general idea of the paper, however, there are several major concerns with the way it is executed. In particular, the major issue is that the theory and experiments are disjoint and do not support each other. (1) The theory is too tied to linear models and it is unclear how it could be extended beyond linear models. (2) The theory is only applicable to linear models with $\lambda \to \infty$. This regime is not even viable in the empirical world when we deal with neural models (as also evidenced in the experiments of the paper). (3) While the experiments are interesting, they are not conclusive on their own and lack important details. For example the regularizer function $\rho$ is not specified. Also, baselines for these benchmarks are not compared against. While there are many things that I like about the paper, it does not tell a coherent story, and especially one that would benefit the ICLR audience, and hence I recommend the paper to be rejected in its current form. I hope the authors can clarify some of the explicit comments/questions that I have raised in my review during the rebuttal.


====== update after author response ======

I would like to thank the authors for their extensive responses to my original comments as well as the follow-up comments, and also for their revisions to the paper which has improved the paper significantly. Thus, I am raising my score from 3 to 5. While some of my previous concerns are addressed, there are many remaining concerns that would require another careful/extensive revision and would require another round of review, which is why I still don't think the paper is ready to be accepted. I would like to emphasize that I like the general formulation of the problem, and the general positioning of the paper and I think the paper would be a nice contribution to the literature once the problems (especially with mathematical exposition) are fixed. Here are some explicit pointers for the authors:

- **Imprecise and inscrutable mathematical exposition:** There are lots of imprecise statements (which also **Reviewer s3Ue** complained about) still in the revised paper. For example, what does $\gg$ mean in Assumption 2 (page 15)? At the same time, the math is not followable. I still could not follow some of the proofs.


- **Hard to gain intuition/takeaways from theoretical results:** While the authors present several results, it is hard to understand takeaways from the developed theory. While they have addressed many concerns, several still remain. For example, RHS of (the revised) Thm 5 does not depend on any of the data augmentation parameters, such as $d_{\text{aug}}$ and $\alpha$. What is the takeaway from this theorem given that the same bound also applies to ERM and DA-ERM?

- **gap between theory and practice not discussed:** Although I explicitly gave feedback to address the gap between the theory and practice, it is not well discussed yet. For example, the theory is developed for $\lambda \to \infty,$ while there is a practical sweet spot for $\lambda$ in the experiments based on the optimization challenges. While I think that this gap should not be the reason to not accept the paper, it warrants a discussion around the results and their practical applicability, which is currently missing from the paper.

I hope the authors would find these comments useful in revising their paper for a future submission.

---

> ### Author Response · Authors · 2021-11-17
> **Response to fxf3 - PART II**
>
> *[4. Clarification of $d_{aug}$]*
>
> $d_{aug}$ can be intuitively understood as the number of **irrelevant** dimensions perturbed (thus the larger the better). For augmentations randomly generated from the original samples, when $\alpha$ is large (i.e., having more augmented samples), it is likely that there are more irrelevant dimensions perturbed. On the other hand, if the augmentation transformation itself is limited, i.e., can only perturb a limited number of dimensions, then when $\alpha$ goes to infinity, $d_{aug}$ is still upper bounded by the transformation’s perturbation limit.
>
> We consider the case where the augmented dataset is given. For the second line of proof on Page 13, the inequality holds for the given dataset and therefore there is no need to take expectation over $d_{aug}$. We have uploaded a revised version of our paper, which adds clarification on the randomness of $d_{aug}$ (see the revised Definition 2).
>
> *[5. Clarification of Thm 1 (Proposition 1 in the revision)]*
>
> We want to kindly emphasize that Theorem 1 is our general framework that anchors the concrete applications (e.g., linear regression, logistic regression, two-layer neural network, etc.) in the later sections. The generalization behavior depends on specific problems, and for each of these concrete applications, we present generalization bounds for the consistency regularization and further discuss the improvement in comparison to learning with ERM on the augmented training set. $C_l$ in Theorem 1 stands for the Lipschitz constant of the loss function.
>
> *[6. Clarification of Thm 3 and 4, and Section 4.3]*
>
> We appreciate the reviewer's suggestions on the concentration statements in the theorems, and we have made revisions accordingly.
>
> For the discussion after Theorem 3, the first term $\sqrt{\frac{d}{(\alpha+1) N}}$ corresponds to learning with $(\alpha + 1) N$ i.i.d drawn samples. However, when $\alpha$ is large, the augmented dataset will be very different from $(\alpha + 1) N$ i.i.d samples, as the augmentations fail to perturb a $(d - d_{aug})$-dimensional sub-space. Therefore when $\alpha$ is large, the generalization error can not be smaller than $O(\sqrt{\frac{d - d_{aug}}{N}})$ (as the ERM can only use the $N$ original samples to learn in the $(d - d_{aug})$-dimensional sub-space). We thus have the generalization bound being the max of the two.
>
> In Section 4.3, we demonstrate that, for classification problems, our framework also applies to a more general notion of data augmentations defined based on expansion that unifies the existing analysis and incorporates nonlinear effects of data augmentations.
>
> We agree with the reviewer that for Theorem 5, with the approximation error being 0, generalization can achieve the fast rate $O(1/n)$. However, instead of achieving a better rate, the main goal of Theorem 5 is to unify the existing theoretical works on the expansion-based augmentations for classification problems with our framework. Although the zero approximation error can be achieved in the simple illustrative scenario described in Theorem 5, this condition does not hold in general (e.g., for Theorem 9 in Appendix B.3 where we consider regularizing with finite samples, as well as in the related works (Wei et al. 2021, Cai et al. 2021)). Since without the zero approximation error assumption, the $O(1/\sqrt{n})$ rate is no longer easy to improve, we present the special case in Theorem 5 with the same rate for unity.
> We would also like to bring up that this is overall fine since results in learning theory focus more on the sample complexities instead of the rates. For Theorem 5, our main point is the $\widetilde O(K)$ sample complexity, instead of the fast/slow rate.
>
> *[7. Clarification for experiments]*
>
> The regularizer is the l2 distance between predictions for the original sample and augmented one, i.e., denote $f$ to be the model that maps from a sample to the probability of each class, the regularizer is $||{f(x) - f(x’)}||_2$.
>
> The theoretical analysis focuses on characterizing the solution of constrained optimization.
> The general practice is to use a regularizer instead of applying constraints directly (e.g, weight decay). The goal of Table 1 is to identify the best $\lambda$ that well regularizes the model, but not enforcing hard constraints.
>
> There is no drop from $\lambda = 0$ to $\lambda = 1$. Notice that the baseline in the original table 1 is ERM on the augmented dataset, which is not equivalent to the DAC with $\lambda = 0$ (since DAC with $\lambda = 0$ is not using any augmentations at all). We also add another experiment with $\lambda = 0$ and update the result into the paper. The testing accuracy is 62.82 (with $\lambda = 0$) compared to 68.63 (with $\lambda = 1$), which matches the intuition in the review.
>
> *[8. Clarification for $d'$]*
>
> We added the exact expression of $d'$ and an intuitive interpretation of $d'$ in our revision.

---

> ### Author Response · Authors · 2021-11-17
> **Response to Reviewer fxf3 - PART I**
>
> **Corresponding to the summary of review**
>
> We appreciate the reviewer’s questions and comments on some statements and experiments. We have made modifications based on these suggestions, and included additional experimental details in our revision (Response 6 and 7). To start with, we would like to kindly emphasize several critical points which hopefully can address the concerns highlighted in the summary of review:
>
> 1. By construction, the general framework in our theory is not limited to the linear regime. Moreover, when concretizing our general framework in Section 4, we discuss several applications of our framework to nonlinear models like the two-layer neural networks and general classifiers.
> 2. The regularization and constraint formulations of DAC are mathematically closely related, and therefore, studying the constraint formulation provides valid and valuable insights for understanding the effects of DAC regularization on the solution (Response 2). Additionally, in the experiments, we empirically demonstrate these effects of DAC regularization suggested by our theory.
>
> We sincerely hope that the reviewer will find these explanations helpful, and will reevaluate our work based on these clarifications.
>
> **Detailed Responses**
>
> *[1. Applications of our framework]*
>
> We want to emphasize that our main focus is the in-distribution generalization (i.e., no distribution shift between training and testing) of consistency regularization, a problem that has not been fully understood yet but has brought up increasing attention recently. The in-distribution generalization is one of the most critical properties of algorithms that have attracted abiding interest and induced numerous works in learning theory. Whereas the out-of-distribution generalization for settings like domain adaptation or domain generalization is an interesting but completely different topic from our main focus, albeit a brief discussion in Appendix B.4 on domain adaptation as a remark.
>
> *[2. On the regularizer]*
>
> First, we hope to kindly point out that realizability is not an issue. We want to highlight that throughout this work (as stated in the first 2 lines of Section 3.2, as well as for all the applications in Section 4), we consider the proper learning setting where the ground truth lies in the hypothesis class $h^* \in H$, and satisfies the constraints. We added a discussion on this before Definition 3.
>
> Second, we assume (on page 4 before Eq(1)) that $\rho$ is a metric (i.e., $\rho$ is non-negative and is zero if and only if the two inputs are equal) on the representation space, which is a stronger assumption than $\rho$ being a divergence. Therefore, when $\lambda \rightarrow \infty$, Eq (1) and (2) are equivalent.
>
> Third, we focus on characterizing properties of the solution for building a theoretical understanding of consistency regularization, instead of solving the optimization problem. In practice, we incorporate consistency regularization as Eq(1) with a proper regularization coefficient $\lambda$, and the problem is solvable via standard optimization algorithms (e.g., stochastic gradient descent).
>
> *[3. Data Augmentation Related]*
>
> It is generally challenging to consolidate all perspectives of different data augmentations into a single definition. In this work, we focus on studying the general augmentation mapping that involves a single input. Our definition is general in this setting and covers commonly used data augmentations (e.g., random crop, color jitter, rotation, flip). Mixup involves two samples and naturally will require a different definition, and we leave it as future work.
>
> Moreover, we want to point out that most common data augmentations in practice (e.g., cropping, flipping, rotation, color jittering, etc) are linear. In addition to covering these common augmentations, our definition provides a more general view in the sense that it does not assume linearity of the data augmentation maps. As a simple example, if an augmentation map arbitrarily twists the first coordinate of each input, this is a nonlinear mapping, and $d_{aug}$ is 1 according to our definition. Furthermore, in Section 4.3, for classification problems, we extend our framework to an even more general, expansion-based notion of data augmentations.

---

> > ### Comment · Reviewer_fxf3 · 2021-11-19
> > **No revised paper**
> >
> > Dear Authors:
> >
> > Thanks for your response! The paper does not seem to have been updated. Am I missing something? Also, I seem to be missing some of the responses to my specific comments/questions.
> >
> > Thanks,
> >
> > Reviewer fxf3

---

> > > ### Author Response · Authors · 2021-11-20
> > > **Revised Paper Uploaded**
> > >
> > > Dear Reviewer fxf3,
> > >
> > > Thank you for your reply! We were finalizing our revision and have now uploaded the revised paper. We also modified some of our responses (in both PART I and II) to match our revision.
> > >
> > > We hope our responses along with the paper revision can address all your concerns. Please let us know if any question needs additional clarification, and we will be happy to clarify. Thank you!

---

> > > > ### Comment · Reviewer_fxf3 · 2021-11-21
> > > > **Quick additional comments**
> > > >
> > > > Dear authors,
> > > >
> > > > Thanks for the revisions. While I have not yet had a chance to go through the entire revised paper and all of your responses, given that the period for revising the paper is fast approaching, I try to provide some quick additional comments below for the time being (while there is time for revising the paper) and will get back on your revisions later.
> > > >
> > > > - Sections 4.1 and 4.2 will be much stronger if the authors can empirically verify their findings with respect to DAC and DA-ERM.  I previously commented that the claim on generalization properties of DA-ERM stated without proof is not sufficient and the empirical justification could further help with substantiating the claim here. Also, directly comparing generalization upper bounds is misguided because an upper bound might be loose and it could be merely an artifact of the analysis and not the method.
> > > >
> > > > - Again, it would significantly strengthen the paper if the authors could empirically show how generalization in Sections 4.1 and 4.2 is affected by various parameters such as $\alpha$, $d_{\text{aug}}$, etc.
> > > >
> > > > - Please add a brief discussion on the limitations of the definition for Data Augmentation (e.g., that it does not apply to mixup) right where it is introduced.
> > > >
> > > > - I appreciate the authors' clarification on the focus of the paper on consistency regularization for in-distribution generalization. Two points are in order: (1) I think it would be good to explicitly mention this in the intro/setup; (2) I suspect DAC might offer benefits on out-of-distribution generalization as well, especially if data augmentation is designed for that, so a brief discussion in the conclusion might be beneficial.
> > > >
> > > > - I understand now how I had previously misinterpreted Table 1 as $\lambda = 0$ does not recover DA-ERM, and it degenerates to ERM. Quick comment/question: Why don't authors apply DAC on top of the DA-ERM loss instead of merely on ERM? How would that affect the theory and experiments? I understand that this is an open-ended question, so some partial (theoretical/empirical) answers are appreciated.
> > > >
> > > > - While I have not checked the proof of the revised Theorem 5, why does the RHS not depend on any of the properties of the regularizer or data augmentation ($\alpha$, $d_{\text{aug}}$, etc)? To me, this reads like a classic fast generalization rate.
> > > >
> > > > - If I understood correctly, the empirical regularizer used in experiments is $|| f(x) - f(x')||_2$. I wonder why the authors have made this choice, given that KL-divergence between the two distributions might be a more reasonable choice and also from an optimization perspective, $|| f(x) - f(x')||_2$ is non-smooth and can make the optimization problem more challenging.
> > > >
> > > > - Finally, I appreciate it if the authors can explain the performance drop in Table 1 for $\lambda = 20$. What happens if $\lambda = 100$ is used?
> > > >
> > > > Thanks,
> > > > Reviewer fxf3

---

> > > > > ### Author Response · Authors · 2021-11-21
> > > > > **Responses for the additional comments**
> > > > >
> > > > > Thank you for your quick comments! For the additional questions:
> > > > >
> > > > > *[Additional 1. On comparing the upper bound in Section 4.1 and 4.2]*
> > > > >
> > > > > We have uploaded a newer revision, added illustrative experiments to demonstrate the impact of $d_{aug}$ and $\alpha$ for logistic regression (see Appendix E). It shows the difference between DAC and DA-ERM comes from the algorithm itself, instead of analysis artifact.
> > > > >
> > > > > Moreover, we hope to kindly emphasize that it is very hard to characterize the exact generalization performance for a general function class (especially beyond linear regression). It is thus a common practice in theoretical work to compare the upper bounds to understand the difference in generalization performance (e.g., the most significant generalization work [1][2] that argued how sharpness and margin impact generalization by merely looking at the upper bound. [3][4][5] respectively showed the benefit of their algorithms in the problems of multitask representation learning, domain adaptation, and semi-supervised learning). We follow this convention for the applications in Section 4.
> > > > >
> > > > > *[Additional 2. Extensions to out-of-distribution setting]*
> > > > >
> > > > > In our paper, we mainly focus on a unique distribution $P^*$, from which we draw i.i.d. training samples and with respect to which we analyze generalization. We believe the effect of data augmentation (especially under different training algorithms) is not yet fully understood in this setting, even though this is the most conventional and fundamental setting. Therefore, we focus on this setting throughout our paper and will consider analyzing out-of-distribution (OOD) generalization as a next step.
> > > > >
> > > > > Nevertheless, we still have some results that demonstrate the advantage of DAC, in terms of better OOD generalization, for linear regression in the domain adaptation setting. Please see the discussion on page 6 at the beginning of Section 4 (which we re-emphasized in the revision thanks to your suggestion) and the detailed setup and analysis in Appendix B.4.
> > > > >
> > > > > *[Additional 3. DAC on top of the DA-ERM v.s. on ERM]*
> > > > >
> > > > > Theoretically, when DAC is enforced, the model will generate identical predictions for the original samples and augmented samples. Notice that the augmented samples also have identical labels as the original ones. Therefore when DAC is enforced, the ERM or DA-ERM will yield the same loss.
> > > > >
> > > > > *[Additional 4. On the interpretation of RHS of Theorem 5]*
> > > > >
> > > > > For the illustration purpose, we start with Theorem 5 in the main text with an ideal circumstance -- enforcing consistency over the population. In such case, with the class invariance and non-trivial expansion properties of the expansion-based data augmentation (Definition 4 in Section 4.3), the label has to be consistent within each ground truth class (Lemma 3 in Appendix B.3). Therefore, the problem can be reduced to generalization over a finite hypothesis class of size $K!<K^K$ in the realizable scenario, which leads to the $N = \widetilde O(K)$ sample complexity.
> > > > >
> > > > > We present a more careful (and technically involved) analysis, and the result is presented in Theorem 12 in Appendix B.3, where $\tau$ (Definition 5 in Appendix B.3) characterizes the strength of the expansion-based data augmentation map and affects the sample complexity for the consistency regularization.
> > > > >
> > > > > *[Additional 5. Further clarification on experiments]*
> > > > >
> > > > > Our consideration is that $l_2$ is a more general choice (e.g., settings where the DAC needs to be enforced on some intermediate layer) and therefore we choose to use $l_2$ even for the model's final output. We believe $l_2$ and KL-divergence will lead to similar performance (with proper scaling for $\lambda$), as they both effectively enforce the model to generate identical predictions for the original and augmented samples.
> > > > >
> > > > > Setting $\lambda = 100$ will lead to an even worse result than $\lambda = 20$. Intuitively, when the regularization is so strong, it will dominate the loss and will empirically make it very hard to learn from the ERM part of the loss.
> > > > >
> > > > > *[Additional 6. Highlights for our new revision]*
> > > > >
> > > > > In our updated revision, we included discussion on the limitation of our DA definition (i.e., not covering mixup), extensions to out-of-distribution, and illustrative experiments on logistic regression (showing the impact of $d_{aug}$ and $\alpha$ as requested).
> > > > >
> > > > > We hope our response can address all your concerns and kindly hope the score could be adjusted accordingly.
> > > > >
> > > > > [1] Neyshabur, Behnam, et al. "Exploring Generalization in Deep Learning."
> > > > >
> > > > > [2] Bartlett, Peter, Dylan Foster, and Matus Telgarsky. "Spectrally-normalized margin bounds for neural networks."
> > > > >
> > > > > [3] Maurer, Andreas, Massimiliano Pontil, and Bernardino Romera-Paredes. "The benefit of multitask representation learning."
> > > > >
> > > > > [4] Ben-David, Shai, et al. "A theory of learning from different domains."
> > > > >
> > > > > [5] Wei, Colin, et al. "Theoretical Analysis of Self-Training with Deep Networks on Unlabeled Data."

---

> > > > > > ### Comment · Reviewer_fxf3 · 2021-11-22
> > > > > > **some additional comments on the discrepancy between theory and experiments**
> > > > > >
> > > > > > Thanks for the quick response. One more thing that is in order is the discussion around the discrepancy between the theory and the practical experiments (which is also one of my main comments on the original submission). While the theory suggests that $\lambda \to \infty$ is desired for the best consistency regularization performance, that is not practically the case always. Adding a discussion around the practical limitations of DAC would benefit the reader.
> > > > > >
> > > > > > - In some experiments, the authors observed that when the augmented data could distort the original information, DAC can hurt the performance. This means that when the augmentation definition is not satisfied exactly, then we might end up with a practical sweet spot for $\lambda$.
> > > > > >
> > > > > > - The authors observed that $\lambda \to \infty$ makes the training of the network (especially learning from the ERM part) hard, resulting in a sweet sport for $\lambda < \infty$.
> > > > > >
> > > > > > - While I agree with the authors that when $\lambda \to \infty$, DA-ERM + DAC regularizer is no different from ERM + DAC theoretically, in practice, the former might give rise to a better performance especially given the practical sweet spot for $\lambda$ as per observation 2.
> > > > > >
> > > > > > - Given the practical sweet spot for $\lambda$, the performance of consistency regularization is also going to be dependent on where in the feature space DAC is enforced, or the function $\rho$ chosen for regularization, which is not discussed in the paper.

---

> > > > > > > ### Author Response · Authors · 2021-11-23
> > > > > > > **Responses for the additional comments II**
> > > > > > >
> > > > > > > Thank you for the questions. For the new comments:
> > > > > > >
> > > > > > > *[1. On the choice of finite $\lambda$ for empirical results]*
> > > > > > >
> > > > > > > From a theoretical perspective, with $\lambda \rightarrow \infty$, we focus on characterizing the solution of this constrained optimization. This leads to the novel viewpoint of casting the consistency regularization as function class dimension reduction. There is indeed a gap between our theory and practice as 1) it is generally hard to solve the constrained optimization; and 2) the augmentations may slightly change the label, and thus do not satisfy Definition 1 exactly.
> > > > > > >
> > > > > > > However, we believe such discrepancy does not invalidate our viewpoint of function class reduction - the finite $\lambda$ is actually equivalent to some relaxed version of the DAC that we studied theoretically. I.e., for any finite $\lambda$, there exists a constant $C(\lambda)$, such that the optimization with finite lambda is equivalent to the constraint $\varrho(\phi_h(x_i), \phi_h(x_j)) \le C(\lambda)$. In particular, when $\lambda \rightarrow \infty$, we have $C(\lambda) = 0$ which corresponds to our current theoretical result. More importantly, the relaxed version of DAC can also be viewed as a function class complexity reduction - although it will befuddle the results when quantifying such reduction in the relaxed DAC setting. We believe in theoretical analysis, it is more compelling to present a straightforward result with a clear conceptual message rather than a more relaxed but overly complicated and hard-to-interpret result.
> > > > > > >
> > > > > > > We will include a discussion on this in the future revision.
> > > > > > >
> > > > > > > *[2. On empirically combining DAC + DA-ERM]*
> > > > > > >
> > > > > > > Yes, we agree with the reviewer that the combination of DAC and DA-ERM can potentially give a better empirical result. This is a well-established empirical finding and is not really the focus of our point.
> > > > > > >
> > > > > > > *[3. On applying DAC on other representation space and other choices of regularization function]*
> > > > > > >
> > > > > > > We agree that it is interesting to see such results. However, the goal of our experiment is to demonstrate the benefit of DAC over DA-ERM to verify the theoretical analysis. The choice of different representation spaces and the regularization function is slightly off the topic.
> > > > > > >
> > > > > > > Further, we believe the benefit of a particular choice of representation and regularization function will depend on both the problem and the model. Therefore it is hard to give general conclusions on the choices of representation and regularization function.

---

### Official Review · Reviewer_s3Ue · 2021-11-02

**Correctness:** 3
**Technical Novelty And Significance:** 3
**Empirical Novelty And Significance:** 4
**Recommendation:** 6
**Confidence:** 4

**Main Review:**



### Strengths

1. The paper introduces a formal framework to study data augmentations and consistency regularization. The ideas are simple but novel, with several meaningful results and implications.
2. The paper is quite well-written, with the main ideas outlined clearly. The approach is well-motivated and seems to be built upon established literature. The review of related works is informative.

### Weakness

1. My main concern with the paper is about the results of Theorem 3 and Theorem 5, which appear (very much) weaker than they should be.
     - Theorem 5: "*For some* δ ∈ (0, 1)*, with probability at least* 1 − δ..." From a mathematical point of view, this is an extremely weak result (delta could be 0.999999). A regular generalization bound would have stated "For all delta>0, when n is large enough...".
     - Theorem 3: "*With constant probability, learning.."* The face value of the statement would be that the probability of the event stated in the theorem remains constant as n changes, which is obviously not true. I suppose that the author means "with probability bounded from below by a constant". Even in that case, this is a very weak result: wouldn't we want the probability to be at least close to one, or say, greater than 1/2?
     - The appendix, which is supposed to contain formal statements and proofs, also runs into that problem. Statements about "constant probability" appear in several places. The main assumption of the analysis, Assumptions 3:

          n ≫ ρ^4 (d − d_{*aug}* + log(1/δ)
         is also stated for **some** delta>0.
     - Theorem 4, in technicality, doesn't share the same problem. However, its result is only meaningful under Assumption 3, which is subject to the same constraint.

   If I understand correctly, the proofs of the paper might be adapted to support typical generalization bounds, but the manuscript in its current state didn't do that, and the results of the theorems do not make much sense.
2. The technical contributions of the other main results of the paper, Theorem 1 and Theorem 2, are limited. The proof (and result) of Theorem 1 is straightforward. The result of Theorem 2 for linear regression is interesting, but the proof involves standard computations.

### Other comments

- Throughout the manuscript and also in the appendix (which is supposed to contain rigorous proof), the author used several asymptotic notations such as <<, <~, and big-O notation, which may not be appropriate since (1) the analyses of the paper is non-asymptotic and involves many parameters (2) in rigorous non-asymptotic analyses, statements such as n ≥ N(\delta) is important but will be obscured using big-O notations
- A significant part of the proof in the appendix (10 pages) was to prove several results that correspond to a three-sentence remark in the main texts with no result statements. If the authors believe that this is a central point, a subsection in the main text should be created to provide the details. Otherwise, I suggest removing those parts out of the text (just the remarks) and the appendix, since those materials are not central to the content, totally not peer-reviewed, and shouldn't be associated with the paper if it is accepted.
- Example 1: "It can be verified that...". It would be helpful if a short verification is included in the appendix.
- Proof of Theorem 2: "The rest of proof is identical to standard regression analysis". This part is central to the result, so more details should be spelled out, or a reference should be given.
- Notations: rho is used both as a metric and as the sub-Gaussian constant. This is further confusing since they appear close to each other at times.
- Proof of Lemma 5:
    - The last statement on page 28 is incorrect. O(d) do not dominate d.
    - There is a typo in the equation "ρ2 ≥ ε2/16"

### Questions

- How are the regularizing constant (lambda =10) chosen in the last 4 experiments, while it is not the optimal value in the first one?


**Summary Of The Paper:**

### Summary

The paper introduces a statistical framework to analyze data augmentation to interpret consistency regularization as a way to reduce function class complexity. Building upon this framework: the paper

- shows that for linear regression, consistency regularization is more efficient than empirical risk minimization
- provides generalization bounds under consistency regularization for logistic regression, two-layer neural networks,
- provides a generalization bound for expansion-based data augmentations for multiclass classification



**Summary Of The Review:**

Overall, my vote for the paper is a (weak) reject. I think the framework of the paper is original and the ideas are intuitive with several interesting results and implications. On the other hand, the statements of some of the results are very weak (to an extent that they are not meaningful). The writing of the main text, as well as the proofs, are non-rigorous (which was partially an intention of the authors, at least for the main text, but might have affected the paper's mathematical quality).

==== Update after response and revisions

The revision addresses my main concern about the weakness of the results of Theorem 3 and 5. On the other hand, I'm still of the opinions that the heavy uses of asymptotic representations and the choices of present thee results of the main text in non-rigorous manner have affected the paper's mathematical quality. A few other concerns are left unaddressed. I thus raise my score from 5 to 6

---

> ### Author Response · Authors · 2021-11-17
> **Response to Reviewer s3Ue**
>
> **Corresponding to the Summary of Review**
>
> We appreciate the constructive suggestions from the reviewer on some potentially misleading phrasing in the theorem statements, and we have made modifications respectively. To start with, we would like to highlight some clarifications that correspond to the summary of the review:
> 1. The different forms of probabilistic bounds do not compromise the strength of statements asymptotically. Specifically, all our results hold with high probability or $1 - \delta$ probability for any $\delta$. (Response 1)
> 2. The goal of Theorem 1 (Proposition 1 in the revision) is to propose a general framework for anchoring the concrete applications that we analyze extensively in the later sections. For a general framework like this, we believe that its value depends on the resulting conclusions and insights, instead of the intermediate building blocks like tools in the proofs. (Response 2)
> 3. We follow the convention of statistical learning theory and present the results in asymptotic forms, which better presents the order-wise dependency on the number of samples, dimensions, etc. (Response 4.1)
> 4. Due to the page limit, it is common practice to present illustrative synopses in the main text and defer rigorous statements to the appendix. In general, this is not considered a compromise of quality. (Response 4.2)
>
> We sincerely hope that our response addresses all your concerns and kindly hope you could reevaluate our work based on these clarifications.
>
> **Detailed Responses**
>
> [1. On the statement of Theorem 3 and 5]
>
> We appreciate the reviewer's suggestions on the probabilistic statements of Theorem 3 and 5, and we have made revisions accordingly. Specifically, all our results hold with high probability or with at least $1 - \delta$ probability for any $\delta$, and we have revised our theorem statements according to the reviewer’s suggestions.
>
> [2. On the significance of Theorem 1 (Proposition 1 in the revision) and Theorem 2]
>
> Please see our general response for discussion on the significance of Theorem 1 and 2. Theorem 1 proposes a general framework for anchoring the concrete applications that we analyze extensively in the later sections (e.g., linear regression, logistic regression, multi-class classification with expansion-based data augmentations, etc.).
>
> Starting with an illustrative instantiation (i.e., linear regression), we tightly characterize the difference in generalization in Theorem 2 to provide some insight for the advantage of consistency regularization based on our framework.
>
> In general, despite our great appreciation of sophisticated techniques, we believe that the significance of theoretical results depends more on the conclusions and insights provided by the results, but less on their building blocks. We believe our main contribution is to characterize the consistency regularization as function class complexity reduction. This may seem natural in retrospect, but it was neither clear in the first place nor was presented in previous literature.
>
>
> [3. Clarification for choosing lambda]
>
> There is no statistically significant difference between $\lambda = 5$ and $\lambda = 10$, therefore choosing $\lambda = 5$ should give similar results and conclusions.
>
> [4. Response to other comments]
>
> 1. In statistical learning theory, people in general only care about the order-wise dependency on the sample size, dimension, etc., but not the leading constant. Therefore, although the analysis is non-asymptotic, the final results generally omit the universal constant.
>
> 2. We believe domain adaptation is another important setting that can benefit from consistency regularization, and we want to give an interpretation on this based on our framework. We leave this part of the results to the appendix because it is not the focus of this work.

---

### Official Review · Reviewer_QuHm · 2021-11-02

**Correctness:** 3
**Technical Novelty And Significance:** 2
**Empirical Novelty And Significance:** 1
**Recommendation:** 3
**Confidence:** 4

**Main Review:**

**Strengths**

This work presents several clear strengths.

  - This paper is one of the several works pioneering the discussions of data augmentation when used together with consistency loss, although several preceding works have been ignored [1, 2].

  - After the general form is introduced, several applications can be directly extended, which shows the potential of this work.

  - An interesting definition of the strength of the augmentation

**Weakness**

However, I also have several major concerns about this work, for example

 - If I understand correctly, Theorem 1 is essentially a re-use of the standard generalization error bound with a replacement of the original hypothesis space to the regularized hypothesis space, and then, the main argument is that since regularized hypothesis space is believed to be smaller, then the new bound is tighter. Overall, I don't think this result is significant enough, especially considering it takes a major position in this paper.

      - I think is too trivial to be considered as an important theorem of a publication at this level. It might be more appropriate to call it a lemma or a proposition of the theorem of the error bound with standard hypothesis space.
      - A smaller upper bound does not really say much of the performances, both of the bounds could be not tight, and even they are tight, some discussions of how smaller the regularized one is will be helpful.

  - While it is very interesting to see the definition of the strength of the augmentation, it does not say much in practice. Without a deeper or broader discussion, the definition seems to be a math brick for the theorem, discussions on how it is linked to the practice could be helpful, especially since the definition plays a central role as the major assumption in following theoretical results.
    - The intuitive explanation offered by the authors are not intuitive enough, in particular, for the examples of using rotation as augmentation (the authors state they use rotation in this paper), what are the intuitions of the definition, how different degrees of rotation corresponds to different strengths, and any numerical evidence can be reported?
    - Also, what's the intuitive explanation of Assumption 1?

  - With Theorem 1 being trivial, as discussed above, Theorem 2 is probably one of the most important result in this paper, yet there seems  also some issues.
      -. The proof of Theorem 2 critically depends on Assumption 1, this is not made clear in the main text.
      -. It might be better to put the full statement, at least the full definition of d' back to the main manuscript.


 - The empirical results do not offer any validation to the new theoretical discussion, but used to show that training with DAC can benefit in comparison to ERM, a fact that the community has known for a while.


[1] Invariance-inducing regularization using worst-case transformations suffices to boost accuracy and spatial robustness

[2] Squared ℓ2 Norm as Consistency Loss for Leveraging Augmented Data to Learn Robust and Invariant Representations

**Summary Of The Paper:**

This paper aims to offer a theoretical analysis of the training with data augmentation and associated consistency loss. While it is intuitive that training with data augmentation and consistency loss will help, this paper offers a theoretical justification of the intuitions. The simple framework (to view DAC as a hypothesis space complexity reduction technique) is neat and intuitive.

**Summary Of The Review:**

Overall, I feel like the theoretical discussion is not significant enough, and the associated experiments are also weak (in the context of theoretical papers)

---

> ### Author Response · Authors · 2021-11-17
> **Response to Reviewer QuHm**
>
> **Corresponding to the Summary of Review**
>
> We appreciate the reviewer’s questions and comments, based on which we have made careful modifications in our revision. We would like to begin our explanation by kindly bringing up several critical points which hopefully can address the reviewer’s general concerns:
> 1. Theorem 1 (Proposition 1 in the revision) is our general framework for interpreting consistency regularization that anchors the concrete applications discussed extensively in the later sections, instead of a different generalization bound. (Response 1)
> 2. The strength of data augmentations can be intuitively interpreted as how much (quantified by the number of dimensions) can the augmentations perturb the original training data. (Response 3)
> 3. The experiments study the impact of different numbers of training samples, different numbers of augmentations, and different augmentation strengths. It supports our theoretical result that DAC is more statistically efficient than ERM with a limited number of augmentations and proper augmentation strength (i.e., not altering the labels). (Response 5)
>
> We sincerely hope that the reviewer will find these explanations helpful, and will reevaluate our work based on these clarifications.
>
> **Detailed Responses**
>
> [1. On the significance of Theorem 1 (Proposition 1 in the revision)]
>
> We agree with the reviewer that Theorem 1 is a reminiscence of the classical generalization bound for bounded losses. The main purpose of Theorem 1 is to provide a general framework that anchors the concrete applications discussed extensively in the later sections (e.g., linear regression, logistic regression, multi-class classification with expansion-based data augmentations, etc.).
>
> [2. Clarification for comparing generalization upper bounds]
>
> As the reviewer said, the upper bound in Theorem 1 alone may not imply better performance and therefore we ground the general framework with concrete applications. For instance, in Theorem 2, with the linear model with a fixed design, we derive excess risk explicitly and further provide a numerical example (Example 1) empirically showing the generalization difference.
>
> Moreover, it is very hard to characterize the exact generalization performance for a general function class. It is thus a common practice in theoretical work to compare the upper bounds to understand the difference in generalization performance (e.g., the most significant generalization work [1][2] that argued how sharpness and margin impact generalization by merely looking at the upper bound. [3][4][5] respectively showed the benefit of their algorithms in the problems of multitask representation learning, domain adaptation, and semi-supervised learning). We follow this convention in Section 4 where we present upper bounds for excess risk, and also discuss the corresponding generalization bounds of the ERM on the augmented train set (e.g., after Theorem 3 and Theorem 4).
>
> [3. Clarification for the strength of the augmentation]
>
> Definition 2 brings an intuitive insight into the data augmentation strength -- how much (quantified by the number of dimensions) can the augmentations perturb the original training data. When the augmentations can perturb a large number of dimensions (i.e., $d_{aug}$ is large), we say the augmentation is stronger. We appreciate the reviewer's suggestion on numerical evidence. We added numerical illustration in Example 1 to show how different augmentations can have different $d_{aug}$s and their impacts.
>
> [4. Clarification on Assumption 1 and Theorem 2]
>
> For a linear regression model to be identifiable (i.e., having a unique solution), we need to assume that $\widetilde A(X)$ admits full column rank (Assumption 1). We have incorporated this clarification and the full statement of Theorem 2 into our revision.
>
> [5. Clarification for Experiments]
>
> Although DAC has been widely applied in various empirical works, its usage is usually coupled with other factors (e.g., applying DAC on unlabeled data for semi-supervised learning). The experiments that we presented are apples-to-apples comparisons of DAC and ERM with augmented data, and it clearly reveals the benefits of DAC over ERM, when everything else is kept the same.
>
> Coupled with our theoretical analysis, we empirically showed the impact of 1) different numbers of training samples; 2) different numbers of augmentations; 3) different augmentation strengths. Those experiments empirically verified our theoretical results.
>
> [1] Neyshabur, Behnam, et al. "Exploring Generalization in Deep Learning."
>
> [2] Bartlett, Peter, Dylan Foster, and Matus Telgarsky. "Spectrally-normalized margin bounds for neural networks."
>
> [3] Maurer, Andreas, Massimiliano Pontil, and Bernardino Romera-Paredes. "The benefit of multitask representation learning."
>
> [4] Ben-David, Shai, et al. "A theory of learning from different domains."
>
> [5] Wei, Colin, et al. "Theoretical Analysis of Self-Training with Deep Networks on Unlabeled Data."

---

### Author Response · Authors · 2021-11-17
**General Response**

We thank all reviewers for their careful reviews and insightful suggestions. It is encouraging to see that Reviewer QuHm appreciates our framework being “neat and intuitive” and our notion of data augmentation and the separation result between ERM and DAC being “interesting”; Reviewer s3Ue enjoys our ideas being “simple but novel, with several meaningful results and implications” and reviewer fxf3 likes our ideas and experiment results.

We would like to address some common concerns here.

First, despite our great appreciation of sophisticated techniques, we believe that the importance of theoretical results depends more on the conclusions and insights provided by the results, but less on their building blocks. We believe our main contribution is to characterize the consistency regularization as function class complexity reduction. This may seem natural in retrospect, but it is neither clear in the first place nor was previously presented in the literature.

Second, on the significance of Theorem 1 (Proposition 1 in the revision). The main purpose of Theorem 1 is to provide a general framework that anchors the concrete applications discussed extensively in the later sections (e.g., linear regression, logistic regression, multi-class classification with expansion-based data augmentations, etc.). Theorem 1 along with the various instantiations (Theorem 2 to Theorem 5) together demonstrate the statistical efficiency of consistency regularization.

Third, we hope to kindly emphasize that our framework and results are not limited to linear models. We do not require either the model or the data augmentation map to be linear. Concerning Theorem 4, although we characterize the benefits of DAC with a linear reduction, both the data augmentation and the model (i.e., two-layer ReLU network) can be non-linear. From the representation learning viewpoint, the first layer is learning a nonlinear representation of the data with consistency regularization, and then the second layer is a linear predictor on top of the nonlinear representation. Therefore we do not see our results being limited to linear models. Furthermore, in Section 4.3, for classification problems, we unify the existing theories and provide even more general notions for both the data augmentations and the model.

Lastly, we agree with reviewer QuHm that Theorem 2 is interesting - it presents a tight characterization for the generalization performance of DAC and ERM on the augmented dataset, and demonstrates how DAC is provably more efficient than ERM. However, we disagree that our paper is a repeat of a phenomenon that we already know - it focuses on presenting theoretical insight on why DAC is more efficient than ERM while keeping everything else the same.

Incorporating other suggestions in the review, we have made the following changes to our paper and have uploaded the revised version.

1. We updated our Example 1 (an illustrative numerical experiment) to include the results under different $d_{aug}$.
2. For Table 1, we added a baseline with $\lambda = 0$ (i.e., no DAC regularization).
3. We revised the statements of Theorem 3, 4, and 5 (as well as their correspondences in the appendix):
    3.1 We rephrased the probability statements with high probability tail bounds.

    3.2 For Theorem 5, we replaced the ‘slow rate’ $O(1/\sqrt{N})$ with the ‘fast rate’ $O(1/n)$ for the generalization bound.
4. More discussion on the data augmentation and strength of augmentation is included.

---

### Decision · Program_Chairs · 2022-01-20

**Decision:**

Reject

**Comment:**

This paper shows how constraining the representation to be invariant to augmentation shrinks the hypothesis space to improve generalization more than just introducing additional samples through augmentation. I agree with the reviewers that this is a novel, intuitive, and interesting finding. However, there were many technical and clarity issues with the original submission. These were partially addressed by the authors in the rebuttal. The reviewers appreciated the authors' efforts and commitment in the rebuttal, but my conclusion from our discussion that this paper requires another round of revisions. I hope the authors would follow the reviewer's comments, improve the paper, and re-submit.